# Gigapixel imaging with a novel multi-camera array microscope

**Eric E Thomson[1†], Mark Harfouche[2†], Kanghyun Kim[3], Pavan C Konda[3], Catherine W Seitz[1], Colin Cooke[3], Shiqi Xu[3], Whitney S Jacobs[1], Robin Blazing[1], Yang Chen[1], Sunanda Sharma[2], Timothy W Dunn[3], Jaehee Park[2], Roarke W Horstmeyer[2,3]*, Eva A Naumann[1]***

[1]Department of Neurobiology, Duke School of Medicine, Durham, United States; [2]Ramona Optics Inc, Durham, United States; [3]Biomedical Engineering, Duke University, Durham, United States

**Abstract** The dynamics of living organisms are organized across many spatial scales. However, current cost-effective imaging systems can measure only a subset of these scales at once. We have created a scalable multi-camera array microscope (MCAM) that enables comprehensive high-resolution recording from multiple spatial scales simultaneously, ranging from structures that approach the cellular scale to large-group behavioral dynamics. By collecting data from up to 96 cameras, we computationally generate gigapixel-scale images and movies with a field of view over hundreds of square centimeters at an optical resolution of 18 μm. This allows us to observe the behavior and fine anatomical features of numerous freely moving model organisms on multiple spatial scales, including larval zebrafish, fruit flies, nematodes, carpenter ants, and slime mold. Further, the MCAM architecture allows stereoscopic tracking of the z-position of organisms using the overlapping field of view from adjacent cameras. Overall, by removing the bottlenecks imposed by single-camera image acquisition systems, the MCAM provides a powerful platform for investigating detailed biological features and behavioral processes of small model organisms across a wide range of spatial scales.

**\*For correspondence:**
roarke.w.horstmeyer@duke.edu (RWH);
eva.naumann@duke.edu (EAN)

[†]E. Thomson and M. Harfouche equally contributed to this work.

## Editor's evaluation

This paper presents a valuable method of mesoscopic imaging for behavioral neuroscience, particularly of high potential in applications such as tracking behaving subjects in 3D arena simultaneously with a neural population activity readout. The technical and conceptual advances are based on solid presentations of the engineering and the pilot experiments. Readers of this paper are advised to first gain a deeper insight of its working principle as well as the consequent advantages and caveats of this method before applying it in their own labs.

## Introduction

Complex biological systems typically include processes that span many levels of organization across spatial scales that vary by multiple orders of magnitude (*Churchland and Sejnowski, 1988*; *Couzin and Krause, 2003*). For instance, processes such as schooling (*Wright and Krause, 2006*) or shoaling (*Larsch and Baier, 2018*) in fish involve social interactions between multiple individuals across tens of centimeters, but also include coordinated eye movements (*Harpaz et al., 2021*), pectoral fin changes (*Green et al., 2013*), and attendant fluctuations in neural activity (*Fan et al., 2019*) at the micrometer scale. Currently, no single imaging system allows for unrestricted access to each scale simultaneously, which requires the ability to jointly observe a very large field of view (FOV) while maintaining a high

spatial resolution. Hence, imaging systems typically make tradeoffs in their measurement process between FOV and resolution.

For example, many neuronal imaging methods typically require an animal to be head-fixed or tethered (*Ahrens et al., 2013*; *Harvey and Svoboda, 2007*), which restricts the natural behavioral repertoire (*Hamel et al., 2015*) and inhibits spontaneous movement (*Kim et al., 2017*; *O'Malley et al., 2004*). Closed-loop mechanical tracking microscopes have recently been developed in an attempt to address this challenge and have acquired impressive high-resolution optical measurements of behavior (*Johnson et al., 2020*; *Reiter et al., 2018*), and both behavior and fluorescence (*Cong et al., 2017*; *Kim et al., 2017*; *Nguyen et al., 2017*; *Susoy et al., 2021*; *Symvoulidis et al., 2017*). While such tracking strategies have recently revealed dynamics between persistent internal states within the brain during unconstrained behavior, for example. Such tracking strategies are powerful, but the optical apparatus has been limited to centering the field of view to a single organism. For instance, such tracking strategies have revealed the dynamics of internal states within the brain during unconstrained behavior (*Marques et al., 2019*), even when an individual is interacting with others (*Grover et al., 2020*; *Susoy et al., 2021*). Such systems are limited to tracking one animal at any given time, and do not effectively scale to the challenge of jointly observing the free movement and spatiotemporally varying interactions of many organisms at high resolution over a large area.

Alternatively, time-sequential scanning-based imaging approaches are routinely employed to capture image data from larger areas at high-resolution. Time sequential measurements can be obtained either by mechanical movement (*Potsaid et al., 2005*), the use of shifting microlens arrays (*Orth and Crozier, 2013*; *Orth and Crozier, 2012*) or via illumination steering as a function of time (*Ashraf et al., 2021*; *Gustafsson, 2005*; *Mudry et al., 2012*; *Zheng et al., 2014*). Unfortunately, such time-sequential imaging strategies are not able to *simultaneously* observe living processes at high resolution over a large area: for instance, the movement of multiple individuals can change position during the slow acquisition cycle, and thus generate ambiguities in the acquired data.

To simultaneously observe many freely moving organisms within a medium petri dish, well plate or an alternative arena that occupies a large FOV, a common strategy is to simply reduce imaging system magnification and resolution (i.e. to 'zoom out'; *Buchanan et al., 2015*). This is commonly performed in high-content screening experiments in toxicology (*Mathias et al., 2012*) and pharmacology (*Bruni et al., 2014*; *Rihel et al., 2010*), for example, where imaging many organisms is critical to discovery. However, such an approach necessarily must trade off spatial resolution for joint observation and can thus miss certain morphological features and behavioral signatures.

To simultaneously observe multiple organisms over a large area at high spatial resolution, another appealing idea is to increase the FOV of a standard microscope by producing a single large-diameter, large numerical aperture lens that captures images at the diffraction limit across the full FOV of interest. Unfortunately, geometrical optical aberrations increase with the surface area of a lens when its diameter increases (*Lohmann, 1989*). The number of corrective lenses required to compensate for these aberrations quickly increases, as does the overall lens system scale, in a nonlinear manner (*Lohmann, 1989*). These additional corrective lenses rapidly increase the size, weight, complexity and cost of imaging optics (*Brady et al., 2018*; *Mait et al., 2018*) which generally limits most commercially available microscope lenses to transfer less than 50 megapixels of resolved optical information to the image plane (*Zheng et al., 2014*). Complex lens systems specifically designed to address this challenge have not been able to increase this quantity beyond several hundred megapixels (*McConnell et al., 2016*; *McConnell and Amos, 2018*; *Sofroniew et al., 2016*). At the same time, the largest digital image sensors currently available also contain at most several hundred megapixels (*Canon, 2022*), which suggests a natural limit to single lens imaging systems precluding direct acquisition of gigapixel-scale image data. Recently, a 3.2-gigapixel digital camera for a specialized Large Synoptic Survey Telescope was developed, but its cost, $168 million, is prohibitive for most individual labs.

To overcome these challenges, and extend observational capabilities to the gigapixel regime, we designed and constructed an inexpensive, scalable *Multi Camera Array Microscope* (MCAM) (*Figure 1A*), a system whose cost scales linearly with the number of pixels. This design is significantly less expensive than other available large-area, high-resolution imaging systems as it leverages relatively inexpensive components (see Methods). The MCAM collects data in parallel from an arbitrarily sized rectangular grid of closely spaced modern CMOS image sensors (each with ~$10^7$ pixels that are 1 µm in size, *Appendix 1—figure 1*), which are currently produced in large volumes for the cell phone

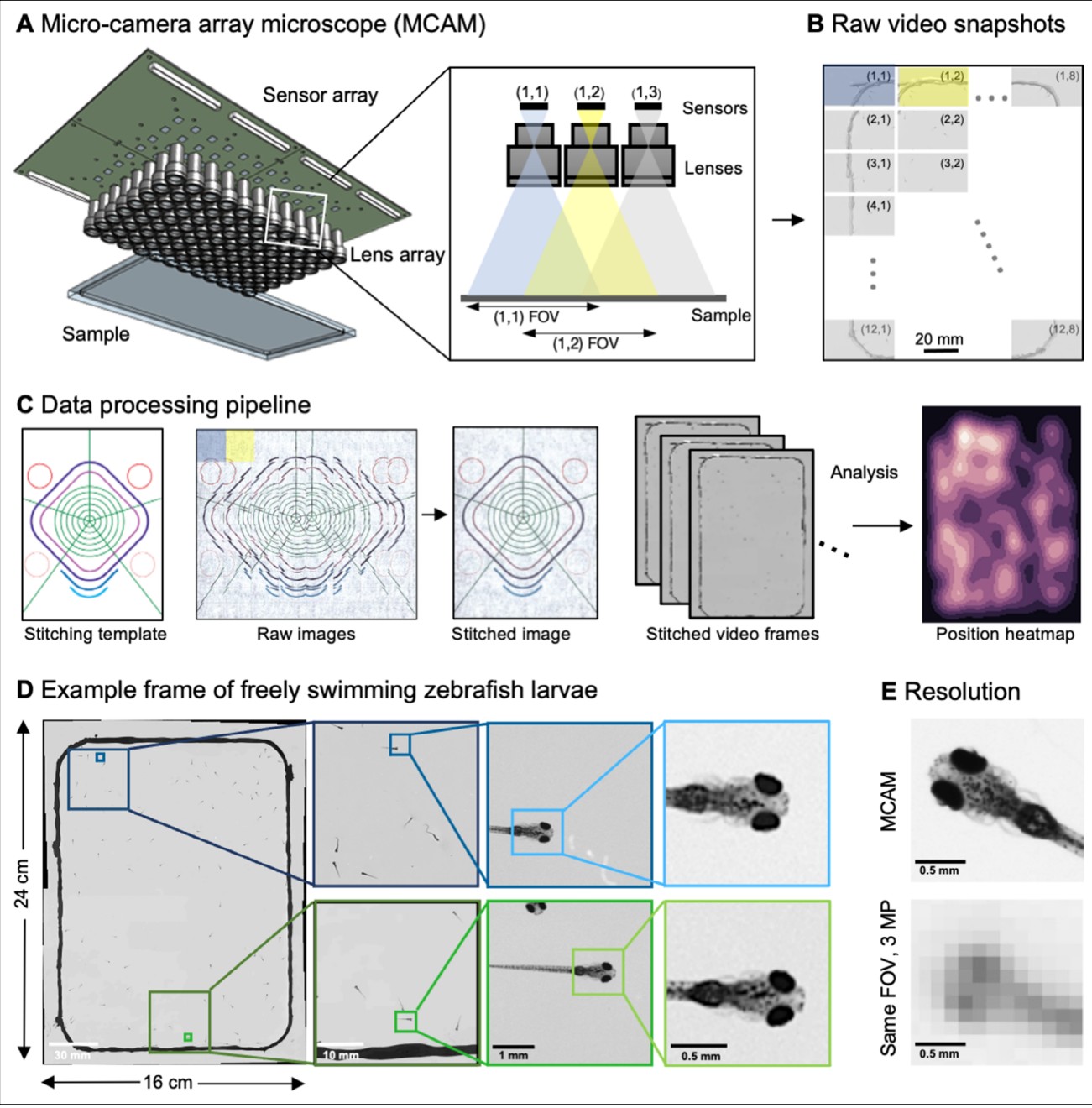

**Figure 1.** Architecture of the multiple camera array microscope. (**A**) Schematic of MCAM setup to capture 0.96 gigapixels of snapshot image data with 96 micro-cameras. Inset shows MCAM working principle, where cameras are arranged in a tight configuration with some overlap (10% along one dimension and 55% along the other) so that each camera images a unique part of the sample and overlapping areas can be used for seamless stitching (across 5 unique test targets and all included model organism experiments) and more advanced functionality. (**B**) Example set of 96 raw MCAM frames (8x12 array) containing image data with an approximate 18 µm two-point resolution across a 16x24 cm field-of-view. (**C**) Snapshot frame sets are acquired over time and are combined together via a calibration stitching template to reconstruct final images and video for subsequent analysis. An example of an analysis is a probability heatmap of finding a fish within the arena at a given location across an example imaging session. (**D**) Example of one stitched gigapixel video frame of 93 wildtype zebrafish, 8 days post fertilization, selected at random from an hour-long gigapixel video recording. (**E**) Resolution comparison of an example larval zebrafish captured by the MCAM and from a single-lens system covering a similar 16x24 cm FOV using a 3 MP image sensor (but capable of high-speed video capture).

camera market and are thus readily available in many varieties and at low-cost. Each MCAM image sensor is outfitted with its own high-resolution lens to capture image data from a unique sample plane area (*Figure 1A*, zoom in). The MCAM sensors and lenses are arranged such that their fields-of-view partially overlap with their immediate neighbors (*Appendix 1—figure 2*), which ensures that image data are jointly acquired from a continuous area (*Figure 1B*). Such a parallelized image acquisition geometry (*Brady et al., 2018*) can, in theory, be scaled to produce an arbitrarily large FOV. Here, we present MCAM designs with up to 96 cameras that allow us to record information-rich bright-field gigapixel videos of multiple freely moving species, including Carpenter ants (*Camponotus* spp.), fruit flies (*Drosophila melanogaster*), nematodes (*Caenorhabditis elegans*), zebrafish (*Danio rerio*), and the amoeboid slime mold (*Physarum polycephalum*) (*Nakagaki et al., 2000*) at 18 µm full-pitch optical resolution (i.e. the ability to resolve two bars with a center to center spacing of 18 µm) over a 384 cm$^2$ FOV. To help distill these large image sets into salient and actionable data, we adapted a variety of modern image analysis tools, including a convolutional neural network-based object tracker, to work at gigapixel scale. As a unique approach to address the FOV/resolution tradeoff, the MCAM offers a scalable platform that overcomes current imaging bottlenecks and allows us to simultaneously observe phenomena spanning from the microscopic to the macroscopic regimes.

## Results
### Multi camera array microscope (MCAM) design and image processing

To achieve large-FOV microscopic imaging, we first designed and constructed a modular MCAM imaging unit that consisted of a 4×6 rectangular array of 24 individual Complementary metal–oxide–semiconductor (CMOS) image sensors (10 megapixels per sensor, 1.4 µm pixels) and associated lenses (25 mm focal length, 0.03 numerical aperture, 12 cm$^2$ FOV per lens). We integrated the 24 sensors onto a single circuit board (*Appendix 1—figure 1*), and directly routed all data through a custom-designed control board containing a set of field-programmable gate arrays (FPGAs). The FPGAs deliver the image data to a single desktop computer via a USB3 cable, where the raw high-resolution video from all 24 sensors is saved for subsequent post-processing. The total pixel count per snapshot from each 24-camera array is 0.24 gigapixels.

To verify that our modular MCAM architecture could scale to image acquisition from much larger regions, we joined four such 24-sensor units together in a two-by-two arrangement to form a *gigapixel imaging system* (*Figure 1A*, *Video 1*; *Appendix 1—Video 1*), which streamed video data from 96 cameras simultaneously (0.96 gigapixels total raw data; *Appendix 1—figure 1E*) to a single desktop computer. We successfully operated this final system at full pixel count at up to 1 Hz frame rate yet note that implementation of on-sensor pixel binning can yield higher frame rates at reduced resolution. Future enhancements to data transmission arrangements could also directly yield higher frame rates (see Discussion). A continuous recording session at 0.74 Hz lasted almost one hour resulted in 2.5 TB of unstitched data (2649 frames, 962 MB per frame) and 1.17 TB stitched data (442 MB per image). The system exploits the rapid current read/write speeds for individual solid state drives on a single personal computer (achieving approx. 1 GB/s), and marks a significant speedup from previously published gigapixel imaging systems (*Brady et al., 2012*).

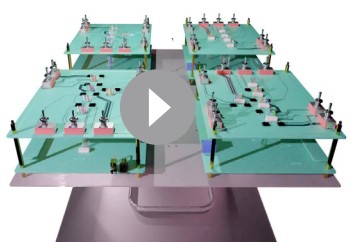

**Video 1.** 3D rendering of MCAM. Three-dimensional rendering of the MCAM hardware. Corresponds to discussion of *Figure 1A*.
https://elifesciences.org/articles/74988/figures#video1

To assess the MCAM resolution, we imaged a custom-designed resolution test target extending across the 96-camera MCAM's entire 16 cm x 24 cm FOV (*Appendix 1—figure 2*) to demonstrate an 18 µm full-pitch optical resolution cutoff, which corresponds to a maximum spatial frequency cutoff of 147 line pairs/mm under incoherent illumination (*Ou et al., 2015*). Importantly, we emphasize that the above resolution performance is independent of the number of cameras included within the MCAM, thus providing a high-resolution imaging platform that can scale to an arbitrary FOV when more cameras are included in the array. System axial resolution defined via the

Rayleigh criterion is 0.28 mm, while the axial sample range that supports a 50% reduction in spatial resolution from that measured at the plane of best focus is 2.54 mm (Methods).

Processing vast amounts of data – sometimes millions of images – from multiple cameras presents many unique opportunities and challenges compared to standard single-camera imaging systems. For all data, we first saved raw images to disk followed by a two-step offline preprocessing procedure. We performed standard flat-field correction to compensate for lens vignetting and illumination brightness variations within and between cameras (Methods). For each MCAM snapshot in time, we then digitally stitched the images acquired by all the micro-cameras together into a single composite image frame (*Figure 1C*, *Appendix 1—figure 3*, *Appendix 1—figure 4*, Methods). Note that while stitching is not technically required for many subsequent image analysis tasks, it is often useful for visual presentation of acquired results, such as organism trajectory assessment or multi-organism location measurement (e.g. heatmap in *Figure 1C*). Following preprocessing, images then entered various application-specific algorithms, as outlined below.

## High-resolution, large-FOV imaging of large groups of larval zebrafish

After validating the hardware and implementing the basic image acquisition workflow, we used custom Python code to produce gigapixel videos from the image data streamed directly to four separate solid state drives. To record the behavior of ~100 wildtype, 8 days post-fertilization (dpf) zebrafish (*Video 2*), we placed them into a 22×11 cm 3D-printed (resin on glass) behavioral arena (*Figure 1D*, **Methods**). As each gigapixel image frame contains information about the zebrafish at multiple spatial scales, from the social structure of the population down to fine-grained morphological features of individual fish, we were able to resolve details like pectoral fin position and the distribution and morphology of individual melanophores, that is, the pigmented skin cells on the body, in addition to arena-wide swim trajectories (*Figure 1D*). In an MCAM image, a zebrafish (8 dpf) is captured by 9699±2758 pixels (n=93 zebrafish), which is significantly higher than in standard imaging systems. For example, when using a single 20 MP camera with the same large-FOV, fish would be represented by less than 400 pixels, and even less in older juvenile fish (*Romero-Ferrero et al., 2019*). For a standard 3 MP camera sensor imaging a comparable FOV, a fish occupies less than ~100 pixels (*Figure 1E*).

We adapted a convolutional neural network (CNN)-based object detection algorithm (Faster-RCNN) to automate the detection of zebrafish in stitched MCAM images (*Figure 2A*), which are too large for training or inference on existing graphical processing units (GPUs). We used transfer learning with a data augmentation pipeline that included supplementing training data with synthetically generated occluder fish (similar to that described in *István et al., 2018*; *Sárándi et al., 2018*), because occlusion was relatively rare in the original data. After training, this yielded a network that could accurately detect individual zebrafish even when they occluded one another (*Appendix 1—figure 5I*, *Video 3*). We discuss the method for training and using the network in some detail in Appendix 1: Large-scale object detection pipeline; (see also *Appendix 1—figure 5*, Methods).

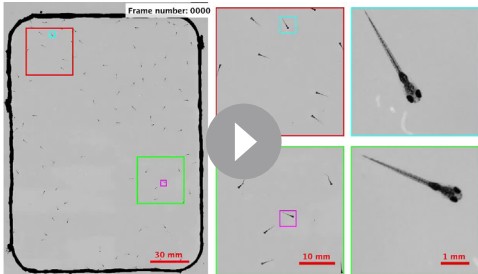

**Video 2.** High-resolution, large-FOV larval zebrafish. Brightfield video of freely swimming, 8-day-old zebrafish, recorded at approximately 1 frame per second for 1 hr (2000 frames shown here). Randomly selected zoom-in locations at two spatial scales demonstrate ability to resolve individual organism behavior at high fidelity across full 16x24 cm field-of-view. Corresponds to discussion of *Figure 1D*.
https://elifesciences.org/articles/74988/figures#video2

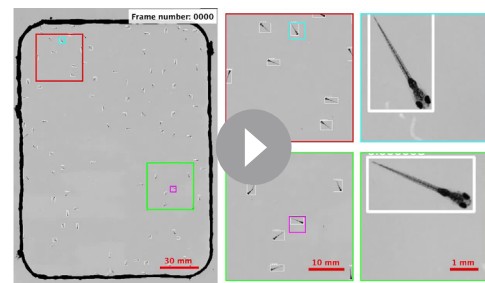

**Video 3.** Example output of gigadetector algorithm. Brightfield video of freely swimming, 8-day-old zebrafish, showing output of automated organism detection software marked as a bounding box and label around each larva. Corresponds to discussion of *Figure 2A*.
https://elifesciences.org/articles/74988/figures#video3

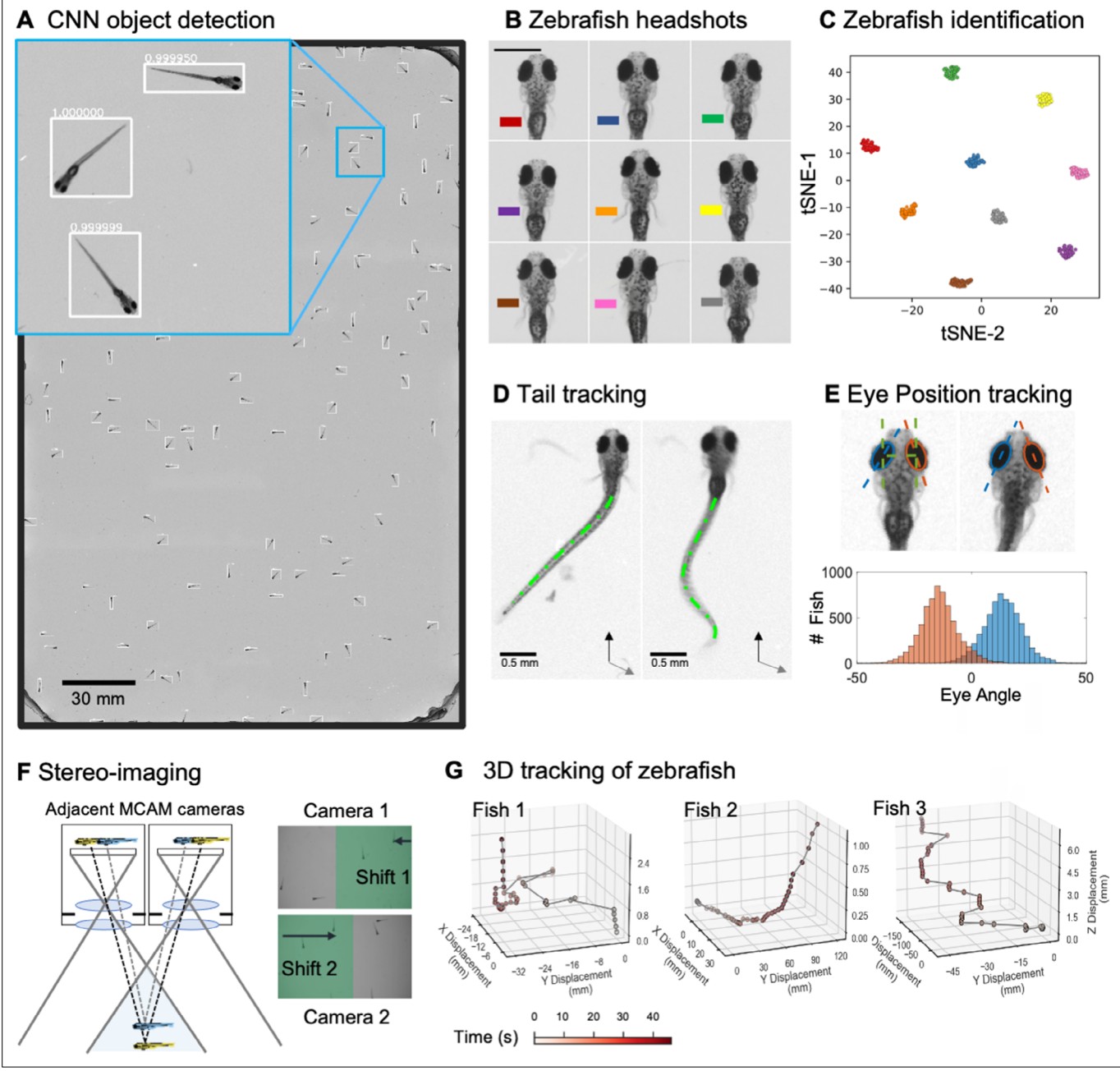

**Figure 2.** Gigapixel high-resolution imaging of collective zebrafish behavior. (**A**) Example frame of stitched gigapixel video of 130 wildtype zebrafish, 8 days old (similar imaging experiment repeated 11 times). An object detection pipeline (Faster-RCNN) is used to detect and draw bounding boxes around each organism. Bounding boxes include detection confidence score (for more details see Methods). Each bounding box' image coordinates is saved and displayed on stitched full frame image. Zoom in shows three individual fish (blue box). Scale bar: 1 cm. (**B**) Headshots of nine zebrafish whose images used to train a siamese neural network (see Methods, *Appendix 1—figure 6*). Colors indicate the fish identity used to visualize performance in (**C**). Each fish has a distinct melanophore pattern, which are visible due to the high resolution of MCAM. Scale bar: 1 mm. (**C**) Two-dimensional t-SNE visualization of the 64-dimensional image embeddings output from the siamese neural network, showing that the network can differentiate individual zebrafish. For 9 zebrafish, 9 clusters are apparent, with each cluster exclusively comprising images of one of the 9 fish (each dot is color-coded based on ground truth fish identity), suggesting the network is capable of consistently distinguishing larval zebrafish (62 training epochs, 250 augmented images per fish). (**D**) Close-up of three zebrafish with tail tracking; original head orientation shown in gray (bottom right). (**E**) Automated eye tracking from cropped zebrafish (see Methods for details); results histogram of eye angles measured on 93 zebrafish across 20 frames. (**F**) Optical schematic of stereoscopic depth tracking. The depth tracking algorithm first uses features to match an object between cameras (top) and then calculates the binocular disparity between the two cameras to estimate axial position with an approximate resolution of 100 µm along z (verified in calibration experiments at 50 axial positions). (**G**) Example 3D trajectories for 3 zebrafish from recorded gigapixel video of 93 individuals, with z displacement estimated by stereoscopic depth tracking.

After automatically detecting zebrafish in each MCAM frame, analysis was performed on image data extracted from within each bounding box containing individual fish, vastly reducing memory and data handling requirements (*Figure 2A*, inset). To track individual zebrafish over time, we exploited the high spatial resolution of the MCAM to consistently recognize individuals by morphological differences (*Figure 2B*), which has not been shown before in zebrafish younger than 21 days old. Specifically, we trained a Siamese neural network (*Bertinetto et al., 2021*; *Appendix 1—figure 6*) that was able to learn to track the same zebrafish across multiple frames (*Figure 2C*). Given the high spatial resolution of the MCAM, this network had unique access to idiosyncrasies in melanophore patterns, structural and overall body morphology when learning to distinguish individual animals. In addition, we used the MCAM's resolution to identify behavioral details of each zebrafish by performing tail tracking (*Figure 2D*) and eye position measurement (*Figure 2E*), which yielded aggregate statistics over populations of zebrafish during free swimming.

In addition, the MCAM's multi-lens architecture offers novel measurement capabilities as compared to standard single-lens microscopes. One example is stereoscopic depth tracking. By using rectangular image sensors (16:9 aspect ratio), the MCAM provides images from adjacent cameras that overlap approximately 50% in the x-direction and <10% in the y-direction to facilitate stitching (*Appendix 1—figure 2*). This overlap ensures that any point in the sample plane is simultaneously observed by at least two cameras and allows us to use positional disparities from matched features (i.e. stereoscopic depth measurement techniques) to calculate the depth of objects within adjacent camera image pairs (*Figure 2F*, Methods). Briefly, after matching features within adjacent frames using a scale-invariant feature transform, pixel-level lateral feature displacements generate depth estimates using known MCAM imaging system parameters (e.g. inter-camera spacing and working distance). Using this stereoscopic depth measurement approach yields approximately 100 µm accuracy in depth localization over a >5 mm depth range (see *Appendix 1—figure 7* and associated Appendix 1 for experimental support). To demonstrate this stereoscopic capability of the MCAM, we tracked the 3D trajectory of zebrafish within the entire swim arena (*Figure 2G*). Example 3D tracking revealed typical quantifiable signatures of fish spontaneous swim trajectories along spatial axes typically inaccessible (*Bolton et al., 2019*) without dual view camera systems.

## Application to multiple model organisms

The MCAM is a general, scalable imaging platform that can be used in the study of multiple model organisms. *Figure 3* demonstrates gigapixel recording of nematode (*Caenorhabditis elegans*) behavior in three different strains: wild type (N2), an unpublished transgenic line (NYL2629) that expresses GFP and mCherry in motor neurons, and a mutant (unc-3) line (*Brenner, 1974*) that has a loss-of function mutation in the unc-3 gene (unc-3 is expressed in motor neurons and is activated when worms become hungry [*Prasad et al., 1998*]). We plated these worms on a large, square (24.5cmx24.5cm) petri dish and employed a CNN-based segmentation algorithm (Appendix 1) Videos captured ~58,000 worms per frame, captured at about 450 pixels per worm within a typical gigapixel video (averaged across 20 frames, single frame in *Figure 3A–C*, top). Surprisingly, both the unc-3 mutants, and especially the NYL2629 strain show previously unreported swarming behavior. Approximately 3 days after being transferred to a new nutrient lawn, the NYL2629 strain began to form tight swarms that tiled most of the petri dish in a semi-periodic fashion (*Figure 3B*), as has been observed in other strains (*de Bono and Bargmann, 1998*). Based on the hypothesis that this could be driven by partial loss of function in the reporter strain, we then tested for such effects more directly in an unc-3 mutant, where we observed qualitatively different swarming behavior: large wavefronts of worm aggregates (2–3 cm wide) (*Figure 3C*). Notably, such aggregative behavior was not observed in the wildtype line (*Figure 3A*). Using computer vision algorithms to track worms and detect swarm formation shows how such large-scale differences in behavior can be observed with the MCAM (*Video 4*, *Video 5* and *Video 6*). Recent research on collective behavior has demonstrated that swarming behaviors are modulated by various factors, including oxygen levels and animal density (*Demir et al., 2020*; *Ding et al., 2019*). While further investigation of the underlying causes and spatiotemporal features of swarming in these different strains is needed, our data clearly demonstrate that the MCAM is a useful tool for the quantitative study of factors that control social clustering and swarming in *C. elegans* (*Figure 3E*).

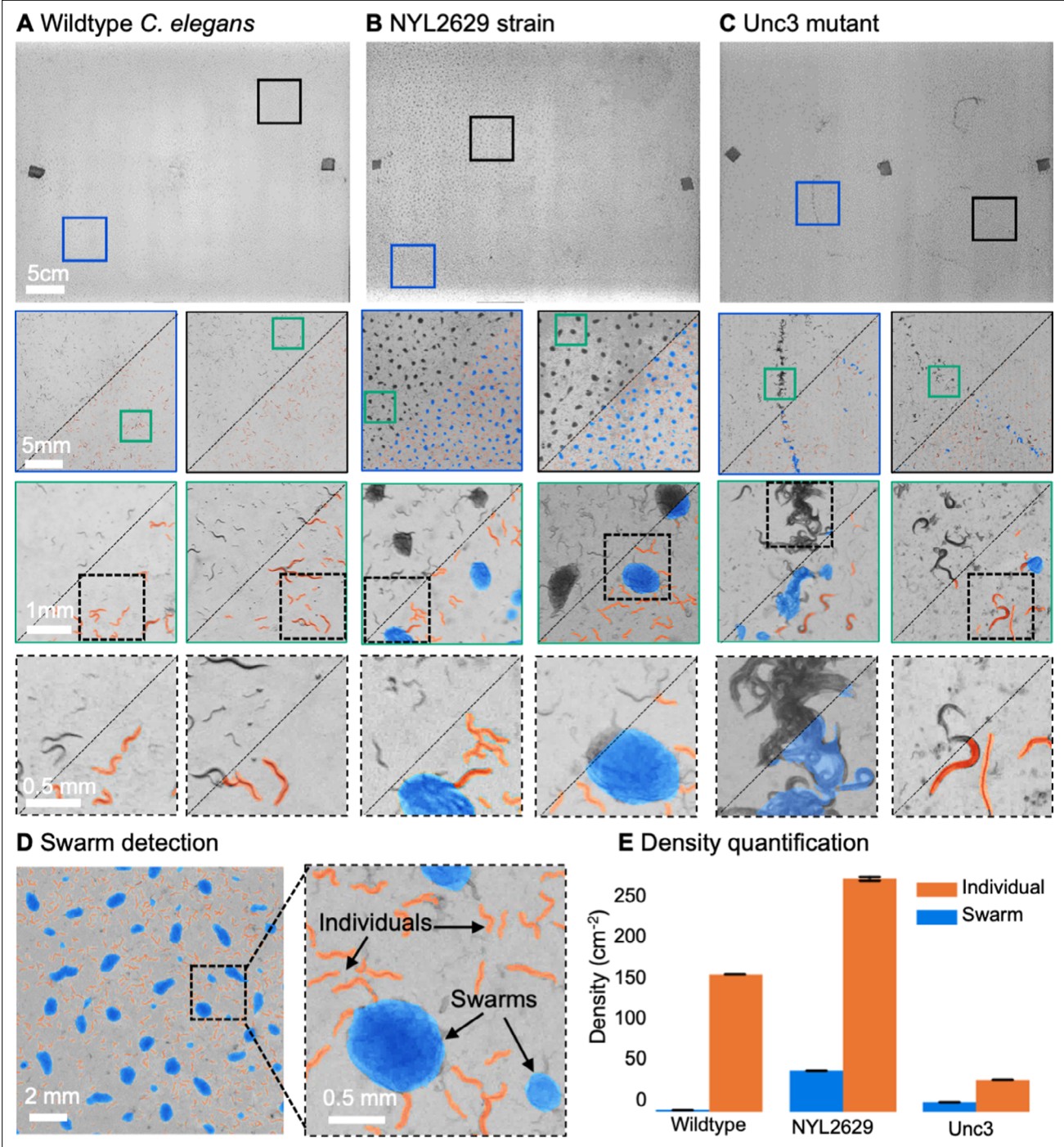

**Figure 3.** MCAM imaging of swarming behavior in *C. elegans*. (**A**) Wild type *C. elegans* (N2 line) spread across the entire arena uniformly. Consecutive zoom ins show no significant swarming behavior. Approx. 30,000 organisms resolved per frame (similar imaging experiment repeated 4 times). (**B**) NYL2629 *C elegans* strain shows a periodic tiling of the petri dish over the majority of the arena as *C. elegans* form swarm aggregates that can be automatically detected. Approximately, 58,000 organisms resolved per frame. (**C**) Unc3 mutant *C. elegans* exhibit large wavefront swarms of activity, without the periodic tiling seen in the NYL2629 strain. Unc3 genotype inhibits a significant swarming behavior and causes swarms, annotated in blue, all over the entire arena. Our system provides sufficient resolution across a large field-of-view to enable studying behavioral patterns of *C. elegans*. In all cases the imaging was performed after day 3 of starting the culture. *Videos 4–6* allow viewing of full video. Entire gigapixel video frames viewable online, ~8000 organisms resolved per frame. (**D**) Segmentation of swarms (blue) and individual *C. elegans* (orange) using a U-shaped convolutional network allows the automatic creation of binary identification masks of both individual worms and swarms. After segmentation, we used a pixel-connectivity-based method to count the number of objects (worms or swarms) in each gigapixel segmentation mask. (**E**) Bar graph quantifying density of worms within and outside of swarms for the different strains in A-C. Error bars indicate S.E.M across the arena. Mean worm density (number of worms/

*Figure 3 continued on next page*

Figure 3 continued

cm$^2$) for wildtype 154+–0.254, NYL2629 262 +- 1.126, Unc3 36+–0.203. Mean swarm density (swarms/cm$^2$) for wildtype 1.85+–0.01, NYL2629 45.92 +- 0.073, Unc3 10.65+–0.025. While more wild type worms than Unc3 are plated, wild type worms form far fewer swarms (p=2.3 x 10$^{-66}$, Cohen's d=0.96) 1 sided t-test. For more details, see Methods.

Another species that displays stark behavioral and morphological variations on spatial scales spanning multiple orders of magnitude is the slime mold *Physarum polycephalum* (*Alim et al., 2013*). This single-celled, multinucleate protist exhibits complex behavior at large spatial scales. For instance, its pseudopodia will envelope nutrients in its environment, foraging over a search path spanning many hundreds of square centimeters (*Gawlitta et al., 1980*; *Rodiek and Hauser, 2015*). Exploiting this, we observed the well-known ability of the slime-mold to traverse mazes, though on much larger spatial scales (*Figure 4A*; *Video 7*; *Nakagaki et al., 2000*). Also, using time-lapse imaging with a 24-camera MCAM array (0.24 gigapixels), we acquired data over a 46-hr period, showing the cumulative growth and exploration of a standard 10 cm diameter petri dish that hadbeen seeded with oatmeal flakes (*Video 8*). Further, by switching to single-camera acquisition mode to acquire at a higher frame rate, we observed the well-known cytoplasmic flow within individual plasmodia (*Durham and Ridgway, 1976*), involving particulate matter of tens of square microns, flowing across multiple centimeters (*Video 9*).

Ant colonies show a highly distributed multi-scale spatial structure, with many individuals spending time in their base colony, and the foragers leaving the colony to retrieve food (*Mersch et al., 2013*). To demonstrate how the MCAM could be useful to study such behavior, we imaged behavior in the Carpenter ant (*Camponotus pennsylvanicus*) (*Hamilton et al., 2011*). We observed dramatic changes in behavioral arousal with change in light levels. Carpenter ants are nocturnal animals, and their activity levels are very tightly locked to ambient light (*Narendra et al., 2017*; *Sharma et al., 2004*). When we alternated light and dark cycles with small groups of ants, we observed clear fluctuations in activity levels with light cycles, including the dissolution of tight clusters of 'sleeping ants' with increased movement (*Figure 4B*), but also social interactions and grooming (pulling on antennae with legs; *Video 10*).

The fruit fly, *Drosophila melanogaster* has become a benchmark model system for the genetic analysis of behavior (*Datta et al., 2008*), development (*Tolwinski, 2017*), and neurophysiology (*Mauss et al., 2015*). We recorded fly behavior at multiple developmental stages using the MCAM. The MCAM allowed us to observe large-scale social behavioral patterns in adult fruit flies, such as face-to-face interactions, and mutations in the wings such as curling (*Figure 4C*, *Video 11*). At the same time, we could see small features of individual flies, such as hairs on individual legs, grooming behavior, and the sex of each animal (*Figure 4C*, inset). This demonstrates the potential use for the MCAM in large-scale genetic screening in addition to the study of collective behavior that involves interactions on very fine spatial scales.

Finally, the MCAM technology also enables a new suite of imaging experiments for fluorescent

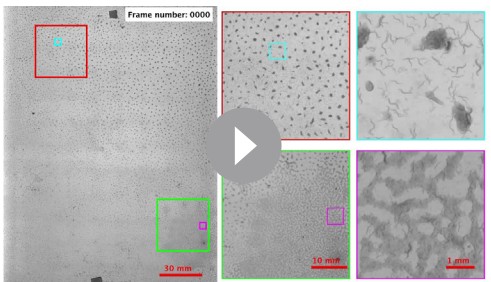

**Video 5.** NYL2629 *C. elegans* strain. Brightfield video of *C. elegans* NYL2629 strain. Randomly selected zoom-in locations at two spatial scales demonstrate ability to resolve individual organisms and jointly observe macroscopic behavioral phenomena, in particular the marked tiling of the dish with periodic swarms of *C elegans*. Corresponds to discussion of *Figure 3B*.

https://elifesciences.org/articles/74988/figures#video5

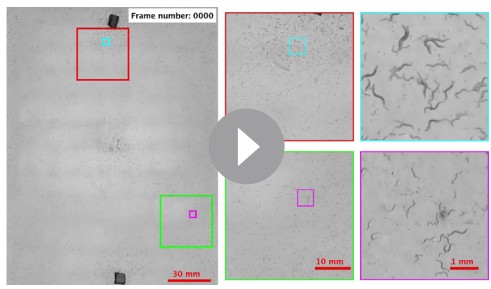

**Video 4.** Wild type *C. elegans.* Brightfield video of *C. elegans* wild type (N2) imaged with MCAM-96. Randomly selected zoom-in locations at two spatial scales demonstrate ability to resolve individual organisms. Corresponds to discussion of *Figure 3A*.

https://elifesciences.org/articles/74988/figures#video4

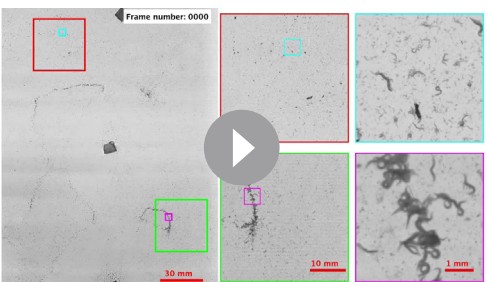

**Video 6.** *C. elegans* super-swarming behavior. Brightfield video of *C. elegans* unc-3 mutant imaged with MCAM-96. Randomly selected zoom-in locations at two spatial scales demonstrate ability to resolve individual organisms and jointly observe macroscopic super-swarming behavioral phenomena. Corresponds to *Figure 3C*.
https://elifesciences.org/articles/74988/figures#video6

specimens, such as tracking freely moving organisms that emit fluorescence via neuronal genetically encoded calcium indicators, such as GCaMP6 (*Dana et al., 2016*). After outfitting the MCAM imaging system for epi-fluorescence imaging, we captured wide-field fluorescence video of multiple freely moving *Drosophila* larvae and freely swimming zebrafish. We also utilized the MCAM's overlapping fields-of-view to jointly capture two fluorescence spectral channels simultaneously (i.e. both red and green fluorescence emission) within its recorded videos, which may be applied to ratiometric fluorescent image analysis in future studies. The methodology and associated fluorescence data is presented in Appendix 1 imaging with the MCAM.

## Discussion

We report on the development of a multi-camera array microscope (MCAM), constructed from closely tiling multiple high pixel-count CMOS image sensors that are each relatively inexpensive ($30 per sensor for this prototype - see Methods). Our parallelized imaging design overcomes the limitations of single-objective and single-sensor systems, allowing the acquisition of high-resolution images over a large, in principle infinite FOV. While prior designs have considered multiple microscopes to image distinct regions in parallel (*Weinstein et al., 2004*), or via cascaded multiscale lens designs (*Fan et al., 2019*) that can yield light-field-type imaging measurements (*Broxton et al., 2013*; *Levoy et al., 2009*), the MCAM provides a novel, flexible arrangement that can monitor phenomena in freely moving organisms at multiple spatial scales. The number of cameras can be varied to produce different FOVs with varying frame rates, depending on the targeted imaging application. The working distance can also be changed to jointly increase or decrease the degree of inter-camera FOV overlap to unlock novel functionality. By adopting a parallelized approach to video acquisition, the MCAM removes a longstanding bottleneck of current single-camera microscope designs to provide researchers with greater flexibility, in particular for observing model organisms at high-resolution in an unconstrained manner.

We have also demonstrated two unique capabilities of the MCAM's multi-lens architecture – 3D tracking and simultaneous acquisition of fluorescence images from two unique spectral bands (see example in *Appendix 1—figure 8*). The former functionality opens up a new dimension to behavioral analysis, while the latter could lead to novel fluorescence imaging experiments with additional development and refinement. There are likely a wide variety of alternative exciting functionalities that MCAM overlapped sampling can facilitate, such as dense height map estimation at high resolution over large areas via photogrammetric methods (*Zhou et al., 2021*).

Our demonstrated MCAM technology exhibited several limitations that can be addressed with future development. First, while we were only able to sample gigapixel images at 1 Hz for up to 60 min – a significant increase in speed and duration from previous published efforts (*Brady et al., 2012*) – many biological applications could benefit from much faster acquisition rates. For instance, our current acquisition speed of 1 Hz is too slow to determine specific patterns in swim kinematics in freely swimming zebrafish (*Marques et al., 2018*). While additional fast low resolution imaging from below could partially resolve this 1 Hz speed limitation (*Johnson et al., 2020*), we note here that a number of strategies are available to increase sampling rates. The most straightforward paths to directly increasing the MCAM frame rate are (1) utilizing an updated version of USB that now allows approximately 10 X faster data transfer, (2) distributing data transmission across multiple desktop computers, (3) adopting an alternative FPGA data transfer scheme (e.g. replacing USB data transmission with a faster PCIe protocol), and (4) executing on-chip pixel binning, image cropping, and/or lossless compression to increase temporal sampling rates at the expense of spatial sampling. Various combinations of the above strategies are currently being explored to facilitate true video-rate (24–30 Hz) MCAM imaging and to achieve significantly higher frame rates and improve cross-camera snapshot synchronization.

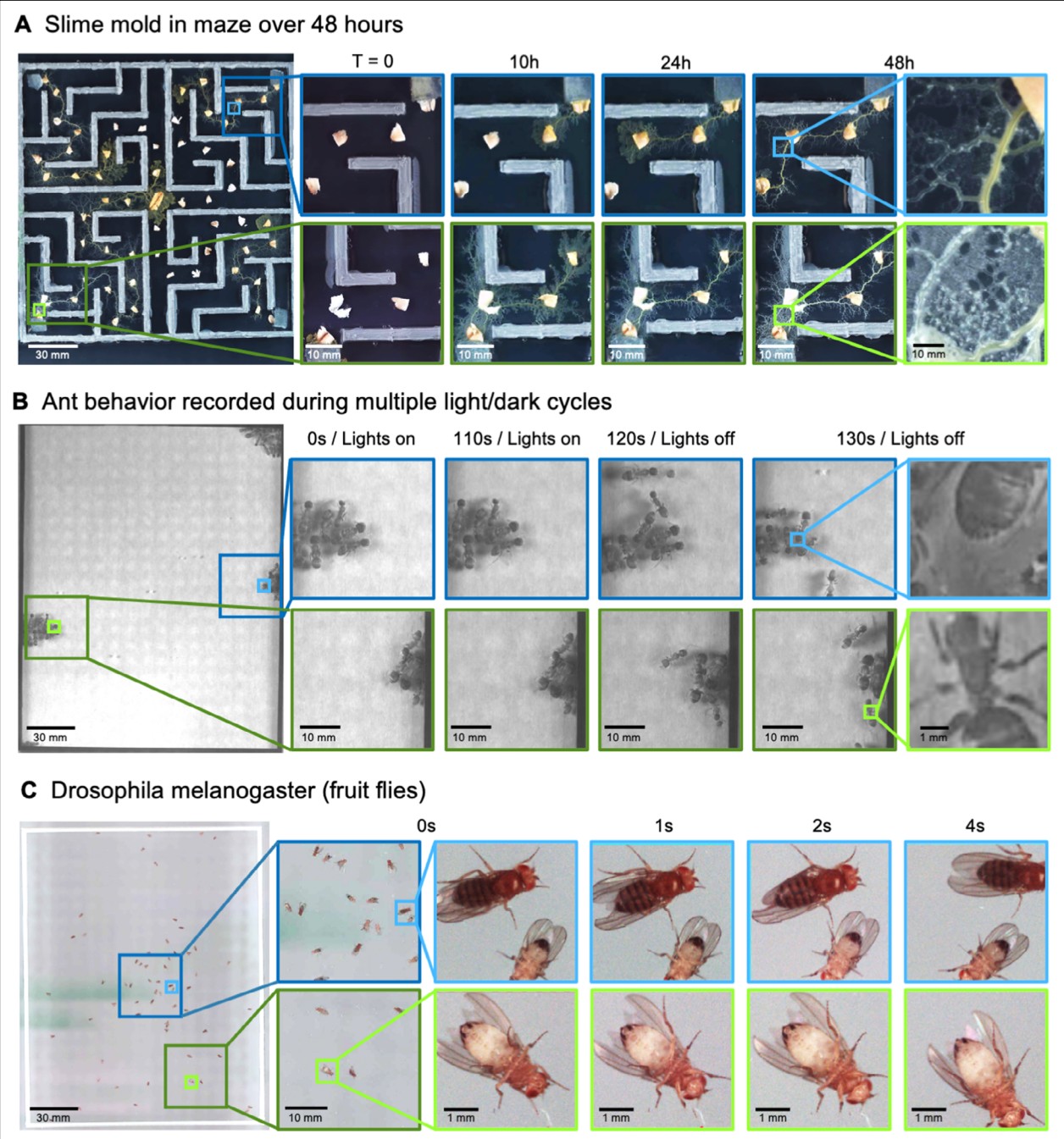

**Figure 4.** MCAM imaging of behaviors in multiple small organisms. (**A**) Gigapixel video of slime mold (*P. polycephalum*) growth on maze (16.5 cm per side). Four mold samples were taken from a seed inoculation on another petri dish and placed in four corners of the maze. Recording for 96 hr shows the slime mold growth within the maze (*Video 7*). highlighting the MCAM's ability to simultaneously observe the macroscopic maze structure while maintaining spatial and temporal resolution sufficient to observe microscopic cytoplasmic flow (imaging experiments with *P. polycephalum* repeated 5 times). (**B**) Gigapixel video of a small colony of Carpenter ants during light and dark alternations. Overall activity (velocity) of ants measured with optical flow increased during light off versus light on phases. Full MCAM frame observes Carpenter ant colony movement and clustering within large arena, while zoom-in demonstrates spatial resolution sufficient for leg movement and resolving hair on abdomen (imaging experiments with Carpenter ants repeated 2 times). See *Video 10*. (**C**) Adult *Drosophila* gigapixel video frame showing collective behavior of several dozen organisms at high spatial resolution, with insets revealing fine detail (e.g. wings, legs, eyes) during interaction (imaging experiments with adult *Drosophila* repeated 2 times). See *Video 11*.

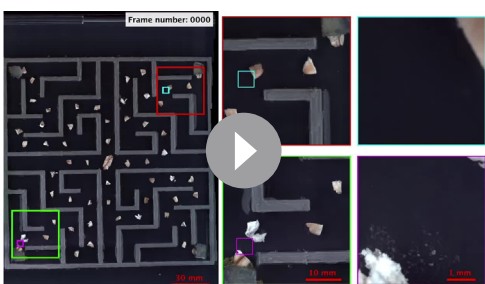

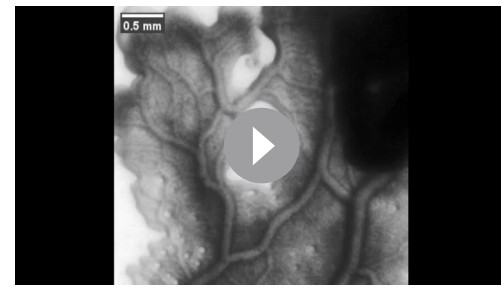

**Video 7.** Slime mold maze traversal. Video of slime mold *Physarum polycephalum* traversing a custom-designed maze from 4 starting locations, imaged in time-lapse mode with the MCAM-96 over the course of 96 hr. Zoom-ins show ability to observe pseudopodia at high resolution during growth and foraging. Corresponds to discussion of *Figure 4A*.
https://elifesciences.org/articles/74988/figures#video7

**Video 9.** Slime mold cytoplasmic flow demonstration. This was recorded running the MCAM in single camera mode at 10 Hz. Cytoplasmic flow, and its reversal, within individual plasmodia, is clearly observable.
https://elifesciences.org/articles/74988/figures#video9

A third promising direction is to dynamically pre-process acquired images on the included FPGAs before directing them to computer memory to lighten data transfer loads. Such pre-processing could range from acquiring only from cameras that contain objects of interest to on-the-fly tracking and cropping of individual organisms or features of interest.

Second, while we have demonstrated the MCAM's fundamental ability to acquire wide-field fluorescence video of dynamic organism movement (see Supplement), CMOS sensors with increased sensitivity and a higher imaging NA could increase fluorescence signal-to-noise ratio and potentially facilitate fluorescence detect at the cellular scale. Improvements in ratiometric acquisition and the potential addition of optical sectioning could also increase fluorescence signal fidelity, most notably during rapid organism movement. With appropriate effort, the MCAM architecture could soon unlock high-quality measurement of functional fluorescence activity during free organism movement over truly macroscopic areas, thus yielding a new imaging platform for neuroscientific experiment.

Finally, while the demonstrated lateral resolution in this work (18 µm full-pitch) is typically not sufficient to resolve individual cells unless they are sparsely labeled, tighter camera packing and updated imaging optics may unlock the ability to observe cellular-scale detail in future designs. We see the infrastructure embodied in the current MCAM implementation as an auspicious initial scaffolding upon which we can build such natural extensions to provide researchers with maximally salient image data spanning the microscopic and macroscopic regimes.

While this work primarily focused on examining how the MCAM can enhance model organism experiments, our new imaging technology can also be applied to a variety of other applications.

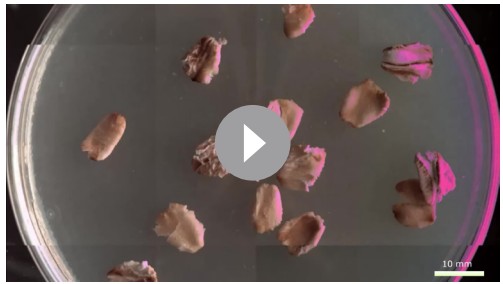

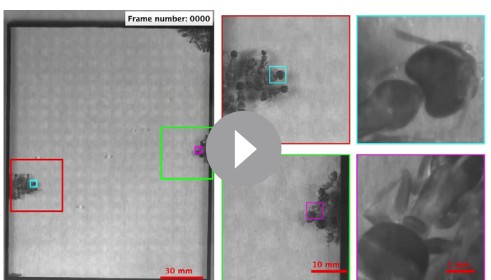

**Video 8.** Slime mold petri dish exploration. Time-lapse Video (image taken every 15 min) shows single slime mold growth from center of petri dish over 46 hr. Petri dish is seeded with multiple oatmeal flakes, and you can observe the characteristic large-scale exploratory behavior of the slime mold over time, as well as finer-scale plasmodia structure.
https://elifesciences.org/articles/74988/figures#video8

**Video 10.** Carpenter ant behavior under multiple light/dark cycles. Brightfield imaging of collective Carpenter Ant behavior using MCAM-96, with ambient lighting sequentially turned on and off every two minutes (6 repetitions). Randomly selected zoom-in locations shown at two spatial scales at right. Corresponds to discussion of *Figure 4B*.
https://elifesciences.org/articles/74988/figures#video10

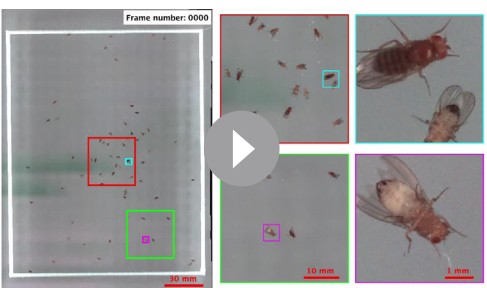

**Video 11.** *Drosophila* adult bright-field imaging demonstration. Adult *Drosophila* during spontaneous free movement and interaction across full MCAM-96 FOV (approx. 16x24 cm), with randomly selected zoom-in locations at two scales demonstrating ability to monitor macroscopic and microscopic behavioral phenomena. Corresponds to discussion of *Figure 4C*.
https://elifesciences.org/articles/74988/figures#video11

Large area, high-resolution imaging is important in industrial inspection of manufactured goods (*Yang et al., 2020*) and materials (*Ngan et al., 2011*), as well as semiconductor metrology (*Huang and Pan, 2015*). For instance, such methods are required for defect detection on semiconductor wafer surfaces. It can also unlock new functionalities in imaging large pathology specimens, such as whole-mount brain slices (*Axer et al., 2011*). Biological researchers may also be able to utilize the MCAM platform to monitor bacterial colonies (*Shi et al., 2017*) and cell cultures over macroscopically large areas. Such a wide variety of potential applications provides many new directions to guide development of array imaging in the future.

In summary, our MCAM architecture fits within a growing body of computational imaging hardware (*Mait et al., 2018*) that is specifically designed not for direct human viewing, but instead with the assumption that data post-processing will be used to extract relevant information for subsequent analysis. While novel display technology currently enables visual assessment of the MCAM's gigapixel videos that can be many Terabytes in raw size, we anticipate that simultaneously improving processing software (e.g. 3D organism tracking, machine learning-assisted analysis) will prove critical to fully realize the microscope's benefits. As image sensor technology continues to improve with such automated processing moving on-chip (*Amir et al., 2018*; *Posch et al., 2014*), we anticipate that our parallelized imaging architecture can help overcome the standard tradeoff between resolution and field-of-view while at the same time yield manageable information-rich datasets to provide new insight into complex biological systems.

## Methods
### MCAM electronics
Our first constructed gigapixel MCAM system consists of 96 individual complementary metal oxide semiconductor (CMOS) sensors, lenses, and associated read-out electronics that transmit recorded image data from all sensors to computer memory for offline processing. The sensors and lenses (i.e. micro-cameras) are arranged in an aggregate 8x12 array and composed of 4 individual 24-sensor MCAM imaging units placed in a 2x2 array, as sketched in *Appendix 1—figure 1*. Each CMOS sensor (Omnivision, OV10823) contains 10 megapixels (4320x2432) and a pixel size of 1.4 µm for a 6 mm x 3.4 mm active area. In the prototype images included here, each CMOS sensor collected color information using a Bayer mosaic of red, green, and blue filters that are arranged in a pattern collecting 50% green, 25% red and 25% blue light. The sensors were placed at a 19 mm pitch, and their rectangular active area allows us to capture overlapping field-of-view information primarily along one dimension for stereo-based depth tracking and dual-channel fluorescence imaging. The imaging system's total number of recorded pixels per frame is 0.96 gigapixels. Several results presented in this paper were from a single MCAM imaging unit that contains only 24 individual micro-cameras in a 4x6 layout (same micro-camera pitch and design), producing 240 recorded megapixels per frame.

When operating in full-frame mode, each CMOS image sensor delivers image data to an associated field-programmable gate array (FPGA) via custom-designed electronic circuitry. Every six CMOS sensors are controlled by one FPGA. Hence, a total of 16 sensor-connected FPGAs controlled and routed the 96-camera MCAM image data to the computer. Each of these 16 FPGAs routes data from the six-sensor clusters to one of four USB3.0 lines. To control data transmission, a 'main' FPGA assigns each group of four FPGAs to port data (from 24 micro-cameras total) over one USB3.0 line (*Appendix 1—figure 5*). The four USB3.0 lines, each routing data from 24 CMOS sensors, are connected to a single desktop computer (Dell, Intel(R) Xeon(R) W-2133 3.6 GHz CPU), then saves

directly to a set of 1TB solid-state drives. Each solid state drive has peak write speeds of more than 1 GB/s (3 x Samsung 970 Pro and 1 x Samsung 960 Pro).

The MCAM offers two modes of operation. First, it can operate in a 'full-frame' mode that records video data from all 96 micro-cameras at an aggregate frame rate of approximately 1 frame per second (see timing diagram in *Appendix 1—figure 4B–C*). The per-frame exposure time for each individual camera can be reduced to approximately 1ms to limit any per-frame motion blur. While we are able to collect data from each CMOS sensor at much higher frame rates, our current electronic design offers a limited data transmission bandwidth to computer memory when in 'full frame' mode, which we address by FPGA-routing one image frame per second from each of the 96 micro-cameras and ignoring the remaining captured frames pre-transmission. Such transmission bandwidth limitations have been resolved in our next generation of MCAM (see Discussion).

The MCAM architecture also offers the flexibility to read data from a smaller number of sensors at a higher frame rate. For example, a 'single-sensor' mode of operation allows us to selectively record video data from up to four electronically assigned micro-cameras at a frame rate of approximately 12 frames per second (*Appendix 1—figure 5*). By allowing rapid evaluation of image quality during sample placement and focus adjustment, 'single-sensor' mode is critical during device setup and can lead to dynamic spatiotemporal data acquisition schemes (e.g. fully digital organism tracking at 12 fps) in future designs. Current design efforts are focused on significantly improving frame rates both in full-frame and single-sensor mode to enable electronic tracking of rapid organism movement (see Discussion).

## MCAM optics

Each of the 96 cameras utilizes a multi-element 12 mm diameter mount lens (Edmund Optics 58–207, 25 mm focal length, 2.5 f-number), with an 18 mm maximum outer diameter that currently sets the MCAM's inter-camera spacing (to 19 mm). As shown in *Appendix 1—figure 2*, these lenses offer approximately 18 µm full-pitch resolution at a 150 mm working distance that extends across the entire per-sensor image FOV with the employed 10 Megapixel rectangular image sensors (1.4 µm pixels, 3.4 mm x 6 mm). Three quantities are used to characterize MCAM depth-of-field ($D$). First, we use the system's theoretical Rayleigh criterion for axial resolution under incoherent imaging conditions to estimate $D_{Ray}$ = 0.28 mm. Second, we experimentally measured the axial sample range across which lateral resolution remains within a factor of 2 of its maximum value at the plane of best focus as $D_{Res}$ = 2.54 mm, which we found to be a useful measure of practical axial imaging extent. Third, we experimentally measured the full-width half-maximum of image contrast (defined as the average mean image gradient magnitude of a resolution target) as a function of axial distance from the plane of best focus as $D_{Con}$ = 1.95 mm. Data and experimental details regarding these measurements are in Appendix 1: MCAM resolution analysis and verification and *Appendix 1—figure 2*.

We selected a working distance of approximately 150 mm – 160 mm to ensure that the images recorded by adjacent micro-cameras overlap sufficiently to enable seamless image stitching (see Methods, Software: Image Stitching). By using rectangular image sensors with a height/width ratio of 1/1.8, the current MCAM design offers approximately 52% image overlap along the longer sensor dimension and <10% overlap along the shorter image dimension. This overlapping field-of-view configuration is diagrammed in *Appendix 1—figures 2 and 4* and guarantees that any point within the object plane (neglecting edge regions) is imaged by at least two micro-cameras. As noted above, this overlapped imaging configuration enables us to track the depth of various objects within the scene using stereovision-based depth tracking algorithms (see **Methods**, Software: Depth detection), as well as obtain dual-channel fluorescence measurements.

## MCAM wide-field illumination

A number of different bright-field illumination configurations were used to create the MCAM data presented here. For the freely moving zebrafish and *C. elegans* results, we relied on a set of four different 32x32 LED arrays, each 13x13 cm in size with a 4 mm LED pitch (Adafruit Product ID 607) placed approximately 15 cm beneath the sample plane, with a diffusive screen placed approximately 4 cm above the LED array. While these illumination LED arrays offer individual control over the specific color channel (red, green, and blue) and brightness of each LED within the array, the experiments presented here typically illuminated all LED color channels at uniform brightness (i.e. uniform white

illumination). For epi bright field illumination for example, for imaging slime mold, ants, and adult *Drosophila*, we used LED arrays held off to the side of the behavioral arena by a 3D printed adapter specifically designed for these purposes. The adaptor was mounted to a plastic flexible gooseneck that we could adjust the light to maximize brightness and minimize glare.

## MCAM component costs

Here is a summary of component prices for the prototype gigapixel MCAM imaging system at the time of purchase. We note that prices can vary depending upon quantities ordered, vendors and time of order. (1) Packaged 10 MP CMOS sensors (Omnivision OV10823) were $30 apiece in quantities of 100. (2) Imaging lenses were $75 apiece in quantities of 100. (3) FPGAs were $267 apiece for 16 units. For the presented MCAM with 96 imagers, the total price of these key components is $14,352. Additional costs were incurred for opto-mechanics, PCB manufacturing, associated electronics, illumination, fluorescence excitation and emission filters for fluorescence imaging experiments, and a desktop computer to read acquired data.

## MCAM software

### Acquisition

All data acquisition was performed using custom-developed Python software. Various Python scripts for controlling image acquisition (e.g. exposure time, gain, number of micro-cameras for image capture, video acquisition parameters) were used for command-line execution and integrated into a custom PyQt graphical user interface. Python scripts were also developed to route and organize acquired image and video data within memory. Once set up and running, video or imaging data can be captured automatically for hours without human supervision. Additional data management details are in Appendix 1: MCAM Data Management.

### Image preprocessing

Before analyzing full-field image data, we first applied a flat-field correction to each micro-camera image to remove vignetting and other artifacts caused by variations in the pixel-to-pixel sensitivity of individual micro-cameras. This correction was performed by normalizing each pixel to the value of the mean of a set of five out-of-focus full-field images of a white diffuser captured at the end of each experiment. After this standard flat-field correction, we then performed image stitching on a per-frame basis as explained below. Additional details regarding inter- and intra-sensor brightness variations are provided in *Appendix 1—figure 2* and its associated discussion.

### Image stitching

For finalized, multi-camera images, we either used commercial stitching software PtGui (Rotterdam, The Netherlands), or a custom Fourier stitching algorithm to combine the images into a final stitched panorama. As computational stitching algorithms require many features across the image to work successfully, and many of the images of biological samples are relatively sparsely populated with features (e.g. *Figure 2A*), at the beginning or end of each imaging session, we captured images selected because they were feature-rich across the field of view, and contained a good deal of variation in the frequency and phase domains (e.g. pictures of natural scenes, text, or other complex features). We used these target images to generate stitching parameters for all images subsequently captured. See Appendix 1: Image Stitching for additional details.

## Depth detection

The MCAM system was configured to have slightly more than 50% FOV overlap in adjacent horizontal cameras to enable the system to image an object with at least two cameras at any given timepoint. Stereoscopic depth tracking was the achieved with custom-written Python software (see details in Appendix 1: Depth Tracking). Briefly, following object detection (outlined below), features were identified in stereo-image pairs of objects of interest from neighboring micro-camera pairs with standard OpenCV packages. Feature pairs were identified via the minimum Hamming distance between the per-image lists of candidate feature vectors. The average distance between feature pairs was then used as a measure of object disparity at the sensor plane, from which object depth is established via basic geometric relations.

## Zebrafish

For all zebrafish experiments, we used zebrafish 5–10 days post-fertilization (dpf). All zebrafish were either wild-type (TL) or transgenic nacre -/- (*mitfa*) zebrafish; these mutants lack pigment in melanocytes on the skin but not in the eye. Zebrafish were maintained on a 14 hr. light /10 hrs. dark cycle and fertilized eggs were collected and raised at 28.5 °C. Embryos were kept in E3 solution (5 mM NaCl, 0.17 mM KCl, 0.33 mM CaCl2, 0.33 mM MgSO4). All experiments were approved by Duke University School of Medicine's standing committee of the animal care and use program. Imaging experiments in this study were performed on transgenic zebrafish *Tg(isl1a:GFP)*(*Higashijima et al., 2000*), *TgBAC(slc17a6b:LOXP-DsRed-LOXP-GFP)*(*Koyama et al., 2011*), and *Tg(elavl3:GCaMP6s)*(*Chen et al., 2013*), generous gifts from Misha Ahrens and Ken Poss. In-vivo epi-fluorescence imaging in the MCAM system was performed in transgenic zebrafish *Tg(elavl3:GCaMP6s)* (*Chen et al., 2013*) 5–7 days post fertilization. Zebrafish were homozygous for GCaMP6s and nacre, *mitfa* (*Lister et al., 1999*) to prevent formation of dark melanocytes, effectively clearing the skin.

## Freely swimming zebrafish larva

Zebrafish were placed in a custom 3D-printed arena filled with E3 medium to a height of 5 mm. Arenas were designed using Fusion360 (Autodesk; CA, USA). For gigapixel imaging, the arena was rectangular with curved corners, with dimensions 11 cm x 22 cm, wall thickness 1.5 mm, and height 5 mm. The arena was printed with black PLA (Hatchbox, CA, USA) and glued onto a glass sheet using black hot glue around the outside of the arena. For data shown in *Figure 1*, we recorded spontaneous swimming behavior of 130 larvae over 1 hr. All data acquisition and preprocessing was performed using custom Python software.

## Object detection at gigapixel scale

To detect zebrafish in stitched images generated from stitched images, we trained a Faster-RCNN object detection network (*Appendix 1—figure 5A*) with *tensorflow*. For more details about the network, training, and parameters used, see Appendix 1: Large-scale object detection pipeline. Briefly, this deep neural network takes in an arbitrary image and generates bounding boxes (and confidence values) around all putative fish in the image (*Figure 2A*). We made several modifications to standard training and inference stages, including adding a novel occlusion step to the augmentation pipeline (*Ren et al., 2016*; *Appendix 1—figure 5D*). Because the images are much too large for training and inference using standard GPU RAM, we broke up the training images into patches that contained the objects of interest (as well as negative examples, such as debris) and used these sub-images for training (*Appendix 1—figure 5B*). For inference, we use a moving window followed by non-max suppression to remove redundant detection events (*Appendix 1—figure 5H*). A repository that contains instructions for downloading and using the fully trained *gigadetector* network can be found at GitHub at https://github.com/EricThomson/gigadetector, (copy archived at swh:1:rev:c94f-f09e4e6f73b803a529b165be68ad3bb0a029; *Thomson, 2021*).

## Zebrafish Identification

To demonstrate the resolution of MCAM, we again captured video of zebrafish larvae and trained a convolutional neural network (CNN) with images obtained from the video's frames to distinguish between detected fish. We devised the CNN in *tensorflow* as a siamese neural network with triplet loss (*Schroff et al., 2015*). The architecture consists of three sets of two convolutional layers with rectified linear unit (ReLU) activation functions and 1 max pool, and a final two fully connected layers. The last fully connected layer outputs 64-dimensional embeddings (*Appendix 1—figure 6B*). Inputs to the network were a set of 3 images: two images of the same fish taken from different frames ($f_i$ and $\widetilde{f}_i$), and one image of a different fish ($f_j$). The network was then optimized during training to minimize the Euclidean distance between embeddings of $f_i$ and $\widetilde{f}_i$ and maximize the Euclidean distance between embeddings of both $f_i$ and $f_j$, and $\widetilde{f}_i$ and $f_j$. We performed a temporal validation of the network, using the first 80% of frames of the MCAM recording to generate training images and the later 20% to generate testing images. After data augmentation we generated 250 images per larval zebrafish, with the 80/20 split resulting in 1800 training images and 450 testing images. We trained using the Adam optimizer (with default parameters) for 62 epochs with a batch size of 32. To speed up computations, we performed 2 x image down sampling to 224×224 pixels; at this resolution, feature

differences across fish, including variegated melanophore patterns, are still apparent. We used t-SNE (*Maaten and van der Hinton, 2008*) to visualize the 64-dimensional embeddings produced as output from the network (*Figure 2C*). For additional details see Appendix 1: Convolutional neural networks for zebrafish identification.

## *Caenorhabditis elegans*

*C. elegans* strains were kindly provided by Dr. Dong Yan, Duke University, by way of the *Caenorhabditis* Genetics Center (CGC) at the University of Minnesota. We used three different strains. One, a wild-type (N2) line. Two, a unc-3 mutant CB151 that is a loss-of-function mutant. Three, a novel transgenic line NYL2629 that is a cross between strains CZ2475 [Pflp-13::GFP(juIs145) II and OH11746] (*Donnelly et al., 2013*) and OH11746 Punc-3::mCherry +pha-1(otIs447) (*Kerk et al., 2017*) that express fluorescent proteins in two sets of motor neurons. To grow the colonies, we followed standard procedures (*Stiernagle, 2006*). Briefly, we prepared nematode growth medium (NGM) plates using medium acquired from Carolina Biological (NC, USA). We melted the solid NGM in a hot water bath at 95 °C, and before pouring, we cooled the bath to 55 °C to minimize warping of the petri dishes and condensation. We used a large petri dish (24.5 cm x24.5 cm Square Bio Assay Dishes; Thermofisher). To prevent contamination, we flamed the bottle lip before pouring. Once the NGM was cured (about an hour), we then grew a lawn of *E coli* in the medium by inoculating the NGM with OP50 and letting it grow overnight in an incubator at 37 °C. OP50 was kindly provided by Dr. Dong Yan at Duke University. To grow a colony of *C. elegans*, we transplanted chunks of pre-existing lines to the *E. coli* lawns and monitored the dish until we observed adult *C. elegans* moving through the lawn on the majority of the dish (typically 2–4 days). Code for *C. elegans* detection and statistical analysis, see https://gitlab.oit.duke.edu/ean26/gigapixelimaging.

## Physarum polycephalum

Desiccated slime mold (*P. polycephalum*: Carolina Biological; NC, USA) was maintained at room temperature until ready for use. To activate the sclerotium, we set it on a plate of agar (1–4%; Becton, Dickinson and Company; NJ, USA), and hydrated it with a drop of distilled water. We liberally sprinkled raw oatmeal flakes on the agar. Under these conditions, the slime mold enters the plasmodium stage, a moist amoeboid state that grows in search of food in its environment. Once the plasmodium has grown, we then used it to seed additional agar plates, either directly with an inoculating glass, or by 'chunking' a volume of agar and placing it on a second agar plate, as we did to transfer *C. elegans*.

We used 3D printing to create plastic mazes, which we designed using Fusion360 (Autodesk; CA, USA). The mazes were square (16.5 cm outer diameter), with wall thickness 3 mm, height 4 mm, and passage width of 12 mm. Mazes were printed with cool gray PLA (Hatchbox, CA, USA). Topographically, the maze was symmetric in that there were four starting positions, one in each corner, and for each starting position the slime mold would have to travel the same minimum distance (25 cm) to the final central position (*Figure 4A*). To construct the maze, we poured 2% agar into a large square (24.5 cm x 24.5 cm) petri dish (Nunc Bioassay Dishes; Thermofisher, MA USA), and then placed the maze into the agar while it was still in its liquid phase. We found embedding the maze in the agar better than placing it on top of cured agar, because the slime mold typically grew underneath any items placed on the agar. After setting up the maze, but before placing the slime mold inside, we disinfected the maze and oatmeal flakes with UV light for three minutes. It is important to keep the petri dish covered during long recordings, so the agar does not dry out. We prevented condensation on the cover by coating it with an anti-fog spray (Rain-X; TX, USA).

Spreading unscented aluminum antiperspirant gel (Dry Idea; Henkel, Düsseldorf, Germany) on top of the maze was an effective deterrent to 'cheating' behavior (crawling over the top of the maze). We found this by trial and error with many toppings that included nothing, Chapstick (menthol/strawberry/cherry) (ChapStick; NC, USA), Campho-Phenique (Foundation Consumer Healthcare; PA, USA), and scented antiperspirant (Degree; Unilever, London, U.K.). We observed that the latter two substances were so aversive that the slime mold tended to simply stop growing even within the main passages of the maze.

## Camponotus pennsylvanicus

Black carpenter ant (*Camponotus pennsylvanicus)* workers were acquired from Carolina Biological, NC. The ants were housed in colonies of ~50 individuals and fed with a mixture of leaves and wood provided by Carolina Biological. The colonies were kept dormant in a refrigerator at ~4 °C. Prior to experiments, the colony was removed from the refrigerator and kept at room temperature for at least 1 hr until ants exhibited baseline levels of movement. To observe the responses of groups of ants to different levels of light, approximately 20 ants were transferred from the colony into a rectangular 3d printed enclosure. The group was imaged for 2 min under visible light, which was turned off for 2 min. This cycle was repeated six times. The entire time, the ants were illuminated from above by IR light sources.

## Statistical analysis

All values reported as mean ± sd unless otherwise stated. In bar graphs, error bars correspond to standard deviation unless otherwise stated.

## Code availability statement

All code used for analysis was made using standard approaches in Python or MATLAB 2017b and open-source code extensions. All code and example data are available at dedicated GitLab repository https://gitlab.oit.duke.edu/ean26/gigapixelimaging. The object detection code is available at https://github.com/EricThomson/gigadetector, (copy archived at swh:1:rev:c94ff09e4e6f73b803a529b-165be68ad3bb0a029; *Thomson, 2021*). Instructions for running the code and reproducing published results is available via the code repository README files and via Jupyter Notebooks, which include code to produce figures.

## Acknowledgements

We thank the Duke School of Medicine as well as Ramona Optics Inc for technical support, as well as Maxim Nikitchenko for helpful comments and discussions. Thanks to the Ken Poss lab for the *Tg(islet1:GFP)* fish line, Misha Ahrens for the Tg(elavl3:GCaMP6s) and *TgBAC(slc17a6b:LOXP-DsRed-LOXP-GFP)* lines. We further thank R Yang for advice and gifts on *Drosophila* and Wesley Grueber and Rebecca Vaadia for advice and gifts of the transgenic *Drosophila* lines and Rebecca Yang for assistance with their husbandry, Jason Liu for help with experiment design and 3D printing, as well as Dr. Benjamin Judkewitz and Dr. Mike Orger for early project assistance. We furthermore thank Dong Yan, Albert Zhang, and the Center for *Caenorhabditis* Genetics Center (CGC) for the *C. elegans* lines and help with setting up *C. elegans*. J Burris and K Olivera for zebrafish husbandry.

Research reported in this publication was supported by the Office of Research Infrastructure Programs (ORIP), Office Of The Director, National Institutes Of Health of the National Institutes Of Health and the National Institute Of Environmental Health Sciences (NIEHS) of the National Institutes of Health under Award Number R44OD024879, by the National Cancer Institute (NCI) of the National Institutes of Health under Award Number R44CA250877, by the National Institute of Biomedical Imaging and Bioengineering (NIBIB) of the National Institutes of Health under Award Number R43EB030979, and by the National Science Foundation under Award Number 2036439. The content is solely the responsibility of the authors and does not necessarily represent the official views of the National Institutes of Health. EAN also was supported by the Alfred P Sloan Foundation. We thank Kristian Herrera for 3D renderings in the demo movie.

## Additional information

Competing interests

Mark Harfouche: is scientific co-founder at Ramona Optics Inc which is commercializing and patenting the multi-camera array microscope. Sunanda Sharma, Jaehee Park: was an employee at Ramona Optics Inc, which is commercializing and patenting the multi-camera array microscope. Roarke W Horstmeyer: is a scientific co-founder at Ramona Optics Inc, which is commercializing and patenting the multi-camera array microscope. The other authors declare that no competing interests exist.

## Funding

| Funder | Grant reference number | Author |
|---|---|---|
| Alfred P. Sloan Foundation | Sloan Foundation | Eva A Naumann |
| Office of Research Infrastructure Programs, National Institutes of Health | SBIR R44OD024879 | Eric E Thomson Mark Harfouche Jaehee Park Sunanda Sharma |
| National Cancer Institute | SBIR R44CA250877 | Mark Harfouche Sunanda Sharma Jaehee Park |
| National Science Foundation | NSF 2036439 | Mark Harfouche Jaehee Park Sunanda Sharma |
| National Institute of Biomedical Imaging and Bioengineering | SBIR R43EB030979-01 | Mark Harfouche Jaehee Park Sunanda Sharma |

The funders had no role in study design, data collection and interpretation, or the decision to submit the work for publication.

## Author contributions

Eric E Thomson, Conceptualization, Data curation, Software, Formal analysis, Validation, Investigation, Visualization, Methodology, Writing – original draft, Writing – review and editing; Mark Harfouche, Conceptualization, Software, Formal analysis, Funding acquisition, Validation, Investigation, Visualization, Methodology, Writing – original draft, Writing – review and editing; Kanghyun Kim, Pavan C Konda, Shiqi Xu, Data curation, Software, Formal analysis, Validation, Investigation, Visualization, Methodology, Writing – review and editing; Catherine W Seitz, Validation, Investigation, Methodology, Writing – review and editing; Colin Cooke, Software, Formal analysis, Methodology, Writing – review and editing; Whitney S Jacobs, Data curation, Software, Formal analysis, Investigation, Methodology, Writing – original draft, Writing – review and editing; Robin Blazing, Conceptualization, Formal analysis, Validation, Investigation, Writing – review and editing; Yang Chen, Formal analysis, Validation, Investigation, Writing – review and editing; Sunanda Sharma, Jaehee Park, Validation, Investigation, Visualization, Writing – review and editing; Timothy W Dunn, Software, Formal analysis, Methodology, Writing – original draft, Writing – review and editing; Roarke W Horstmeyer, Eva A Naumann, Conceptualization, Resources, Data curation, Software, Formal analysis, Supervision, Funding acquisition, Validation, Investigation, Visualization, Methodology, Writing – original draft, Project administration, Writing – review and editing

## Author ORCIDs

Eric E Thomson http://orcid.org/0000-0002-7118-2249
Mark Harfouche http://orcid.org/0000-0002-4657-4603
Kanghyun Kim http://orcid.org/0000-0002-4557-9525
Shiqi Xu http://orcid.org/0000-0002-8450-9001
Timothy W Dunn http://orcid.org/0000-0002-9381-4630
Roarke W Horstmeyer http://orcid.org/0000-0002-2480-9141
Eva A Naumann http://orcid.org/0000-0002-7215-4717

## Ethics

All experiments followed the US Public Health Service Policy on Humane Care and Use of Laboratory Animals, under the protocol A083-21-04 approved by the Institutional Animal Care and Use Committee (IACUC) of Duke University School of Medicine. All experiments on zebrafish were performed according to these standards and every effort was made to minimize suffering.

Decision letter and Author response
Decision letter https://doi.org/10.7554/eLife.74988.sa1
Author response https://doi.org/10.7554/eLife.74988.sa2

## Additional files

### Supplementary files
• MDAR checklist

### Data availability

All data generated or analyzed during this study are included in the manuscript and supporting files; Source Data files and associated analysis code have been provided on https://gitlab.oit.duke. edu/ean26/gigapixelimaging. To view associated MCAM videos with flexible zooming capabilities see https://gigazoom.rc.duke.edu/team/Gigapixel%20behavioral%20and%20neural%20activity% 20imaging%20with%20a%20novel%20multi-camera%20array%20microscope/Owl. Other MCAM source data can be viewed at https://gigazoom.rc.duke.edu/ Raw MCAM video data as well as other relevant manuscript data for all experiments is publicly available at https://doi.org/10.7924/r4nv9kp8v.

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

# Appendix 1

## MCAM hardware

### MCAM-24

As described in the main text, the multi-camera array microscope (*Appendix 1—figure 1A*) consists of an array of cameras, each composed of a lens, sensor, and optomechanical housing. Each camera images a unique area of a large sample plane in parallel with the other cameras. The base unit of an MCAM array consists of 24 cameras arranged in a 6x4 array (an MCAM-24: *Appendix 1—figure 1A*). The 24 CMOS image sensors (1.4 µm pixel, 10.5 Megapixels total, Omnivision OV10823) that comprise an MCAM-24 are integrated onto a single PCB which is controlled by custom-developed electronics (four follower FPGAs and one leader FPGA) to route data via USB 3.0 to a desktop computer. The sensors are arranged at a 19 mm pitch. Customized mounts fit over the sensor array and attach to a 24-lens adjustable mount, which holds a 6x4 array of individually focusable M12 lenses (*f*=25 mm, Edmund 58–207) at 19 mm pitch which are focused at a 150 mm object distance (*Appendix 1—figure 1B*). The resulting raw image data from this arrangement (*Appendix 1—figure 1C*) contains 240 Megapixels (MP), which is then sent to our image stitching and analysis software to create the final resulting images and video (*Appendix 1—figure 1D*).

### MCAM-96

To create the gigapixel multi-camera array system in *Appendix 1—figure 1E*, we combine four of the MCAM-24 systems into a 2x2 configuration, via the use of optomechanical mounting plates (*Appendix 1—figure 1F*). This results in an 8x12 array of lenses and sensors, with 4 sets of electronics to send image data via four USB 3.0 lines to a single desktop computer. The MCAM-24 arrays were designed to ensure that 19 mm inter-sensor spacing is maintained when 4 individual systems are joined together into an MCAM-96 arrangement. Similar image stitching and analysis software is then applied to each of the 960 MP raw captured frames to create the final video (*Appendix 1—figure 1G*).

Illumination: LED arrays were used to illuminate samples either from above (epi-illumination), from below (trans-illumination), or both. When using trans-illumination, one and four commercially available LED arrays (32x32 RGB LED Matrix, Adafruit) were used for the MCAM-24 and MCAM-96 setup, respectively. In some experiments, we placed a sheet of diffusive paper over the LED array to provide more even illumination. For epi-illumination, a custom-developed LED array was used that contained approximately 500 LEDs arranged on a PCB around a 4x6 grid of 19 mm holes within the board, for each micro-camera lens to image through.

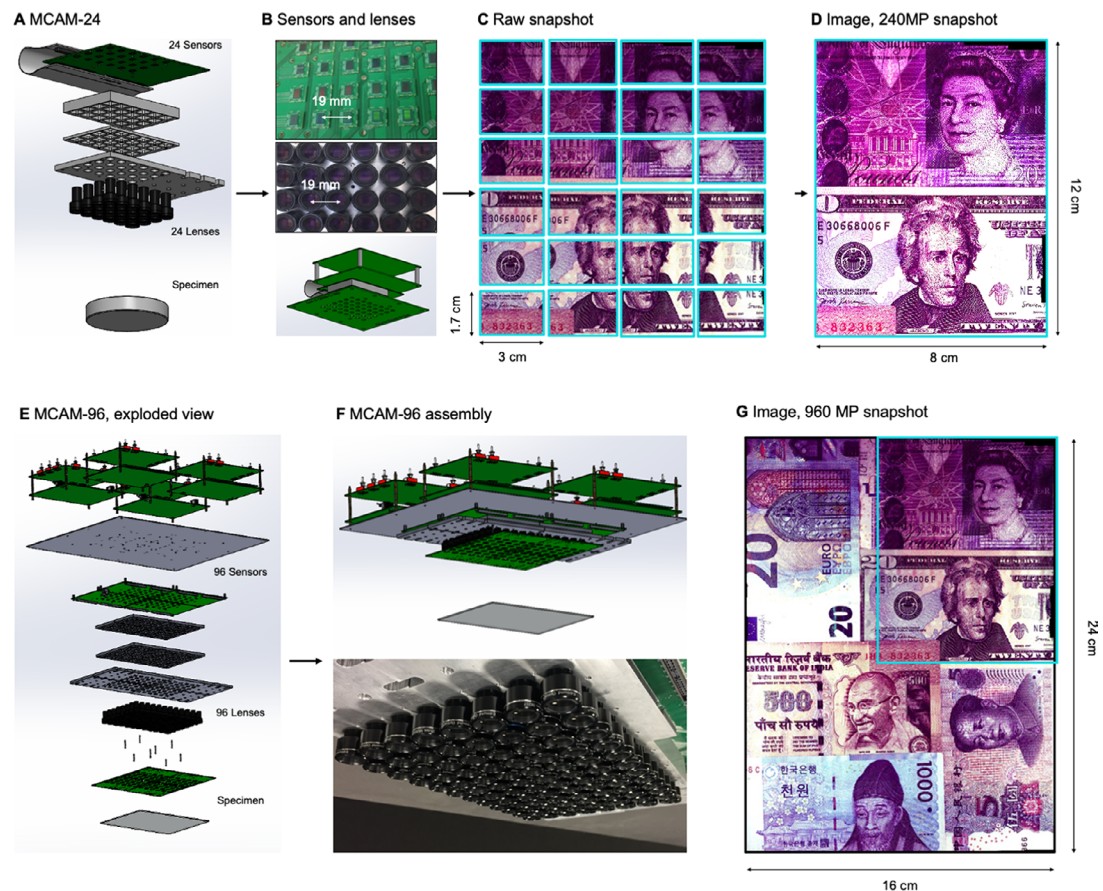

**Appendix 1—figure 1.** MCAM hardware. (**A**) MCAM-24 hardware arrangement in an exploded view, showing an array of 24 individual CMOS sensors (10 MP each) integrated onto a single circuit board with 24 lenses mounted in front. (**B**) Photos of bare MCAM-24 CMOS sensor array and lens array. (**C**) Example snapshot of 24 images acquired by MCAM-24 of common currency. (**D**) Stitched composite. (**E**) Four MCAM-24 units are combined to create the gigapixel MCAM-96 setup, with 96 sensors and lenses tiled into a uniformly spaced array for a total of 960 MP captured per image snapshot. (**F**) CAD render and photo of complete MCAM-96 system. (**G**) Example stitched composite from MCAM-96, with MCAM-24 field-of-view shown in teal box.

## MCAM resolution analysis and verification

The MCAM imaging system employed optics that were designed to provide an object-side numerical aperture (NA) of approximately 0.03 (9 mm optical lens diameter at an approximately 150 mm working distance). Under incoherent illumination (with wavelength $\lambda$ =500 nm), this NA leads to an incoherent transfer function cutoff spatial frequency of $f_c$ = 2NA/$\lambda$ =0.12 $\mu$m$^{-1}$ at the object plane (120 line pairs per mm). This corresponds to a two-point Sparrow resolution limit cutoff of $d_{min}$ = 0.47 $\lambda$ /NA = 7.8 $\mu$m under incoherent illumination, and $d_{min}$ = 0.73 $\lambda$ /NA = 12.2 $\mu$m under coherent illumination (*Ou et al., 2015*) at the object (i.e., sample) plane. The use of relatively small micro-camera lenses in our experimental system led to minimal aberrations across each camera field-of-view, and thus minimal aberrations across the entire MCAM field-of-view.

The lens focal length (*f*=25 mm) was selected to produce an imaging system magnification M=$d_i$/$d_o$ = 30 mm / 150 mm=0.2 to fulfill our tiled field-of-view specifications. With an imaging sensor pixel size of 1.4 $\mu$m, we were able to sample at nearly this maximum achievable optical resolution limit. In other words, the imaging system resolution (optical and pixel spacing) was designed to be at the limit of critical sampling for coherent imaging. With a 0.2 imaging system magnification, the pixel size projected into the sample plane is 7 $\mu$m for the MCAM, which is close to half the coherent imaging Sparrow limit noted above.

Our final demonstrated resolution was close to both the expected diffraction-limited and pixel-limited system resolution. As sketched in *Appendix 1—figure 2A, B*, we used the lens, sensor and spacing specifications outlined above to create an arrayed system to capture image data from across

a continuous field-of-view for subsequent post-processing. By using rectangular image sensors, we were able to create slightly more than 50% overlap along one dimension for depth tracking, as revealed by the MCAM's cross-sectional field-of-view geometry and discussed further in the main text.

*Figure 2C, D* show an MCAM full-field image of a custom-designed resolution target (Photo Sciences Inc, https://www.photo-sciences.com/) spanning its full 18x24 cm imaging area. Zoom-ins highlight small, custom-printed US Air Force targets that allow us to assess the impact of aberrations from randomly selected areas. Using this custom-designed target, we were able to experimentally verify a resolution cut-off that consistently lies between group 5 element 5 and 6, which exhibit full-pitch resolutions of 17.54 μm and 19.68 μm, respectively, leading to our claim of 18 μm resolution in the main text. This experimental resolution is slightly worse than the theoretically predicted full-pitch resolution and can be attributed to mild aberrations and the potential contribution of slight defocus that can vary as a function of field-of-view position. To provide additional data regarding lateral and axial resolution, we measured the 3D fluorescent point-spread function (PSF) response of a randomly selected micro-camera. Specifically, we outfitted the MCAM for wide-field fluorescence (see Appendix 1 - Fluorescence imaging with the MCAM) and acquired images of a monolayer of 6 μm fluorescent microspheres (Fluoresbrite YG Microspheres CATALOG NUMBER: 17156–2, Emission max.=486 nm) axially translated by 100 μm across 7.7 mm. The similarity of PSFs at the center and edge of the FOV show a minimal impact of lens aberrations.

We note that as a finite-conjugate optical system, the magnification of MCAM images do slightly change as a function of sample depth at a fixed object/image plane distance. In this work, we set the object and image plane distances to approximately $d_o$ = 150 mm and $d_i$ = 30 mm to produce an M=0.2 magnification at the plane of best focus. With a system depth-of-focus of approximately 2.5 mm, we can evaluate the magnification at $d_o'$=149.75 mm and $d_o''$=151.25 mm to find that the magnification can vary between M'=0.2017 and M''=0.1983 for an imaged object within that range. The approximate FOV of a single micro-camera within the array is 39 mm x 19.5 mm.

Given the multiple definitions of depth-of-field (DOF) within the imaging community, we here provide 3 unique characterizations of MCAM DOF. First, the Rayleigh criterion for axial resolution under incoherent imaging conditions (applicable to all behavioral and fluorescence imaging experiments reported here) is defined as $\lambda/2NA^2$, where $\lambda$ and NA is numerical aperture (see example derivation in Section 3.2 of *Latychevskaia, 2019*). Assuming $\lambda$ =500 nm and NA = 0.03 (see above) yields a Rayleigh criterion-defined DOF as $D_{Ray}$ = 0.28 mm. Second, we directly measured the specimen axial range in which MCAM resolution stays within a factor of 2 of its optimal value at the focal plane. Specifically, we determined the axial distance at which the full-pitch resolution deteriorated from 18 μm to 36 μm. This corresponds to a drop in resolution from between USAF target Group 5 Element 5 and 6 to Group 4 Element 5 and 6 in our resolution target. As shown in *Appendix 1—figure 2I*, resolution decreases to between Group 4 Element 5 and 6 at an axial distance of +/-dz = 1.27 mm, leading to a 2 X resolution drop-defined DOF as $D_{Res}$ = 2.54 mm. Finally, we obtained an image contrast-based measure of DOF, $D_{Con}$, by plotting the normalized image mean image gradient magnitude of USAF target images captured at 20 μm axial increments. The resulting curves in *Appendix 1—figure 2I* (left) exhibit a FWHM of 1.96 mm and 1.94 mm, respectively, leading us to contrast-based DOF value of $D_{Con}$ = 1.95 mm.

Future MCAM designs can easily achieve higher lateral imaging resolution. A tighter inter-camera spacing can lead to a larger per-camera image magnification, which in turn can increase their maximum spatial resolution. We selected an inter-camera spacing of 19 mm here and used imaging lenses that had an 18 mm maximum outer diameter. The use of alternative lenses can lead to tighter packing. A smaller sensor pixel size will be used in future system designs, along with a tighter micro-camera spacing, to achieve a higher imaging NA and resolution. In addition, we did not account for the effects of a Bayer filter over the sensors' CMOS pixels in the above analysis. We used image normalization techniques to remove the effects of the Bayer filter for our grayscale resolution tests. The inclusion of a Bayer filter over each CMOS pixel array further limits detector resolution when utilized to produce snapshot color images. Furthermore, the Bayer filters also reduce sensor sensitivity in general. Specifically, when used for fluorescence imaging, the Bayer filter pattern leads to two problematic effects. First, it leads to reduced resolution per fluorescence channel image (below the

18 μm pixel resolution for white light). Second, it leads to reduced sensitivity. For applications that could benefit from higher spatial resolution, future designs will use unfiltered monochrome pixels.

Finally, as noted in the main text, we employed a standard flat-field correction procedure to account for brightness variations both within and across the employed CMOS sensors before proceeding to image stitching. We hypothesize that brightness variations were primarily caused by our employed large-area bright-field illumination source, which was slightly non-uniform and not fully diffuse at the specimen plane, as well as lens vignetting and possible sensor vignetting effects. To explore the properties of these brightness variations in more detail, we experimentally imaged a large diffuser with the MCAM, under illumination from our standard trans-illumination source. We captured and averaged a series of 3 images while slightly moving the diffuser to unique locations to produce an average "white" image. The resulting mean of the raw pixel values (0–255) of this white image are plotted in *Appendix 1—figure 2F* (left). The inter-sensor standard deviation of the raw pixel values is similarly reported in *Appendix 1—figure 2F* (right). The spatial profile of per-sensor brightness variations is assessed in *Appendix 1—figure 2G*. From this data, we measured the standard deviation of the mean pixel value across all 96 image sensors to be 10, where the mean of all pixel values was 120. Similarly, via our measurements of intra-camera brightness variation, we found the average of the standard deviation of the raw pixel values of each sensor to be 6.

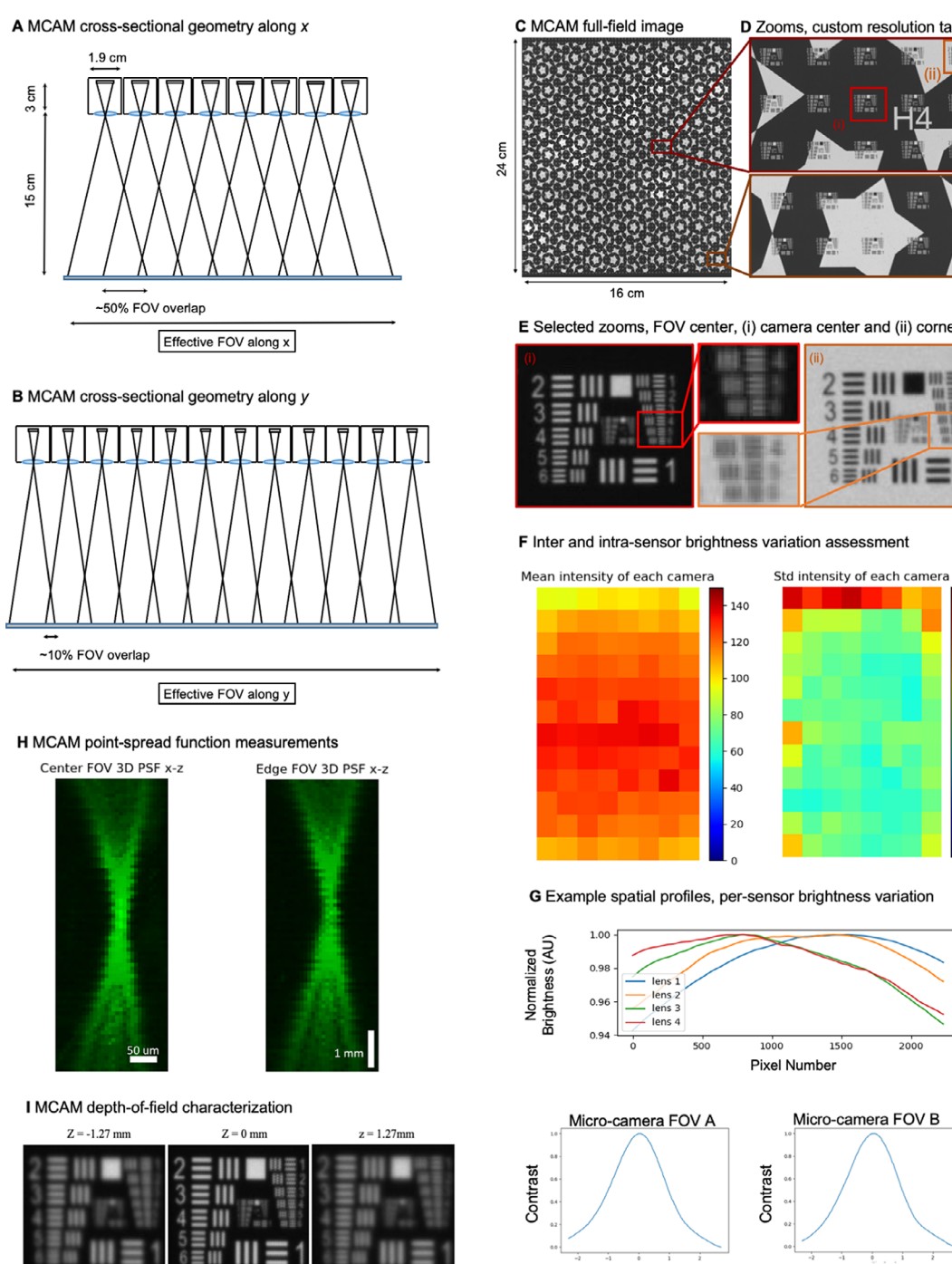

**Appendix 1—figure 2.** 96-camera MCAM geometry, resolution, and brightness measurements. (**A**) MCAM geometry cross-section along x direction of 8x12 array. The rectangular image sensor exhibits a longer dimension along x, yielding camera images sharing >50% FOV overlap for depth tracking and dual-fluorescence imaging. (**B**) MCAM geometry cross-section along y direction of 8x12 array, where image sensor's shorter dimension leads to approximately 10% overlap between adjacent camera FOVs to facilitate seamless image stitching. (**C**) Full field-of-view MCAM-96 image of a custom-designed resolution target covering over 16 cm x 24 cm, with (**D**) Zoom-ins. (**E**) Two further zoom-ins from marked boxes in (**d**) that lie at the center and edge of a single camera FOV. Resolution performance varies minimally across each camera FOV. For this custom-designed resolution target, Element 4 at the resolution limit exhibits a 22.10 μm full-pitch line pair spacing, Element 5 a 19.68 μm full-pitch line pair spacing, Element 6 a 17.54 μm full-pitch line pair spacing. The MCAM consistently resolves Element 5 but at times does not fully resolve Element 6, leading to our approximation of 18 μm resolution. (**F**) Plots of inter- and

*Appendix 1—figure 2 continued on next page*

*Appendix 1—figure 2 continued*

intra-camera brightness variation for a uniformly diffuse specimen. (left) Mean of raw pixel values for each camera within the 8x12 array, and standard deviation across all per-sensor raw pixel values for each sensor within the 8x12 array. (**G**) Plot of average normalized pixel value across the vertical camera dimension for 4 randomly selected micro-cameras when imaging a fully diffuse target. Relative brightness varies <6% across the array. Primary contributions arise from illumination, lens, and sensor-induced vignetting effects. As described in Appendix 1 – Image stitching below, digital flat-field correction is used for correction. (**H**) X-Z and Y-Z cross-sections of point-spread function (PSF) measurements acquired by a single micro-camera at FOV center and edge. For acquisition, monolayer of 6 µm fluorescent microspheres was mechanically translated in 100 µm steps axially across 7 mm. (**I**) Depth of field (DOF) characterization data. (left) USAF target images at axial locations used to identify axial range across which resolution stays within a factor of 2 of its maximum value as $D_{Res}$ = 2.54 mm. (right). Plots of normalized mean USAF target image gradient magnitude as a function of axial position across 5 mm depth range with average FWHM $D_{Con}$ = 1.95 mm.

## Image stitching

The MCAM relies on image stitching software to combine images from all micro-cameras into a final integrated composite image (a 'stitched' image). Stitching is not required for many applications (e.g., organism detection and cropping, organism tracking, fluorescence analysis), as application-specific software can be applied directly to the raw image data. However, for applications such as image segmentation, or to produce finalized, multi-camera images for visualization or presentations, stitching can be a helpful post-processing step.

The general workflow of our stitching process is outlined in *Appendix 1—figure 3A*. In this work, we relied on two unique stitching algorithms written in Python but applied the same workflow to each. In short, we took advantage of calibration targets to help establish accurate stitching parameters for our experimental data, which we then applied to produce our final stitched video frames.

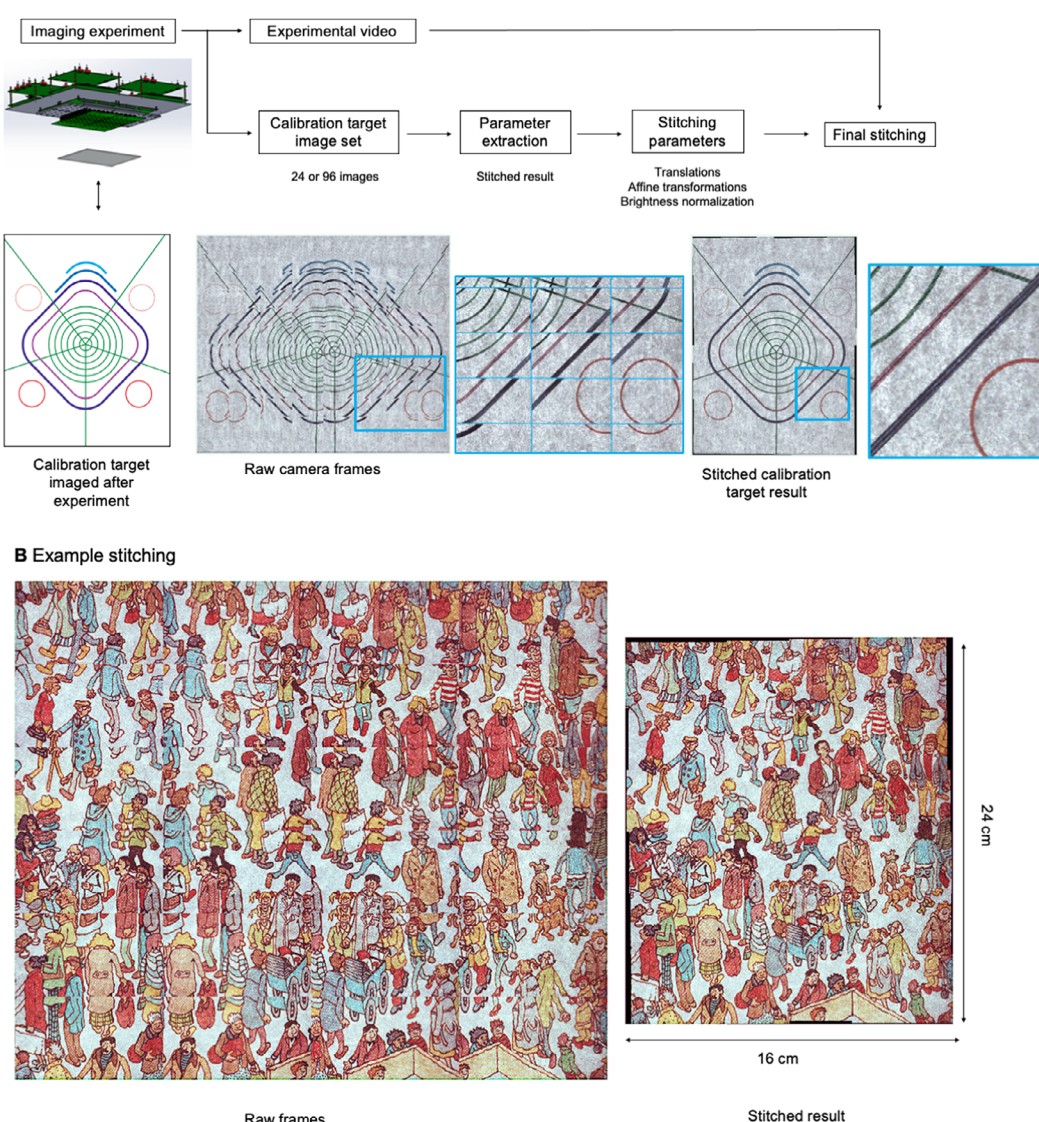

**Appendix 1—figure 3.** MCAM Stitching Process. (**A**) Schematic of workflow of the MCAM stitching process. (**B**) Example raw frame are stitched together using parameters extracted from calibration target.

1. Calibration target: Immediately before or after the image or video data collection step for our different sets of experiments, and before altering the position of the camera lenses or sample plane (i.e., before moving anything except the sample), we captured 2–4 images of a calibration target. The purpose of calibration target imaging is to acquire stitching parameters that may be otherwise challenging to extract from the raw experimental data, because the experimental data often did not contain a rich set of features (e.g., a few fish against a blank background contains mostly background). As detailed below, image stitching algorithms perform best with targets that include a large range of non-zero spatial frequencies across the sample plane. For instance, pieces of paper covered with text and wavy lines, photocopies of full-page photographs, or our custom-designed resolution target all worked well as calibration targets.

2. Parameter extraction: Before stitching each dataset, we processed the MCAM image set of the calibration target acquired in step 1 to extract key parameters. Details of the processing steps using each stitching approach are detailed below. Each approach generates a stitched calibration target image, as well as a set of parameters (e.g., a set of coordinates (x,y) to place each image, and/or any additional affine transform parameters such as per-image shearing and

rotation), which we saved for later use. We visually assessed the stitched calibration target images and, if any issues were identified (rarely encountered in later experiments), we would alter stitching algorithm hyper-parameters (discussed below) and re-run.

3. Final stitch: After extracting a set of parameters for the MCAM image set (24 or 96) for the calibration target, we used the same software to apply the saved stitching parameters to stitch the experimentally acquired MCAM image data; see example in *Appendix 1—figure 3B*. Video data was stitched on a per-frame basis.

We use two different stitching methods at different stages of our experimental development and testing.

### Fourier stitching algorithm

Fourier-domain image stitching is a well-established method to efficiently align different overlapping image segments (*Szeliski, 2007*). Briefly, shifts in the spatial domain correspond to a slope change of the phase component of an image's Fourier domain representation (i.e., its 2D spatial frequencies). Conveniently, by digitally computing the 2D fast Fourier transform of each image and determining the phase slope variation between them by computing their normalized product (i.e., phase correlation *Szeliski, 2007*), and then extract relative spatial image shifts by performing an inverse Fourier transform of the result.

### PtGui

We also adopted a commercial stitching software platform, PtGui (Rotterdam, Netherlands), to stitch MCAM images. This software comes with a simple batch processing mode, so that once you extract the stitching parameters for one calibration target, you can then apply the same parameters to all of the images in a directory, which we found it to be extremely flexible, accurate, and fast.

## MCAM Data Management

The outline of MCAM image data management is diagrammed in *Appendix 1—figure 4A*. MCAM-96 comprises four MCAM-24s, each of whose sensors are integrated on a common PCB board. Data from each set of 24 cameras is routed to a single USB 3.0 cable via a leader-follower FPGA configuration, yielding four USB 3.0 lines that connect the gigapixel system to a single desktop computer. During video capture, approximately 240 MB/sec of image data is sent along each USB 3.0 line for a 1 GB/sec approximate total data rate, which is subsequently saved in solid-state memory.

Due to limited data transmission speeds, our "full-frame" image capture and transmission strategy, delivering data from all of the cameras, utilized frame interleaving from consecutive cameras within each 24-unit array. The resulting timing sequence of this interleaved arrangement is sketched in *Appendix 1—figure 4B*. Each second (which defines the overall system frame rate), one image from each of the 24 cameras is exposed and then routed via the FPGAs to computer memory in a staggered manner. While the per-sensor exposure time can exceed 42ms (i.e., 1/24 of a second), the delay between captured snapshots from one camera to the next is set (approximately) at this quantity. This type of timing sequence is followed by each of the four 24-unit MCAMs comprising the full gigapixel system. Such asynchronous image capture is a limitation of the electronics layout of our first MCAM design. It can be addressed by several enhancements, including the potential addition of an external trigger, the output a precise timestamp for each frame, and/or a shift to an updated FPGA architecture that enables synchronous frame capture and collection.

In an alternative configuration, the MCAM system can also be operated in a "single-frame" image capture mode, where image data from any one of the 24 cameras can be streamed at approximately 10–15 frames per second, and the user can programmatically switch which camera they are streaming data from with a minor delay (100–200ms, depending upon exposure and frame rate), as sketched in *Appendix 1—figure 4C*. Within the gigapixel MCAM, it is possible to simultaneously stream from up to 4 cameras at this higher frame rate. By providing a 10 X higher imaging speed, this second streaming option is helpful for monitoring organism dynamics at a higher frame rat and troubleshooting during system setup and alignment.

**A** Sensor data diagram

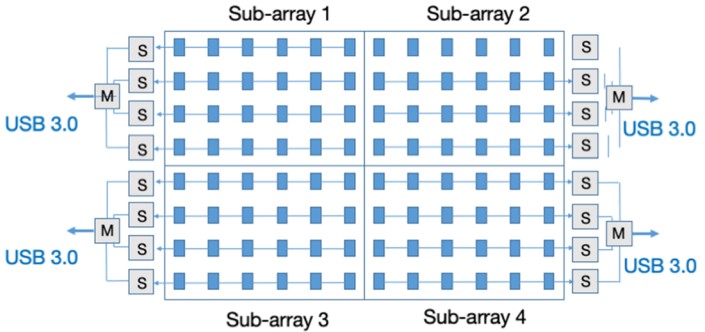

**B** Full-frame timing diagram

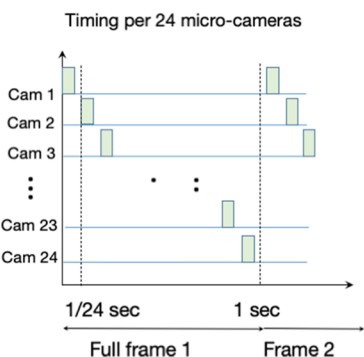

**C** Single-frame timing diagram

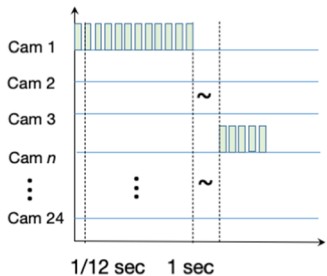

**Appendix 1—figure 4.** MCAM data management and timing. (**A**) MCAM-96 (0.96 gigapixel) data layout and readout geometry from four MCAM-24 sub-arrays. (**B**) Timing sequence of frame data for each MCAM-24 running in full-frame mode. (**C**) Timing sequence of frame data for each MCAM-24 running in single-frame mode.

## Large-scale object detection pipeline

We adapted a well-known deep-learning based object detection algorithm (Faster-RCNN) to work at gigapixel scale. We used this architecture because of its high accuracy (*Huang et al., 2017*). The Faster-RCNN architecture has two main components: a traditional Convolutional Neural Network (CNN) that extracts features from the original image, and a region proposal network (RPN) that is trained to find regions in the image that are likely to contain objects. Once the RPN suggests good candidate regions to look for objects, then a traditional classifier can be applied to features in those regions (*Appendix 1—figure 5A*). We used a Faster-RCNN with a resnet-101 backbone CNN that was pre-trained on the COCO dataset for feature extraction (*Lin et al., 2014*), and then subsequently fine-tuned using our zebrafish data, as described below. In what follows, we provide details on the four main steps required to train and run the network. While we discuss the pipeline in the context of bright field zebrafish data, the framework is applicable to any type of data collected using the MCAM.

### 1. Annotation of images

Training a network for object detection requires a set of ground-truth images that contains objects with bounding boxes or segmentation masks circumscribing their perimeter. To generate such annotated data, we selected a random sample of frames from our library of movies of freely moving zebrafish. Before annotation, we preprocessed the images by normalizing and stitching together images from individual cameras as discussed previously (*Figure 1*; Appendix 1: Image Stitching; *Appendix 1—figure 3*).

After stitching, the images have dimensions as high as 15000×23,000 pixels (width x height), which is far too large to fit in GPU memory during training. To reduce peak memory usage for training and inference, we broke up the images into smaller sub-images. Specifically, we selected all

1024×1,024 sub-images that contained at least one fish as well as *negative* examples such as debris, arena walls, etc. (*Appendix 1—figure 5B*).

We annotated these sub-images using the open-source Python package *labelImg* (*Lin, 2018*). Annotation consisted of drawing bounding boxes around, and supplying the category for, every object of interest. Because each sub-image is relatively small, this process went much faster than the equivalent process of drawing bounding boxes in full-sized MCAM images: an experienced annotator typically annotated more than 100 sub-images an hour. For the negative examples no bounding boxes need to be drawn: the images simply need to be verified as valid images in the training data.

The goal with annotation was to maximize precision (i.e., minimize false positives), and draw boxes as we hoped the algorithm would learn to discover them in the inference stage. When trying out different annotation strategies, we found that the decisions we made during annotation were directly reflected in the performance of the network. Over time, we adopted the following conventions: for purposes of annotation, we defined a "fish" as any image of a fish that included the head (which by definition had to include at least 1/3 of an eye down to 3/2 of the distance from the tip of the nose to the caudal tip of the swim bladder). Headless tail pieces ("sushi") were not annotated.

We needed to adopt such unusual conventions because of stitching artifacts, which sometimes made annotation challenging. Approximately 7% of training images that contained fish (65/940) included stitching artifacts. These artifacts included duplication of body parts (12/65), translocation of body parts (26/65), and complete detachment of segments of zebrafish, so they simply appeared as isolated body parts (27/65) (*Appendix 1—figure 5C*). For translocation artifacts, if the tail piece was detached from the head piece by fewer than 20 pixels, then we annotated all these parts as belonging to the same fish. If a fish part was duplicated due to asynchronous image acquisition between cameras, we annotated all those individual parts that satisfied the above definition of a fish.

Following the above conventions, we ended up with 1138 bounding boxes containing fish in 1097 of the 1024x1024 annotated sub-images. Of those sub-images, 157 were pure negative examples and 940 contained fish (763 sub-images contained one fish, 157 contained two, 19 contained three, and one sub-image contained four fish).

## 2. Augmentation of training data

We split the 1097 sub-images into training and testing data (an 80/20 split of 877 training images and 220 testing images). We did not train the network directly on the training images but used an image augmentation pipeline to generate five augmented images per training image for actual training (a total of 4385 augmented images). We augmented the training data using the open-source Python package *imgaug* (*Jung et al., 2020*). This package lets you arbitrarily transform each image in the training set, while also performing appropriate corresponding transforms of the bounding boxes so that your annotations will remain intact (for instance if you scale the original image during augmentation, the bounding box will be scaled appropriately). The annotation pipeline consisted of the following operations applied in the following order (with probability that it will be applied given in parentheses): left/right flip (0.5), up/down flip (0.5), rotate by +/-90 degrees (0.5), change brightness between [−30,80] (1.0), scale from [0.75, 1.5] (0.9), sharpen, blur, or neither (1/3 each), motion blur (0.15), rotate (from +/-5 degrees) (0.9), and a custom-built occlusion step (1/3). For examples, see *Appendix 1—figure 6D*. When a value, such as brightness, is given as a range, it was randomly selected from that range with a uniform probability.

Occlusion events were relatively rare in our actual data (7% of bounding boxes overlapped with other boxes), and the overlap was typically small, with a mean Intersection over Union (IoU) between bounding boxes of 10.9 (*Appendix 1—figure 5E*). Hence to ensure the network was able to detect fish in those rare cases when there was overlap, we built a custom occlusion augmentation step where we added an additional synthetically generated occluder fish to the training image (similar to the process in *Sárándi et al., 2018*). Briefly, we artificially generated occluder fish at a random location within the bounding box of the most central fish in the original sub-image. More specifically, we segmented ten fish images by hand in Photoshop, and saved each of them at eight 45 degree angles in an occluder library. The occluder fish image was augmented before being placed on the background, but with its brightness (i.e., mean pixel value in the center of the fish) set to match that of the background fish to give a more realistic appearance. This custom occlusion pipeline was written in Python using OpenCV. You can see two examples of occluder fish added in *Appendix 1—*

*figure 5D* (the second and third augmented figures contain occluder fish whose bounding boxes overlap with the original).

### 3. Training the network

Once we created the set of test images and (augmented) training data, we then encoded the data for training using the Tensorflow Object Detection API (*Huang et al., 2017*). Our first step was to encode the annotated data into Tensorflow record files, which contain the image and bounding box data in a byte-encoded format that is used by the object detection API to speed up processing.

Once the files are encoded, they are fed to Tensorflow's object detection API to train the Faster-RNN mentioned above. We trained the Faster-RCNN 75 k steps where we set the learning rate to decay during training: it was 0.003 for the first 10 k steps, 0.0003 up to step 15 k, and then dropped to 0.00003 after step 25 k. The network converged relatively quickly (typically within 30 k steps: *Appendix 1—figure 5F*) but we found in practice that letting it train longer gave better performance on rare events such occlusions, and also the continued decay in the loss function on the test data indicated that we were not simply overfitting the network (*Appendix 1—figure 5F*).

### 4. Inference with the frozen network

Once the RCNN network had undergone transfer learning, the network was frozen and used for inference on new data. Even for this step, the size of the stitched MCAM images was too large for running inference in GPU RAM. Hence, we sliced the large images into 1024x1024 sub-images using a sliding window beginning at the top left corner of the image (with a stride of 512 pixels) (*Appendix 1—figure 5G*). We found it important to ensure that each image patch was exactly 1024x1024, as used in training. So, for instance when the sliding window would have surpassed the right or bottom edge of the image, we shifted the coordinates back to select a full 1024x1024 sub-image rather than a cropped edge. Without this adjustment, we found the recall of the network suffered considerably. Note that within this modified faster-RCNN architecture, each sub-image window is processed independently, so this algorithm could easily be parallelized across multiple GPUs for significant speed gains.

We applied the frozen network to each sub-image and retained those bounding boxes to which the network assigned a confidence above 0.95. This procedure typically yielded multiple highly overlapping bounding boxes around the same organism from multiple sub-images (false positives were effectively non-existent, as we trained the network to be very selective, as discussed above). To filter out repeats, we first applied non-max suppression (NMS) to select the highest confidence bounding box from an overlapping cluster (NMS threshold 0.5, confidence threshold 0.95). However, even after this step, there were sometimes small boxes (below the NMS threshold) that remained as a subset within the larger boxes, so we applied a final subset filter that removed boxes that were contained within other boxes. These steps yielded the final estimate of the object detection pipeline: a set of bounding boxes, with their associated confidence estimates, like those seen in the original paper (*Figure 2*, *Appendix 1—figure 5H*)

The entire inference pipeline, including the frozen network and some example images and scripts to help people get started using the network, are available online at the gigadetector repository at GitHub: https://github.com/EricThomson/gigadetector, (copy archived at swh:1:rev:c94ff09e4e6f73b803a529b165be68ad3bb0a029; *Thomson, 2021*).

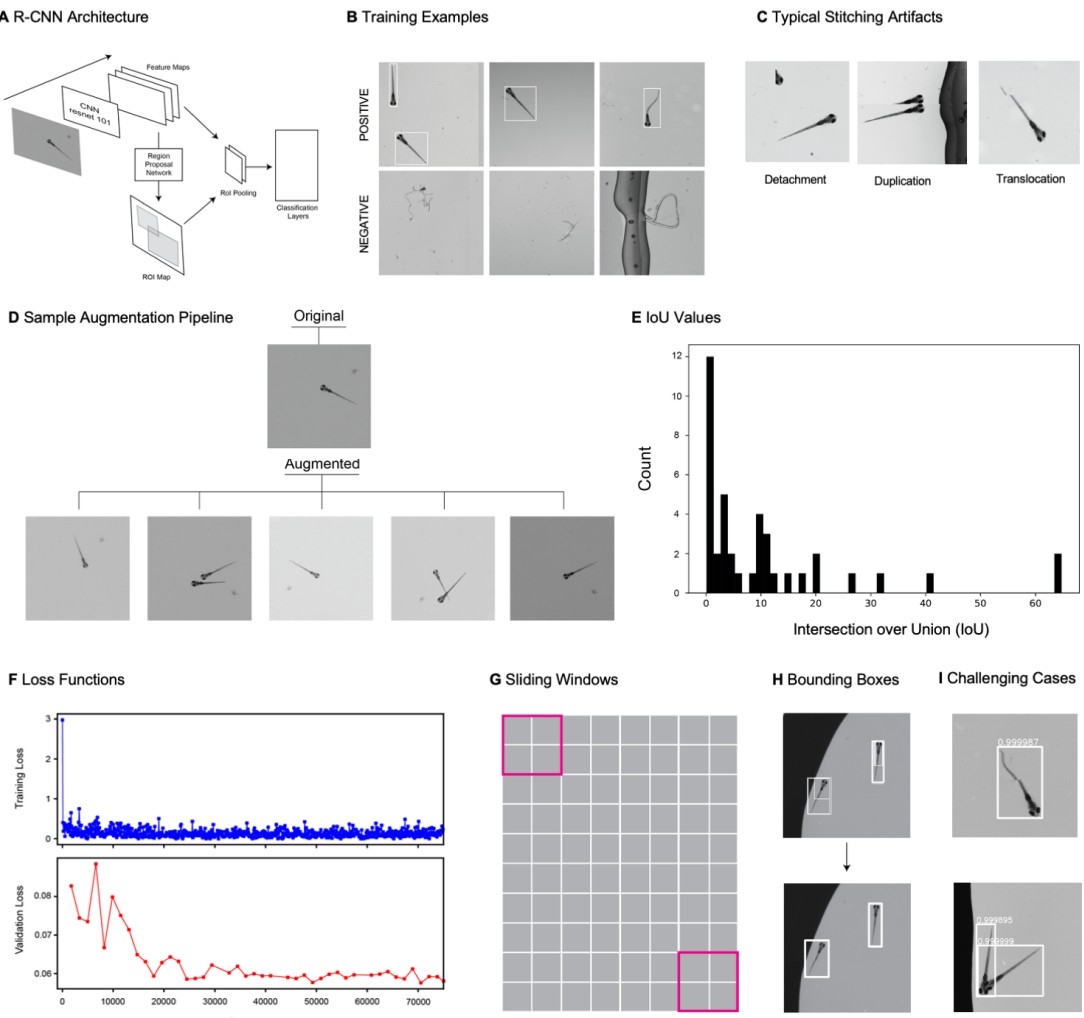

**Appendix 1—figure 5.** Object detection using gigapixel video. (**A**) Faster R-CNN architecture. The architecture consists of a traditional CNN (in this case a resnet-101 backbone) for feature extraction, which feeds into a region proposal network (RPN) and an RoI pooling layer. The RPN generates an ROI map, proposing potential locations of objects. The features and potential object locations are fed to an RoI pooling layer which normalizes the size of each bounding box before the information is fed into a classifier that predicts the object type for each ROI. (**B**) Examples of positive (top row) and negative (bottom row) examples used for training the faster R_CNN. (**C**) Examples of the main types of stitching artifacts, including detachment (left), duplication (middle), and translocation (right). See main text in Appendix 1: Large-scale object detection pipeline for more details. (**D**) Augmentation pipeline example: top image is the original image, and the bottom five images are five augmented instances generated from the original. We generated five augmented instances of each training image in our training data. (**E**) Histogram of intersection over union values for all fish from our actual data (test data and unaugmented training data). (**F**) Loss function in training and validation data during 75000 step training of the faster-R-CNN. (**G**) Depiction of sliding windows used for inference over stitched images. (**H**) Example from a fraction of a stitched frame showing multiple fish of the original set of multiple overlapping bounding boxes, and the final set of unique bounding boxes enveloping individual fish (obtained using a combination of a nonmax suppression followed by a proper subset filter, as described in Appendix 1). (**I**) Examples of successfully detected zebrafish in the presence of stitching artifacts (top) and occlusion (bottom). Please see additional example in *Appendix 1—Video 3*.

## Siamese networks for zebrafish identification

We designed a siamese neural network to differentiate among larval zebrafish. Prior deep learning algorithms have distinguished zebrafish at the juvenile and adult stage (*Romero-Ferrero et al., 2019*), yet these algorithms have not been shown effective on larval zebrafish, which do not differ significantly in size, and canonical striped skin coloration only emerges in the later, juvenile, stage (*Singh et al.,*

*2014*). However, when imaged using the relatively high-resolution MCAM, wildtype zebrafish larvae do exhibit multiple unique differences in features such as brightness, size, and arrangements of their melanophores, the dark-pigmented cells spread across the body (*Appendix 1—figure 6*, *Figure 2B, C*). We aimed to determine whether we could use deep learning to utilize these differences to distinguish individual larval zebrafish, in images obtained from the MCAM.

We trained our Faster-RCNN object detection algorithm on MCAM images of an arena containing nine zebrafish and used the resulting bounding box coordinates to crop out each individual animal for subsequent analysis. We then augmented the cropped images via rotations of random angle ($\theta \leq 360°$) (*Appendix 1—figure 6A*) and generated 250 images per zebrafish or 2250 images total. We used an 80/20 split for training and testing data to obtain 1800 training images and 450 testing images. The 80/20 data split was performed temporally so that training and testing frames came from non-overlapping periods. We halved the resolution of images to improve training speed and found the melanophore patterns were still discernable.

The architecture of our CNN consists of six convolutional layers and two fully connected layers (*Appendix 1—figure 6B*). We designed the CNN as a siamese neural network with a triplet loss, based on successful deep learning face recognition algorithms (*Taigman et al., 2014*; *Schroff et al., 2015*). The network receives multiple images and learns their similarities, as reflected by the Euclidean distance between images in a lower-dimensional embedding space within the network. Two images of the same zebrafish (anchor and positive) and one image of a different zebrafish (negative) are fed into three subnetworks with identical parameters and weights. Over training, the distance between the anchor and positive input decreases while the distance between the anchor and negative input increases, resulting in clusters containing images of the same fish.

We used a batch size of 32 and Adam optimization (*Kingma and Ba, 2015*). We generated 64-dimensional embeddings of the images and used t-distributed stochastic neighbor embedding (t-SNE) to visualize the embeddings in two-dimensions (*Maaten and van der Hinton, 2008*). We found that by 62 epochs the embeddings separated in two-dimensional space into nine clusters, each cluster corresponding to an individual fish (*Appendix 1—figure 6C*, *Figure 2B, C*). This demonstrates our CNN was effective at distinguishing among the nine larval zebrafish, presumably due to the MCAM resolving individual differences in their anatomical details such as their melanophore patterns (*Figure 2B*).

**Appendix 1—figure 6.** Siamese network with triplet loss differentiates individual zebrafish. (**A**) Strategy for training data generation from two example larval zebrafish in a section of the behavioral arena. Bounding boxes from Faster-RCNN algorithm were used to crop fish from images captured by MCAM. Data was augmented via rotations of cropped images by a random angle (θ≤360°). Individual differences, such as melanophore patterns are discernable in training images and presumably used to distinguish larval zebrafish. (**B**) Cartoon of Siamese Network with triplet loss architecture used to differentiate larval zebrafish. Three subnetworks with shared weights and parameters have six convolutional layers and two fully connected layers. The subnetworks receive triplets containing two training images of the same fish (anchor and positive) and a third training image of a different fish (negative) and generate 64-dimensional embeddings. Triplet loss compares the anchor to the positive and negative inputs, and during training maximizes the Euclidean distance between the anchor and negative inputs and minimizes the distance between the anchor and positive inputs. (**C**) Validation loss versus epoch for training shown in *Figure 2C*. Loss decreases and ultimately converges over training. Here, the network is trained for 62 epochs, which takes approximately 20 minutes.

## Segmentation and counting of *C. elegans*

To segment and quantify the number of *C. elegans* and their swarming behavior, we utilized a U-shaped fully convolutional neural network (*Ronneberger et al., 2015*), as also known as a *U-net*, to create binary identification masks of both individual worms and swarms of worms. To supervise the training, we created a manually labeled dataset composed of 655 manually labeled, 256x256 sized images, which was then augmented to 13100 labeled images using rotation. We choose a U-Net architecture (*Buda et al., 2019*) that is publicly available from the PyTorch library. To make the network output per-organism and per-swarm masks, we set the output channel numbers to two and initialized the network via a uniform random distribution of weights bounded by the square root

of input features of each convolution layer. During training, we input image segments containing either individual *C. elegans*, swarms, or both, with the output being the associated segmentation masks. The network was trained with the Adam optimizer using a batch size of 12, a learning rate of $1 \times 10^{-4}$ for 200 epochs. Once trained, we input the network with 512 x 512 sized tiled images from our MCAM-96 video sequences, cropped from the stitched gigapixel images with a 25% overlap (i.e., a 384-pixel step size). After the network predicted the segmentation masks for each image, we stitched all the segmentation masks together to form the segmentation results for the original gigapixel images. Once segmented, we used a straightforward pixel-connectivity-based method to count the number of objects (worms or swarms) in each gigapixel segmentation mask (*Haralick and Shapiro, 1992*).

## Depth tracking

The multi-camera array microscope (MCAM) imaging system uses multiple high-resolution cameras to simultaneously image across a large field of view (FOV). Although these cameras could be configured such that there is no spatial overlap between them, or a limited amount of spatial overlap that just enables effective image stitching, the system presented in this work utilized a larger degree of overlap to enable depth detection (see *Appendix 1—figure 2A–B*). Specifically, the cameras used rectangular sensors (2432x4,320 pixels; 5:9 form factor) and were configured such that the FOV of horizontally adjacent cameras overlapped by slightly more than 50% and the FOV of vertically adjacent cameras overlapped by approximately 10% to enable accurate stitching along the vertical direction. Therefore, any object within the collective FOV of the MCAM was captured by at least 2 cameras, with the exception of the cameras along the vertical edges at the end and start of horizontal rows (see *Appendix 1—figure 7B*).

The fact that adjacent cameras have overlapping fields of view means that, for a large central subset of each frame, there exists stereoscopic image data. Stereoscopic imaging is a robust method of estimating depth information from two 2D images (*Birchfield and Tomasi, 1998*). In essence, stereo-depth estimation uses the spatial disparity when an an object appears in two images to estimate an object's distance along the optical axis at one or more points, much in the same way that the relative lateral disparity between objects seen by our own two eyes can be used as an indication of object depth.

## Depth-from-stereo methodology

The key to establishing a depth estimate from a stereo pair of images is mapping pixel disparity of a particular point on the sample of interest to a physical distance. *Appendix 1—figure 7D* shows the geometry of a camera pair from the setup used in this work. The goal is to move from the total pixel disparity on the captured images ($d = d_1 + d_2$) to a physical object depth (Z). Under the assumption that our lenses and sensors obey certain geometric constraints (*Vázquez-Arellano et al., 2016*), we can use a simple similar triangles relationship (*Equation 1*) to establish the mapping between pixels and physical geometry:

$$\frac{d}{L} = \frac{B}{Z} \quad \rightarrow \quad Z = BL/d \tag{1}$$

This relationship is sketched in *Appendix 1—figure 7D*. Here, B is the inter-micro-camera spacing (19 mm) and L the lens-to-sensor (i.e., image) distance (~29 mm), which is computed via the lens-maker equation for *f*=25 mm lenses assuming a 150 mm working distance. We assume both parameters are fixed, as they are based on the mechanical construction of the MCAM system and remain constant across all camera pairs. Relevant parameters are listed in *Appendix 1—figure 7E*. The final pixel disparity parameter of interest, d, which is the sum of the feature offset distances from the centers of each image pair measured along each image plane, is a function of object depth and dynamically changes based on the relative positioning of a point of interest across each pair of adjacent cameras. To estimate object depth (Z) from our stereo system, we thus need a way of estimating the pixel disparity d on a per-object basis.

## Depth-from-stereo computational pipeline

The first step to depth estimation is thus to select corresponding regions of interest (ROI) from images of adjacent camera pairs, which allows us to isolate a single object (e.g., the tail of a particular zebrafish). Our process of ROI selection utilizes an object detection algorithm, as detailed in the description of the Large-scale object detection pipeline (*Appendix 1—figure 5*). However, any method can be applied to isolate specimen areas of interest. After selecting an object or specimen area of interest, we then apply a feature detection pipeline to estimate the average pixel disparity between the two perspectives of the object of interest. This process is illustrated in *Appendix 1—figure 7F*.

There are two critical components of depth estimation from feature detection: (1) feature creation (keypoint detection and description), and (2) feature matching. When combined, these two components allow us to find pairs of keypoints within stereoscopic images that are likely to represent the same physical location on the object. To detect and describe keypoints, we utilized the OpenCV implementation of ORB (Oriented FAST and Rotated BRIEF) (*Rublee et al., 2011*). The ORB process first performs keypoint detection on each sub-image, attempting to find areas which contain high amounts of unique information, which is performed using a modified version of the FAST (Features from Accelerated Segment Test) algorithm (*Rosten and Drummond, 2006*). Once keypoints are identified, the BRIEF (*Calonder et al., 2010*) (Binary Robust Independent Elementary Features) algorithm is used to form a feature descriptor. In this case, the descriptor is a binary feature vector that summarizes the spatial contents of a fixed-sized patch of pixels centered at the keypoint location.

After generating a list of candidate features and corresponding feature vectors for each of the two images comprising the stereo-image pair, we then match keypoints between images by minimizing the Hamming distance of their feature descriptors vectors (with all combinations of features tested via brute force). We apply a maximum Hamming distance threshold to the matching process to exclude spurious or poor matches from being included. To further increase the quality of keypoint correspondence, we applied a geometrically constrained RANSAC (*Fischler and Bolles, 1981*) to the matching keypoint pairs. This last step ensured that the final set of matched keypoints conforms to a Euclidean transform, which is a constraint driven by the mechanical properties of our imaging system. Finally, we computed the average distance between matched keypoints (in pixels), and multiplied this by the pixel size ($P$=1.4 μm) to obtain an estimate of the object's pixel disparity, d. As an average, this statistic exhibited sub-pixel-level resolution. Using *Equation 1*, combined with the known physical parameters of the MCAM (b and L) we transform our digital measurement of d into a physical estimate of the depth Z.

## Depth-from-stereo experimental verification

Once we established our depth detection pipeline, we constructed an experiment to validate the MCAM's depth detection accuracy and range. For this experiment, a standard 1951 USAF resolution test chart was placed on a motorized stage within the imaging area of the microscope. The stage used had a calibrated accuracy of 2 μm in the direction of travel (vertical) across all cameras and extending well beyond (5 mm+) the expected lens depth of field (approx. 0.5 mm).

For validation, the stage was moved between 50 positions, each 100 μm apart, across a total depth range of 5 mm. At each position, a set of images was captured (one from each of the two cameras used in this validation procedure). Since the resolution target (shown in *Appendix 1—figure 7G*) includes few but accurate features, the standard ORB key-point detection and feature description process was replaced with a simple centroid detection step. The remaining steps in the depth detection process remain unchanged. By removing the need for feature detection, we isolate the validity of tracking depth with the MCAM imaging system from the ability of algorithms to detect features accurately and robustly, which can vary from sample to sample.

After applying centroid detection on 3 bars within the resolution target, the relative position of each centroid is compared across adjacent cameras to estimate depth, following *Equation 1* and using the parameters found in Fig. *Appendix 1—figure 7E*. The results of this experiment are shown in *Appendix 1—figure 7H*, which show a clear match between predicted and measured depth across the entire 5 mm region, which extends well beyond the depth of field of the MCAM (approximately 0.5 mm).

The axial resolution of MCAM depth measurement can be directly related to the size of the image sensor pixel. As noted above, if multiple features are available per object area of interest, then statistical averaging can yield "sub-pixel" resolvability. However, assuming just one feature is used, we can estimate this resolution using *Equation 1*. Due to the inverse relationship between sensor disparity d and Z, the effect of a single pixel difference changes depending on total disparity. To obtain an approximate value of depth precision, we use the effect of a single pixel change when the sample is at a depth of 150 (the working distance of the MCAM system). Thus, plugging in the image distance L=28.9 mm, the baseline distance b=19 mm and $Z_a$ = 15 cm into *Equation 1* we obtain $d_a$ = 3.66 mm. We can then use this quantity to find the effects of a 1 pixel shift on Z by computing $Z_b$ = bL/($d_a$ +p)=149.943 mm, where $P$=1.4 µm is the sensor pixel size. The result suggests that that for every pixel of displacement measured on the sensor, the object of interest is axially displaced by approximately $Z_a$ – $Z_b$ = 57 µm in the sample plane, which can be used as an approximate measure of MCAM depth resolution via stereo-matching.

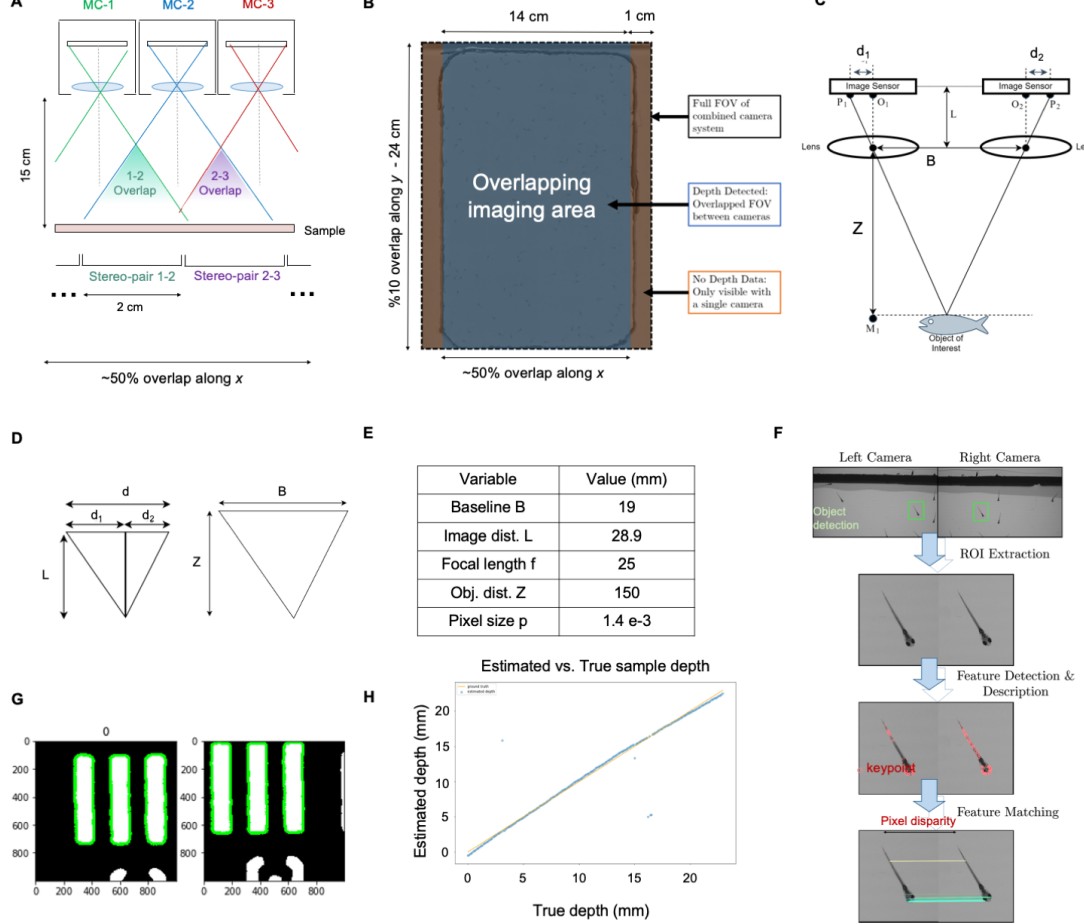

**Appendix 1—figure 7.** MCAM Stereoscopic imaging. (**A**) Diagram of MCAM imaging geometry for stereoscopic depth tracking with marked stereo-pair regions (**B**) Stereo-pair regions exist within an overlapping imaging area (other than a small boarder region). (**C**) Schematic of stereoscopic imaging arrangement with variables of interest marked. Object distance (**Z**) is calculated via trigonometry, after digital estimation of $d_1$ and $d_2$. (**D**) (**d**) Similar triangles from (**c**) used for depth estimation via *Equation 1*. (**E**) Table of variables and typical experimental values. (**F**) Feature Detection and Matching Pipeline. Matching ROI extracted from a stereo pair, a feature description algorithm is applied to each image and are then matched to compute $d_1$ and $d_2$ for pixel disparity measurement (**G, H**) Experimental results, indicating approximately 100 µm resolution over the entire depth range. Outlying points are due to the inability to accurately match image pair features. For more see main text of Appendix 1: Depth tracking section.

## Fluorescence imaging with the MCAM

The high-resolution and large field of view (FOV) of the MCAM makes its a powerful tool for recording freely moving organisms at high resolution, or jointly across multiple individual organisms (e.g., in 96 well plates). To examine its potential for fluorescence image/video capture, we integrated fluorescence imaging capabilities into a separate standalone 24-sensor MCAM unit (*Appendix 1—figure 8A*). Specifically, we added two excitation sources (high-powered light emitting diodes (LEDs)) to illuminate the entire arena. Lateral excitation illumination uniformity was experimentally assessed by measuring the intensity response of the MCAM to a uniform diffuser at the specimen plane, which yielded <6% intra-camera intensity variation (primarily caused by vignetting) and <5% inter-camera mean intensity variation. We outfitted each LED with its own short-pass excitation filter (Thorlabs, FES500) to reduce the excitation bandwidth. We mounted a bank of 24 emission filters via a 3D printed filter array holder over each camera lens (*Appendix 1—figure 8A and B*). With these two simple additions, the MCAM easily captured epi-illuminated wide-field fluorescence images from each MCAM sensor.

In a first set of experiments, we imaged freely moving third instar *Drosophila* larvae (*Appendix 1—figure 8C*, *Appendix 1—video 1*), expressing GCaMP in the brain and ventral nerve cord (*Grueber et al., 2003*; *Vaadia et al., 2019*). The epi-fluorescent setup allowed us to observe an extremely bright convergence of proprioceptive neurons in the anterior/ventral regions of the larva. Similarly, *Appendix 1—figure 8D* shows GFP expression localized to the cranial nerves of a 6 day old zebrafish *Tg(isl1a:GFP)* (*Higashijima et al., 2000*). *Appendix 1—figure 8E* displays an example cropped image frame of a freely swimming 6 days post-fertilization (dpf) transgenic zebrafish larvae with GFP-labeled hair cells (labeled following the process outlined in *Monesson-Olson et al., 2014*).

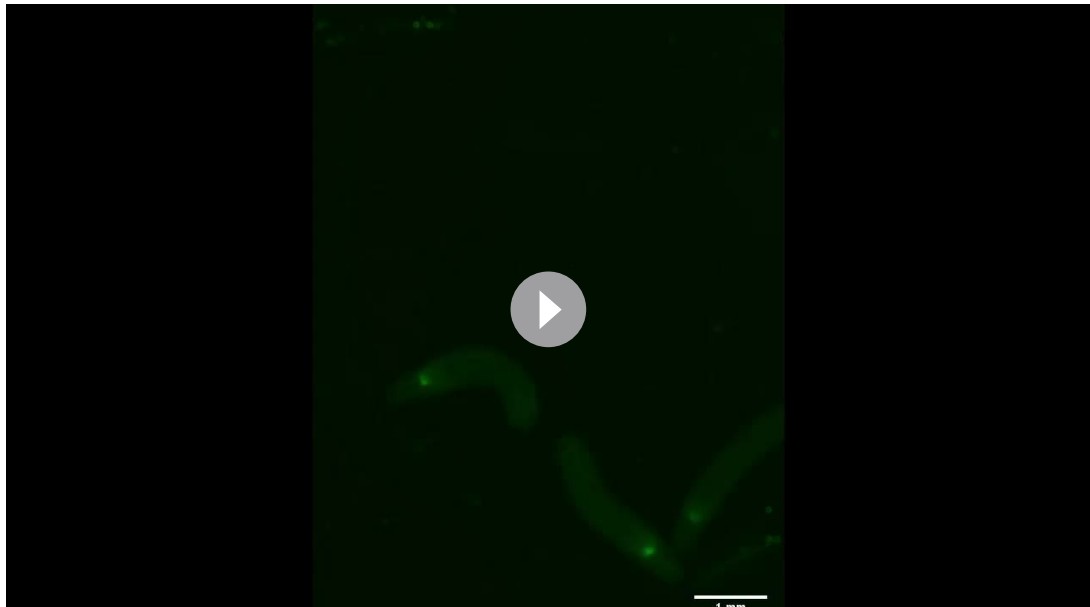

**Appendix 1—video 1.** *Drosophila* larva fluorescence demonstration. Fluorescence video of three freely moving *Drosophila melanogaster* larva expressing GFP imaged at 10 Hz (genotype: w; UAS-CD4tdGFP/cyo; 221-Gal4). https://elifesciences.org/articles/74988/figures#video1

To test the capability of the MCAM to record fluorescence from within freely moving organisms, we used the 24-camera MCAM to record video at 1 frames/sec. for 5 minute imaging sessions of 14 freely moving zebrafish within an 8x12 cm swim arena (*Appendix 1—figure 8F*). These zebrafish genetically encoded fluorescent calcium indicator GCaMP6s across almost all neurons across the brain via the *elavl3* promoter. Example tracking trajectories and snapshots of cropped frames surrounding tracked and registered zebrafish brains for five example organisms (fish 3, 4, 8 11 and 14) are shown on the right hand side of *Appendix 1—figure 8F* Similar to the case of bright-field imaging, fluorescence acquisition of signal from moving organisms also suffers from axial blurring. Effective results were collected when organism axial movement was constrained by using a relatively shallow water depth (typically several mm). Additional details about the

fluorescence image tracking algorithm are provided below (*see Localizing and tracking zebrafish under fluorescence*).

We also utilized the MCAM's FOV overlap to perform dual-channel fluorescence imaging (i.e., capturing two unique spectral bands of fluorescence emission simultaneously). We installed alternating red and green, fluorescent emission filters over the MCAM's lenses to image both red and green fluorescence (*Appendix 1—figure 8G–H*). Here, we imaged double transgenic zebrafish *Tg(slc17a6b:LOXP-DsRed-LOXP-GFP-) x Tg(elavl3:GCaMP6s)* (*Dana et al., 2019*; *Koyama et al., 2011*) which express a red fluorescent protein (DsRed) in GABAergic neurons and the green calcium sensor in almost all neurons. Since the red fluorescent protein is not changing, the red channel image can serve to normalize the GCaMP6s fluorescence and therefore can compensate for movement artifacts that might otherwise be interpreted as neural activity dependent fluorescence changes (*Appendix 1—figure 8I and J*). We note that this double transgenic zebrafish (without cre) only expresses DsRed and GCaMP (no GFP is expressed in these fish). The current MCAM setup measures wide-field fluorescence, and thus integrates fluorescent signal from across its full depth-of-field.

While this capability can potentially be useful for non-localized (i.e., low-resolution) fluorescence monitoring across many freely swimming larvae, effects caused by organism axial movement and tilt must be accounted for during such an analysis. Other sources of noise include slightly non-uniform excitation illumination, lens vignetting, animal pose and brain angle, and a low imaging numerical aperture. Additionally, the use of Bayer-patterned image sensors further limited the fluorescence sensitivity of these initial experimental tests. Finally, we note that because of the alternating spatial layout of the red and green emission filters (*Appendix 1—figure 8h*), red and green fluorescence emission channels from a particular spatial location are captured from unique perspectives, leading to a depth-dependent stereoscopic shift between the two channels within thick specimens. In other words, the same stereoscopic effect that facilitates 3D behavioral tracking may serve as a confound for accurate ratiometric detection and must be taken into account within future post-processing strategies.

In the reported experiments, we employed constant wide-field excitation throughout the video recording, which can potentially lead to photobleaching effects. Our longer video acquisitions (e.g., of the freely moving zebrafish) lasted 250 seconds, within which we did not observe significant photobleaching effects. A strobed excitation source could help to alleviate photobleaching issues. We further note that the MCAM presented here collects 8-bit image data. Due to inhomogeneities in sample fluorescence intensity and illumination, 8-bit image data can lead to severe dynamic range and quantization noise effects, as compared to the use of 12-bit or 16-bit image data acquisition. We aim to address many of these shortcomings with updated hardware and software for MCAM fluorescence imaging in future research efforts.

## *Drosophila melanogaster* **for fluorescent imaging**

Green, fluorescent flies (genotype: w; UAS-CD4tdGFP/cyo; 221-Gal4) were kindly provided by Dr. Rebecca Vaadia. Flies were raised with standard molasses food at room temperature. For fluorescent imaging, 3rd-instar larvae were transferred from the fly food to a drop of water using wood applicators. A drop of water was used to constrain their moving area (decrease forward movement and increase rolling behavior to visualize the side view). For adult imaging, a white 3D-printed arena (dimensions 12 cm x 20 cm, wall thickness 1.5 mm, and height 5 mm) was set between two glass plates.

## Freely swimming zebrafish larva for fluorescent imaging

In-vivo epi-fluorescence imaging in the MCAM system with freely swimming larval zebrafish was performed in transgenic zebrafish *Tg(isl1a:GFP)*(*Higashijima et al., 2000*), *TgBAC(slc17a6b:LOXP-DsRed-LOXP-GFP)*(*Koyama et al., 2011*), and *Tg(elavl3:GCaMP6s)*(*Chen et al., 2013*) 5–7 days-post-fertilization. Zebrafish were placed in a custom 3D-printed arena filled with E3 medium to a height of 1–2 mm. Arenas were designed using Fusion360 (Autodesk; CA, USA). For fluorescence imaging, the arena was rectangular with curved corners, with dimensions 9 cm x 7 cm, wall thickness 1.5 mm, and height 5 mm. The arena was printed with black PLA (Hatchbox, CA, USA) and glued onto a glass sheet using black hot glue around the outside of the arena. For data shown in *Appendix 1—figure 8F*, 14 zebrafish were placed in the arena and spontaneous activity was recorded for 10 minutes, at 1 Hz.

## Localizing and tracking zebrafish under fluorescence

After each fluorescent image set was stitched to form a video of stitched composite frames (following the same process as with the bright field images described above), we used OpenCV to segment and localize the fish for each individual frame via a simple thresholding algorithm. We calculated the center point of each segmentation mask and used two consecutive center points to obtain the Euclidean distance, which was then used to calculate speed, as specified in detail below. For those frames in which the fish moved too fast to track, and there was motion blur, frames were not used for analysis; however, omitted frames were accounted for in final speed calculations.

Once the center point of each segmentation mask for each individual organism was localized, we used each organism's 2D x-y coordinates $(x_i^t, y_i^t)$ for organism $i = 1, 2, \ldots M$ at frame $t$ to predict organism speed and trajectory. We accomplished this via an optimization approach that allowed us to jointly identify matching organisms and their corresponding 2D spatial displacement in subsequently recorded video frames. We aimed to minimize the total cross-organism displacement D by finding the binary assignment variables $g_{i,j}^t$ that minimized the sum of per-organism displacements g between two consecutive frames,

$$D = \sum_{i,j}^M g_{i,j}^t d_{i,j}^t$$

Here, the per-organism displacement $d_{i,j}^t = \sqrt{\left(x_i^t - x_j^{t-1}\right)^2 + \left(y_i^t - y_j^{t-1}\right)^2}$ is the displacement if the organism at $\left(x_j^{t-1}, y_j^{t-1}\right)$ in frame $t-1$ travels to the position $(x_i^t, y_i^t)$ in frame $t$. Since we assume that only one organism can travel to one specific position within a particular next frame, we assume $g_{i,j}$ must satisfy,

$$\sum_i^M g_{i,j} = 1, \sum_j^M g_{i,j} = 1, g_{i,j} \epsilon \{0, 1\}$$,

In other words, if we construct a matrix whose $i, j^{th}$ element is $g_{i,j}$, the only one element of each column or row of such a matrix can have the value 1, and the remaining elements of the matrix must be zero. $g_{i,j}^t = 1$ if the $j^{th}$ fish at frame $t-1$ swims to $i^{th}$ position at frame $t$, and is $=0$ otherwise. We attempt to find the assignment of binary values within the matrix g that minimize the total travel displacement with the Hungarian algorithm, which has a time complexity of $O\left(n^3\right)$. Once we figure out the binary assignments of g, we can calculate the distance that each fish travels between frame $t$ and $t-1$. For example, if it is determined that $g_{3,5} = 1$, this implies that zebrafish 3 at frame $t$ moves from position 5, $\left(x_5^{t-1}, y_5^{t-1}\right)$ at frame $t-1$. Hence, we can compute the total displacement of zebrafish 3 between the two frames as $d_{3,5}^t = \sqrt{\left(x_3^t - x_5^{t-1}\right)^2 + \left(y_3^t - y_5^{t-1}\right)^2}$, and its approximate speed is $d_{3,5}^t / \epsilon$, where $\epsilon$ is inter-frame time delay.

## Quantitative Fluorescence Analysis

Captured images were first de-Bayered into three channels (red, green, and blue). We used only the green channel for GFP analysis from those individual cameras that included an associated GFP emission filter (in both single and dual imaging). For red (dsRed) fluorescence analysis in dual-channel imaging, we used only the red channel from those individual cameras that included an associated RFP emission filter. We computed the total energy within each fluorescent channel per organism per frame by computing the sum of associate GFP/RFP channel intensity values within each segmented fish brain area *Appendix 1—figure 8G*. For accurate comparison in the dual-channel fluorescent imaging scenario, we required that the same number of pixels to be considered for both channels, which we achieved by first registering the two segmentation masks from the green and red channels and then producing a common segmentation mask as the union of these two masks. Using the common mask, we then measured each channel's total energy.

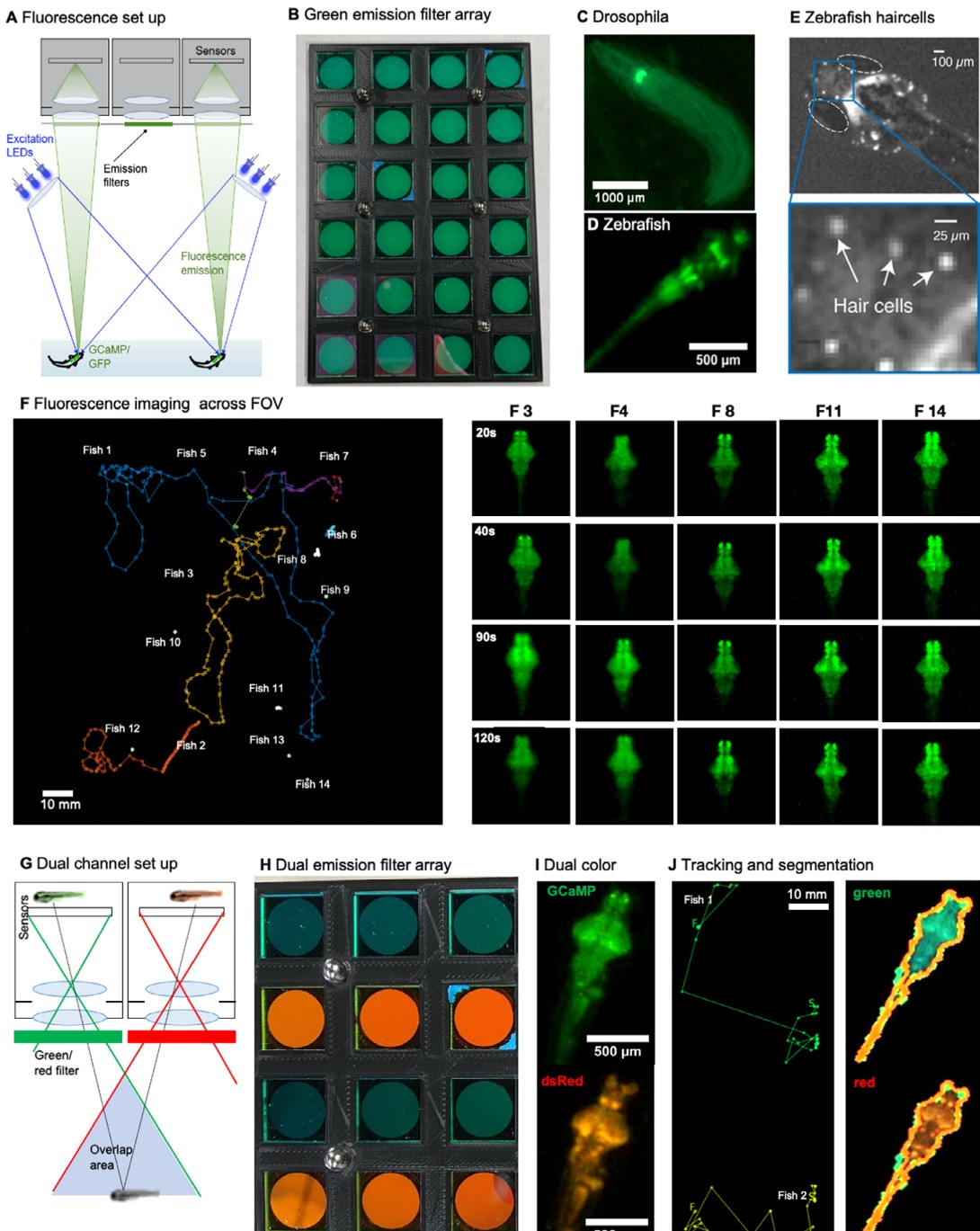

**Appendix 1—figure 8.** Single and dual channel wide field fluorescence imaging with the MCAM. (**A**) Schematic of MCAM fluorescence imaging set up. Wide-field single-channel fluorescence imaging was implemented with two external high-power LEDs with a wavelength of 470 nm (Chanzon, 100 W Blue) for excitation. Each excitation source additionally included a short-pass filter (Thorlabs, FES500). We then inserted a custom-designed emission filter array containing 24 filters (525±25.0 nm, Chroma, ET525/50 m) directly over the MCAM imaging optics array to selectively record excited green fluorescent protein and GCaMP. Dual-channel fluorescence imaging was accomplished with a set of blue LEDs (470 nm, Chanzon, 100 W Blue) combined with a 500 nm short-pass filter (Thorlabs, FES500). For emission, we used 510±10.0 nm filters (Chroma, ET510/20 m) for *Tg(elavl3:GCaMP6s)* (green) signals and 610±37.5 nm filters (Chroma, ET610/75 m) for slc17a6b (red) signals. (**B**) Custom 3d-printed array of 24 emission filters fitted over each camera lens. (**C**) Example fluorescent MCAM image zoom of fluorescent *D. melanogaster* instar 3 larva (410Gal4x20XUAS-IVS-GCaMP6f), showing fluorescence can be detected in proprioceptive neurons. See *Video 11*. (**D**) Localized expression of GFP in cranial nerves and brain stem motor

*Appendix 1—figure 8 continued*

neurons in *Tg(Islet1:GFP)* transgenic zebrafish. (**E**) Snapshot and zoom in of 6 day old zebrafish with fluorescently labeled hair cells demonstrates imaging performance that approaches cellular level detail. (**F**) Fourteen freely moving zebrafish, *Tg(elavl3:GCaMP6s)* in a 24-camera MCAM system, tracked over a 120 s video sequence, similar imaging experiments were repeated 17 times. The right hand panel shows the detail of five fish at four different time points. See *Appendix 1—video 2*. (**G**) Dual channel fluorescence imaging configuration. As each area is imaged by two MCAM sensors, it is possible to simultaneously record red and green fluorescence emission.
(**H**) Close-up of emission filters within array used for simultaneous capture of GFP and RFP fluorescence signal for ratiometric video of neural activity within freely moving organisms (4x3 section of 24-array bank of filters shown here). (**I**) Example dual-channel images of the same double transgenic zebrafish *TgBAC(slc17a6b:LOXP-DsRed-LOXP-GFP)* x *Tg(elavl3:GCaMP6s)*. (**J**) Example tracking and segmentation from ratiometric recording setup. Left: Example stitched MCAM frames across 8x12 cm FOV of simultaneously acquired green (GFP) and red (RFP) fluorescence channels, with two freely swimming zebrafish larvae exhibiting both green *Tg(elval3:GCaMP6s)* and red *Tg(slc17a6b:loxP-DsRed-loxP-GFP)* fluorescence emission. At right is tracked swim trajectories over 240 sec acquisitions, from which swim speed is computed. Right: Example of automated segmentation of a double transgenic fish in the red and green channel from the recording shown on the left. See *Appendix 1—video 3*.

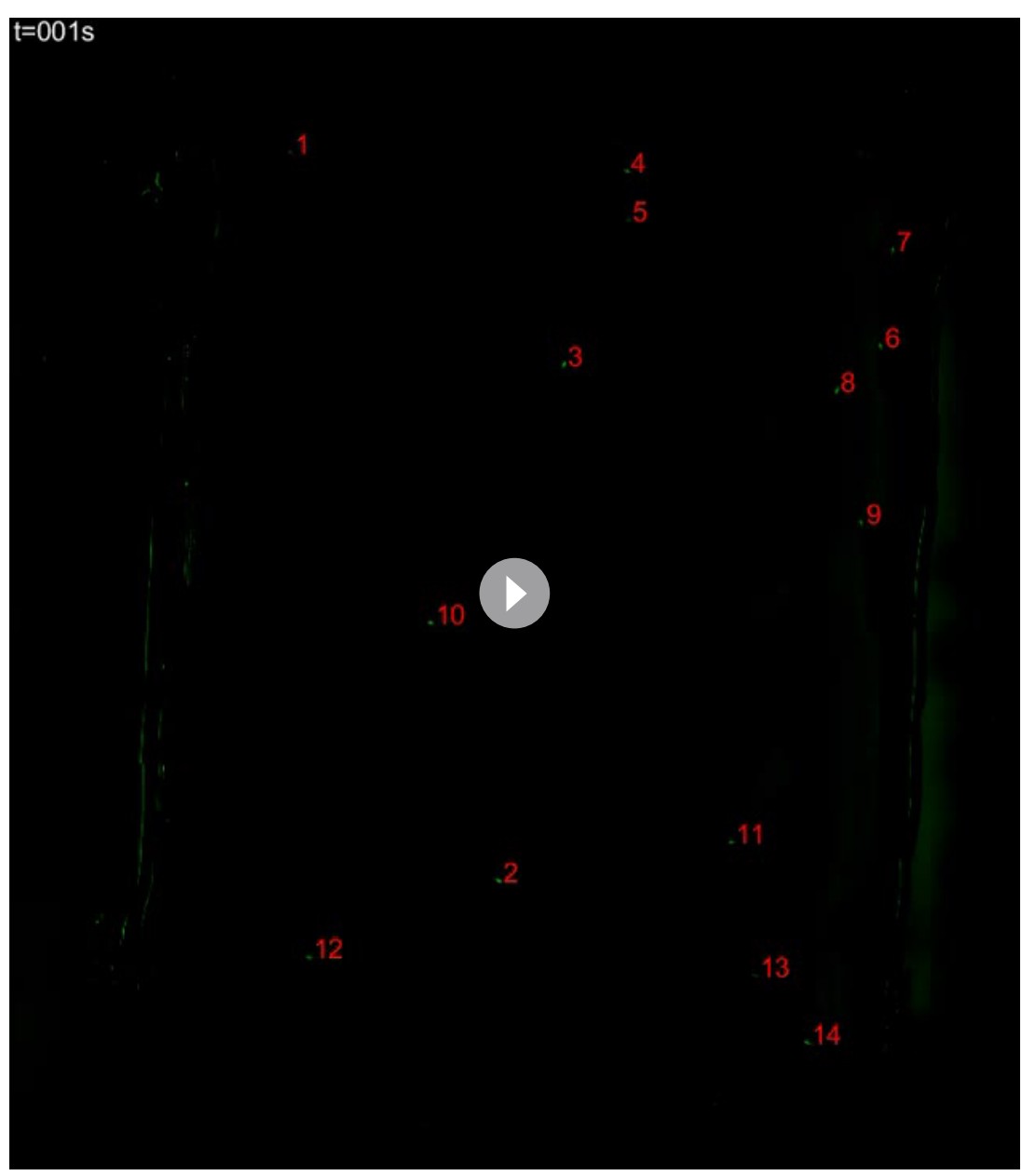

**Appendix 1—video 2.** Tracking of green fluorescent zebrafish expressing GCaMP across the brain. Fluorescence video and tracking demonstration, 14 unique freely swimming zebrafish expressing GCaMP quasi panneuronally across the brain. 8x12 cm FOV at 18 μm full-pitch resolution using MCAM-24 array (240 captured megapixels/frame, imaged at 1 Hz).

https://elifesciences.org/articles/74988/figures#video2

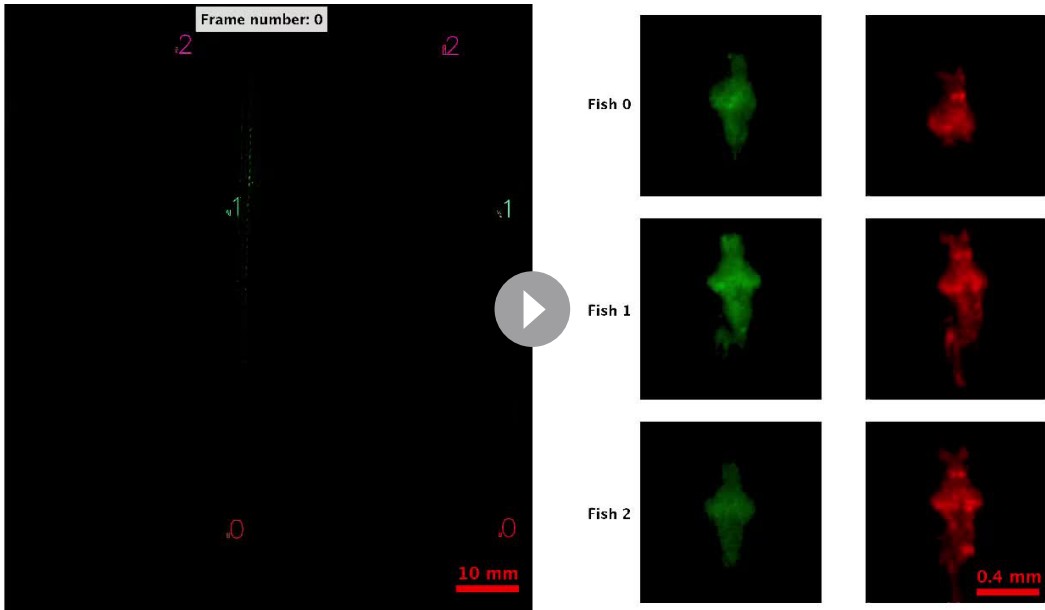

**Appendix 1—video 3.** Dual mode zebrafish fluorescent imaging. Video of freely swimming transgenic zebrafish expressing the green fluorescent genetically encoded calcium sensor (GCaMP6s) and stable red fluorescent protein (dsRed) in almost all neurons and only GABAergic neurons, respectively. In future design iterations, simultaneous dual-channel MCAM recording of both red fluorescence (dsRed) and green fluorescence (GCaMP6s) could potentially allow ratiometric measurements by normalizing the fluctuating green fluorescence reflecting changes in neural activity with the stable signal in the red channel. Two color channels are shown on the left across full MCAM FOV, with tracked zoom-ins of each fish shown to right.

https://elifesciences.org/articles/74988/figures#video3

