## [Editor Report]

This paper presents a valuable method of mesoscopic imaging for behavioral neuroscience, particularly of high potential in applications such as tracking behaving subjects in 3D arena simultaneously with a neural population activity readout. The technical and conceptual advances are based on solid presentations of the engineering and the pilot experiments. Readers of this paper are advised to first gain a deeper insight of its working principle as well as the consequent advantages and caveats of this method before applying it in their own labs.

---

## [Decision Letter]

**Decision letter after peer review:**

Thank you for sending your article entitled "Gigapixel behavioral and neural activity imaging with a novel multi-camera array microscope" for peer review at *eLife*. Your article is being evaluated by 3 peer reviewers, and the evaluation is being overseen by a guest Reviewing Editor and Didier Stainier as the Senior Editor.

The current reviewer consensus is that you are invited for a revision in which, the fluorescence imaging data shall be removed from the main text and the behavior imaging data shall be further improved primarily by addressing reviewer 1's comments. By doing so, the data quality and scientific value will meet the expectations of the journal. Other reviewer comments shall also be adequately addressed. If you accept these suggestions, please reply with a confirmation and proceed with revising your manuscript and resubmission. We will then advise further as soon as possible. Thank you very much.

*Reviewer #1 (Recommendations for the authors):*

There are really two applications here, one on large-scale behavioral imaging of multiple animals in parallel, and another on fluorescence imaging. The behavior story is clearly viable – it (nearly) achieves the titular gigapixel claim, makes effective use of stereo imaging to localize animals in 3D, and has already provided much of the characterization necessary to merit publication. The same cannot be said for the unconvincing attempt at fluorescence imaging, especially quantitative neural imaging, in behaving animals.

If this is just about behavior, with some added clarification and characterization, the manuscript would be fine. For fluorescent imaging, it's unconvincing that this can produce credible neural data. There are some serious conceptual issues with the fluorescence imaging experiments (discussed in the fluorescent imaging section below).

It might be advisable for the authors to split this into two, so that the behavioral side is not held up by serious design flaws on the fluorescence side.

Behavior

1. The manuscript contains repeated claims to be "inexpensive", but doesn't state how much it costs. The lenses alone appear to be >$7,000. 16 FPGAs and large custom circuit boards aren't cheap either. For fluorescence, filters alone would apparently add another $8,000 per MCAM or >$31,000 at gigapixel scale. Since the hardware and firmware are not open source, the company will presumably add significant additional costs on top of the basic material costs, which are already considerable.

2. "…comprehensive high-resolution recording from multiple spatial scales simultaneously, ranging from cellular structures to large-group behavioral dynamics." With 18 μm resolution (or worse, discussed below), the present invention does not record cellular structures except under extreme sparsity conditions. Even then, the measurements may not be sound (see fluorescent imaging discussion). It's unclear if "high-resolution" refers to behavioral imaging or fluorescence imaging, a conflation that occurs throughout the manuscript (see point #3 below).

3. The introduction conflates systems designed for neural imaging and systems designed for behavioral imaging. Throughout, it's unclear whether the authors are attempting to claim an advance in behavioral imaging or neural imaging. Existing fluorescent microscopy methods attempt to address real issues such as axial resolution, dynamic range, photobleaching etc. These are all problems that the current MCAM system does not attempt to address. For example, consider this statement: "Alternatively, closed-loop mechanical tracking microscopes have recently been developed in an attempt to address this challenge, but can only follow single organisms and thus miss inter-organism interactions." This statement is inaccurate in numerous ways: The citations jumble together systems for behavioral imaging only and systems for neural and behavioral imaging, which have fundamentally distinct requirements. It also misses recent work (Susoy et al., 2020) that used closed loop tracking to simultaneously image behavior and neural activity during worm mating, a form of inter-organism interaction. Susoy et al. were able to achieve both cellular resolution brain-wide imaging with sound microscopy techniques and observe sustained inter-organism interactions. Similarly, Grover et al. 2020 simultaneously imaged neural activity and behavior in copulating flies. The authors should take a more scholarly approach here, and make a clearer case for the merits and limitations of their system.

4. Regarding inter-organism interactions, this is overstated and confusing. In tracking microscopes, the field of view restriction is typically only on the microscopy side – the associated behavioral field of view is usually far less restricted. It is not difficult to simultaneously observe the behavior of multiple animals. Numerous existing methods using off-the-shelf parts and open source software already do so (Buchanan et al. 2015 as one of many examples).

5. The authors' claim of 147 lp/mm resolution appears to neglect depth of field / defocus issues as animals move axially. According to the lens datasheet, even a 20 lp/mm sample loses substantial contrast as it moves axially over a range of a couple millimeters. This degradation in resolution will be even more severe at 147 lp/mm.

6. "We performed standard flat-field correction to compensate for vignetting and other brightness variations within and between cameras (Methods)". This needs more detail. What are "other brightness variations"? Give numbers for the amount of brightness variation within and between cameras. This is especially important because inexpensive smartphone sensors assume a tilted chief ray angle (CRA) as a function of image height, which the authors do not discuss. How mismatched is the CRA with respect to the sensor's design assumption, and how much does this impact vignetting?

7. "Using this stereoscopic depth measurement approach yields approximately 100 μm accuracy in depth localization over a >5 mm depth range". How was this 100 μm accuracy number determined?

8. There is a troubling disregard to axial extent. In one instance, there is a reference to 5 mm height. In the case of zebrafish fluorescence imaging, the height is stated as 1-2 mm. Axial defocus should be characterized, and some statement of the practical usable axial imaging extent should be provided.

9. With regard to the siamese network / melanophore patterns, are the embeddings stable over recording time, animal pose, etc? There is insufficient detail to evaluate this. With regard to "250 augmented images per fish", how many real images of the animal were used, before augmentation? Is Figure 2c based on training or test? The number of real images used for training and test appears much too low to address questions of stability over time and animal pose, for example. Why is there no quantitative comparison of performance of training vs test?

Fluorescence imaging

There are some deeply concerning issues with the design that are not discussed in the manuscript. The manuscript seems to neglect the fact that existing microscopes have axial resolution, in addition to lateral resolution. Setting aside the necessity for more accurate characterization, there are some fundamental issues that are inherent to the imaging approach.

Fundamental problems with the design:

1. Without any axial sectioning capability, the fluorescent signal can be corrupted by the position and orientation of the animal. As the animal moves across the field of view of each microcamera, the detection PSF (point spread function) tilts in a position-dependent way, such that a different set of signals across the axial extent of the thick tissue are projected onto the same sensor pixel. Similarly, each time the pose of the animal changes, each pixel now is the summation of signals from a fundamentally different set of cells and even regions across the brain.

2. Ratiometric imaging cannot be done when the green and red fluorescence comes from different cells! This combined with point #1 above, means that every time the animal changes its pose, a given red fluorescent pixel is potentially summing photons from a different set of cells and brain regions than the "matching" green fluorescent pixel. This is fundamentally not sound. (The dual-color images in Figure 3h show obvious labelling differences, as expected from the use of different promoters and cell types in the two color channels.)

3. Related to #2, the red and green fluorescent light is collected from different cameras. This means that due to point #1, even if the same promoter was used for the green and red channels (as it should), each sensor pixel of the green image is sampling a different projection through the thick tissue compared to the corresponding sensor pixel of the red image. Applying cross-channel normalization of thick fluorescent tissue using this imaging geometry is fundamentally unsound.

4. Both lateral and axial resolution are depth dependent. Without axial sectioning capability, as the animals moves up and down in z, each point in the sample (e.g. a cell) contributes a different angular distributions of photons to a given pixel on the camera. To put this simply, a pixel on the camera is sampling a different distribution of signals depending on the axial position of the animal. This cannot be solved by ratiometric imaging. The information contained within each pixel is fundamentally changing as a function of depth. There is a reason that standard microscopy techniques go to great length to adjust focus as a function of depth.

5. The authors do not provide standard lateral and axial point spread functions (PSF) (i.e. using a submicron fluorescent bead), or demonstrate how both lateral and axial PSF change across x, y, and z. This is a standard optical characterization for a fluorescent imaging system.

6. Illumination brightness is not expected to be homogenous across x y and z. There is no discussion of brightness correction for either excitation or collection light across 3 dimensions. For characterization, the authors would need to show the results of a thin (submicron) fluorescent sample across the entire field of view, and across z.

7. Need a detailed analysis of signals from the red fluorescent protein as control. In the ideal case, the measurement from the red channel should of course be flat, but reporting the reality is important to establish the effective noise floor of the technique. The most problematic is systematic noise – intensity changes related to movement, lateral or axial position, animal pose or brain angle, all of which can lead to spurious correlations that complicate the use and interpretation of the fluorescence data in quantitative studies of animal behavior. Claiming detection of movement-related GCaMP activity is not that meaningful, as numerous studies have recently demonstrated that movement causes widespread activation throughout the brain.

8. The system is designed for visualizing inter-animal interactions, but there is no fluorescent data obtained during inter-animal interactions. When animals come into contact with one another, or overlap on z, a given pixel on the camera may become the summation of photons from both animals, further corrupting the neural signal.

9. If the system is to be evaluated as a microscope, the authors should provide a bleaching curve. Is the blue light on all the time, or is it strobed? How long can a user realistically image with this system?

10. There is no discussion on how a user is supposed to calibrate this system in 3D for fluorescent imaging. Though it's unclear to what extent calibration can alleviate the core conceptual issues embedded in the design.

11. The 2x2 Bayer filter is problematic, particularly for the fluorescence imaging case, where the effective number of imaging pixels isn't a gigapixel, or even a quarter of that, but in fact an 1/8th (green) or a 1/16th (red) of that. In the red channel, the effective pixel pitch appears to be around 18 um, which means the claimed 18 μm two-point separation is demonstrably false. For example, consider two (red fluorescent) point sources located at the midpoints of the 18 μm effective pixel pitch, spaced apart by 18 um. These will be detected as one bright red pixel flanked by two half-intensity red pixels, with no separation whatsoever. In the green channel, the effective pitch on one diagonal axis is twice the effective pitch of the other diagonal axis, and the same resolution argument holds.

12. The system that has been presented collects only 8 bit image data. Typical bit depths in microscopy systems range from 12 bit to 16 bit. The restriction to 8 bit is particularly problematic because due to inhomogeneities in fluorescence intensity of the sample and intensity of the illumination, combined with the desire to avoid saturated pixels which prevent extraction of quantitative signals, with 8 bit data one ends up with a severely limited dynamic range and severe quantization noise. The authors make no mention of the unconventional use of 8 bit data for fluorescence imaging, nor the associated caveats.

Claims that are not sufficiently substantiated or slightly misleading:

"To reduce motion artifacts, we started by embedding zebrafish in low melting point agarose and recorded spontaneous neural activity for 5 min". This should be removed from the paper, as it avoids all of the problems introduced by the MCAM imaging approach, most importantly brightness and focus changes over the FOV of each microcamera and over the 1-2 mm bath depth, as well as animal position-dependent and pose-dependent axial projection of thick brain tissue onto the 2D image sensor. The legend text greatly overstates: "Average ∆F/F traces of these ROIs show that differences in neural activity are easily resolved by the MCAM." To even partially address the many challenges posed by a freely behaving MCAM calcium imaging experiment, the authors should suspend the embedded animal from a motorized stage (e.g. a glass slide overhanging a typical lab XYZ translation stage) and then capture images of the brain over the ~ 38 mm x 19 mm field of view (including the corners of the rectangular field of view, which represent the most aberrated condition of the imaging lens), as well as the 1-2 mm depth axis that freely behaving animals move in. Note, I estimated the field of view of each microcamera to be 38 mm x 19 mm based on the sensor pitch of 19 mm and the 2:1 overlap strategy, but the authors should clearly state the actual numbers in the methods.

" While the MCAM in the current configuration cannot detect single-neuron fluorescence activity in the zebrafish brain, the data do show that our multi-sensor architecture can measure regionally localized functional brain activity data in freely moving fish". The data do not demonstrate sound optical principles and analysis. It's not just a matter of sensor resolution – the complete lack of axial sectioning/resolution is a fundamental problem. The wording appears to imply that MCAM eventually could, but this is impossible to evaluate.

Calibrated videos of freely swimming transgenic zebrafish with pan-neuronal GCaMP6s expression verify that our system can non-invasively measure neural activity in >10 organisms simultaneously during natural interactions. The word "calibrated" is unclear. What precisely was done? If the authors simply mean "flat-field corrected", then say that, or else add a description of this "calibration" in the methods and refer to it here.

"…the latter can increase both the information content and accuracy of quantitative fluorescence measurement in moving specimens". Due to the differences in view angle dependent axial projection of thick tissue onto a 2D sensor, angle-dependent sample occlusion (e.g. overlying pigment or blood cells), and other above-mentioned uncorrected differences in imaging performance across a large 3D imaging area, the current approach is fundamentally unsound. Claiming that it suitable for accurate quantitation is indefensible.

"…there are likely a wide variety of alternative exciting functionalities that MCAM overlapped sampling can facilitate". As an existence proof, please give one or more examples.

"i.e., the ability to resolve two 9 μm bars spaced 9 μm apart": This is misleading as stated. This should be described as "a center to center spacing of 18 um".

"…we used custom Python code to produce gigapixel videos from the image data streamed directly to four separate hard drives". A solid state drive is not a "hard drive". Just say "solid state drives" or "rapid storage drives".

*Reviewer #2 (Recommendations for the authors):*

This manuscript provides a useful tool for mesoscale imaging in complex biological systems. Simultaneous imaging at a large field-of-view and high resolution has been a challenge for biological studies, resulting from the optical aberration and data throughput of sensors. In this work, Thomson et al. proposed a flexible framework that enables multi-channel parallel imaging of 96 cameras from a field of view up to 20 cm*20cm and at a resolution of 18 μm. They have demonstrated various interesting experiments, which only be achieved with these mesoscale imaging and may arouse great interest.

Strengths:

This paper remains a great engineering effort. In order to achieve a large field view and high resolution, the authors build the 0.96 gigapixel camera array with 8*12 camera sensors, with 16 FPGA routing the image data to the computer. Two different imaging modes are developed: a full-frame mode with 96 cameras at 1 HZ and a single sensor mode with 4 cameras at 12 Hz, corresponding to fast and slow applications.

Also, I like the idea of stereo-imaging. Spatial overlapping is inevitable in image stitching, and most previous works tried to reduce the overlapping area to increase the data throughput and to reduce the inhomogeneous image. In this manuscript, all the imaging area is imaged simultaneously by at least two cameras, so the whole image is homogenous and the extra multi-view information is utilized for axial localization.

The authors have verified their system parameters in a series of biological experiments, such as zebrafish, *C. elegans*, and Black carpenter ants, which indicates its wide scope of applications. I believe all these experiments are quite interesting to a broad audience, e.g. social behavior for sociologists. Overall, I think the MCAM configuration is a promising initial scaffold for parallel image acquisition.

Weaknesses:

1. Although the paper does have strengths in principle, the weakness of the paper is that the strength of this system is not well demonstrated. The fundamental limitation of this system is the imaging rate is far too slow for the dynamic recording of various demonstrated applications, such as zebrafish's collective behaviors. The authors have realized this drawback and discussed it in the manuscript, but they do not give a practical estimation of how fast it could be, with up-to-date technologies. Is it possible to improve the speed by using different computers simultaneously for distributed data acquisition, as in Fan et al. Nature Photonics 13:809-816, 2019.?

2. Another concern is the asynchronized trigger mode for different cameras. When the exposure time of each image is short to avoid image blur, it is likely that all images are taken at different time points. Is it possible to use an external trigger to synchronize all cameras, or is it possible to output timestamp for each image, so that the error of time difference could be estimated?

3. In abstract and supplementary figure 2, the authors claim their system has 5 μm sensitivity, but this is confusing. The fact that this system can capture light from 5 μm beads, but not from 3 μm beads, is that the signal from 3 μm beads is not strong enough to distinguish itself from background noise. For example, if the illumination is strong enough, it is possible for microscopic systems to image small nanoparticles, but this does not indicate those systems have nanometer sensitivity. I understand that light collection efficiency is a serious concern for such low NA microscopic systems, but they should compare the light efficiency in a more persuasive way.

4. Another concern is the dual-color imaging. The fact that this system enables stereo-imaging seems to contradict dual-color imaging for 3d objects. Imagine there are two light sources at different depths, they could overlap on one camera but separate on another. I think more proper demonstrations are 2D samples, such as a disk of dual-color cells or worms.

*Reviewer #3 (Recommendations for the authors):*

The authors aimed at a low-cost technical solution for large-array gigapixel live imaging for biological applications. The endeavor is highly appreciated by the community as existing devices that are capable of such high volume of biological imaging are unaffordable or inaccessible by ordinary biological research labs. However, their current manuscript presented with design and engineering caveats that causally resulted in poor biological signal acquisition performance, as follows:

1) The low numerical aperture (NA, = 0.03) is inacceptable for imaging applications that require cellular resolution. The statement of the authors that their device can acquire cellular-level neuronal activities is unsupported by both optical principle and their demonstrated imaging data (Figure 3). This is due to the authors' optical design using simple lens per each sensor chip. Correct design strategy should be combinatorial, i.e., one set of lens per a subarray of sensor chips, that the optical NA can be sufficiently high to satisfy realistic cellular resolution in living organisms.

2) The image data streaming rate as 1Hz for full-frame Gigapixel image is too low for most biological applications that require temporal resolution to study behavior or cellular dynamics. However, this is an engineering bottleneck, not methodological. As a single sensor chip used in this study, the OMNIVISION's OV10823, does indeed support full-frame streaming at 30 Hz – which is nowadays commonly accepted in biological imaging community as video-rate imaging. The bottleneck is caused by the choice of low-cost USB3.0 transfer protocol and a single PC workstation. Solution to this issue is simple: parallelize the data transfer by using multiple PCs with either timestamp markers (e.g., synchronized indicator LEDs through the optical system per each sensor) or synchronized triggered acquisition. There is of course, a new concern to balance the cost/performance ratio.

3) the claim of 'cellular-level' signal or resolution has to be removed throughout the manuscript as the authors did not provide any evidence that what they claim as cellular objects or signals are indeed from individual cells. Proper wording for this type of low-resolution, low-sensitivity signal can be: cell population activity.

4) The authors made major efforts in applying machine learning algorithms for identifying and tracking objects in the stitched Gigapixel image, however, the authors did not consider the cost for implementing such working pipelines to handle such high volume of raw imaging data for obtaining biologically meaningful signals and analysis results. In particular, for tracking just a few animals as the authors presented in data, how much gain in performance is enabled by their device as compared to existing well-established methods using a single or few imaging sensors?

These concerns together suggest that the authors' design and engineering did not enable their claim of expected performance. Low-cost device is good for advancing science, however, only in condition when scientific criteria can be met.

---

## [Author Response]

Reviewer #1 (Recommendations for the authors):There are really two applications here, one on large-scale behavioral imaging of multiple animals in parallel, and another on fluorescence imaging. The behavior story is clearly viable – it (nearly) achieves the titular gigapixel claim, makes effective use of stereo imaging to localize animals in 3D, and has already provided much of the characterization necessary to merit publication. The same cannot be said for the unconvincing attempt at fluorescence imaging, especially quantitative neural imaging, in behaving animals.If this is just about behavior, with some added clarification and characterization, the manuscript would be fine. For fluorescent imaging, it's unconvincing that this can produce credible neural data. There are some serious conceptual issues with the fluorescence imaging experiments (discussed in the fluorescent imaging section below).It might be advisable for the authors to split this into two, so that the behavioral side is not held up by serious design flaws on the fluorescence side.Behavior1. The manuscript contains repeated claims to be "inexpensive", but doesn't state how much it costs. The lenses alone appear to be >$7,000. 16 FPGAs and large custom circuit boards aren't cheap either. For fluorescence, filters alone would apparently add another $8,000 per MCAM or >$31,000 at gigapixel scale. Since the hardware and firmware are not open source, the company will presumably add significant additional costs on top of the basic material costs, which are already considerable.

Thank you for pointing out the lack of clear information regarding the total expense of various components of the MCAM. The reviewer’s ballpark estimates for components are relatively close to the costs that we paid (see details below). We had chosen not to include the exact cost of components in the original manuscript as these prices can vary depending upon vendor and quantity of components ordered. We have now revised the manuscript to provide more details regarding the exact cost. Specifically, we discussed system expenses three times in the manuscript. First, we noted the entire system as inexpensive in the abstract. We have since changed this word to “scalable” and only address the costs later:

“We have created a scalable multi-camera array microscope (MCAM) that enables comprehensive high-resolution recording from multiple spatial scales simultaneously.”

Second, we noted the system was inexpensive in the introduction. Our original intent here was to highlight the system’s relatively low cost when compared to prior gigapixel-scale imaging systems, which include multi-million dollar sensor arrays developed for astronomy applications (also noted in the introduction), large lithography lenses, and multi-scale systems such as in (Brady et al. 2012) and (Fan et al. 2019), which cost anywhere between 1-2 orders of magnitude more than the proposed approach. To avoid any confusion, we have rewritten this section of the introduction to read:

“To overcome these challenges, and extend observational capabilities to the gigapixel regime, we designed and constructed an inexpensive, scalable Multi Camera Array Microscope (MCAM) (Figure 1A), a system whose cost scales linearly with the number of pixels. This design is significantly less expensive than other available large-area, high-resolution imaging systems as it leverages relatively inexpensive components (see Methods)”.

Third, we specifically noted that the systems’ digital image sensors were inexpensive in the Discussion section. They are indeed relatively inexpensive ($10-30 each depending on quantities ordered). We now specify their price in the Methods section, along with the approximate price of other components such as the lenses and FPGAs. The modified Discussion section sentence now states the price and points the reader to the Methods section for additional price information:

“…constructed from closely tiling multiple high-pixel count CMOS image sensors that are each relatively inexpensive ($30 per sensor for this prototype – see Methods). Our parallelized imaging design overcomes the limitations of single-objective and single-sensor systems…”

Furthermore, we have added a new Methods section in the revised main manuscript that now contains the following price details:

“MCAM Component costs

Here is a summary of component prices for the prototype gigapixel MCAM imaging system at the time of purchase. We note that prices can vary depending upon quantities ordered, vendors and time of order. (1) Packaged 10 MP CMOS sensors (Omnivision OV10823) were $30 apiece in quantities of 100. (2) Imaging lenses were $75 apiece in quantities of 100. (3) FPGAs were $267 apiece for 16 units. For the presented MCAM with 96 imagers, the total price of these key components is $14,352. Additional costs were incurred for opto-mechanics, PCB manufacturing, associated electronics, illumination, fluorescence excitation and emission filters for fluorescence imaging experiments, and a desktop computer to read acquired data.”

2. "…comprehensive high-resolution recording from multiple spatial scales simultaneously, ranging from cellular structures to large-group behavioral dynamics." With 18 μm resolution (or worse, discussed below), the present invention does not record cellular structures except under extreme sparsity conditions.

We agree with the reviewer that our current resolution can only resolve cellular structures under sparse conditions. For example, it is not possible to resolve single fluorescent neurons in densely packed brain regions in the zebrafish from the data included in the original. However, other sparsely labeled cellular structures, such as melanophores in Figures 1-2 and fluorescently labeled hair cells in the original Appendix 1 -figure 8D, are resolved with a high signal-to-noise ratio. Nonetheless, we appreciate the reviewer’s feedback that this might be misleading, and we have changed the associated text to reflect realistic expectations for this technique. In the revised abstract we now write:

“…comprehensive high-resolution recording from multiple spatial scales simultaneously, ranging from structures that approach the cellular-scale to large-group behavioral dynamics."

Even then, the measurements may not be sound (see fluorescent imaging discussion). It's unclear if "high-resolution" refers to behavioral imaging or fluorescence imaging, a conflation that occurs throughout the manuscript (see point #3 below).

We very much appreciate this feedback and the many helpful comments regarding fluorescence imaging. As noted above, we have removed the fluorescence imaging experiments and their associated discussion from the main manuscript text. This should hopefully remove any confusion about the conflation of the meaning “high resolution” in each context.

3. The introduction conflates systems designed for neural imaging and systems designed for behavioral imaging. Throughout, it's unclear whether the authors are attempting to claim an advance in behavioral imaging or neural imaging. Existing fluorescent microscopy methods attempt to address real issues such as axial resolution, dynamic range, photobleaching etc. These are all problems that the current MCAM system does not attempt to address.

We thank the reviewer for raising this helpful distinction. We originally intended to present the MCAM as a general imaging platform that offers a relatively high resolution and significantly larger field-of-view (FOV) than existing imaging systems for model organism behavior. While it suffers from a variety of real issues, we aimed to show the utility of its large FOV across a variety of different biological experiments and modalities. The two modalities that we focused on were bright-field imaging and wide-field fluorescence microscopy. However, we appreciate the desire to focus on clear technical advances on a specific modality within a single publication. Accordingly, following the *Revision Action Plan*, we have removed the discussion of fluorescence neural imaging and related literature from the introduction and main text, and pushed a revised version to Appendix 1 (previously Supplementary Materials). We hope that this change helps clarify the utility of the platform for bright-field behavioral imaging experiments while showing that in principle wide field fluorescence recordings are possible which may be useful in situations in which cellular resolution is not necessary or desired (e.g., general detection of fluorescence in a fish, or behavioral tracking).

For example, consider this statement: "Alternatively, closed-loop mechanical tracking microscopes have recently been developed in an attempt to address this challenge, but can only follow single organisms and thus miss inter-organism interactions." This statement is inaccurate in numerous ways: The citations jumble together systems for behavioral imaging only and systems for neural and behavioral imaging, which have fundamentally distinct requirements. It also misses recent work (Susoy et al., 2020) that used closed loop tracking to simultaneously image behavior and neural activity during worm mating, a form of inter-organism interaction. Susoy et al. were able to achieve both cellular resolution brain-wide imaging with sound microscopy techniques and observe sustained inter-organism interactions. Similarly, Grover et al. 2020 simultaneously imaged neural activity and behavior in copulating flies. The authors should take a more scholarly approach here, and make a clearer case for the merits and limitations of their system.

We thank the reviewer for their very helpful suggestions regarding methods to improve the clarity of our introduction. We also appreciate that the reviewer pointed out these important missing references. We have now separated statements about prior tracking systems for behavioral recordings and for fluorescence imaging in the text. We have also now updated the revised document to additionally include these Susoy et al., 2021, Grover et al., 2020 and other references as well as worked towards a clearer description of the merits and limitations of the MCAM.

“Closed-loop mechanical tracking microscopes have recently been developed in an attempt to address this challenge and have acquired impressive high-resolution optical measurements of behavior (Johnson et al., 2020; Reiter et al., 2018), fluorescence (Kim et al., 2017; Nguyen et al., 2017), and both behavior and fluorescence (Cong et al., 2017; Susoy et al., 2021; Symvoulidis et al., 2017). While such tracking strategies have recently revealed dynamics between persistent internal states within the brain during unconstrained behavior, for example (Marques et al., 2019). While such tracking systems can in principle observe the behavioral interactions of several organisms at high resolution (Grover et al., 2020; Susoy et al., 2021) they can only center their tracking trajectories to follow one organism at any given time. These systems accordingly do not effectively scale to the challenge of jointly observing the free movement and spatiotemporally varying interactions of many organisms at high resolution over a large area.”

[Susoy et al. 2021], V. Susoy et al., “Natural sensory context drives diverse brain-wide activity during *C. elegans* mating,” Cell 184(20), 1522 (2021) DOI: 10.1016/j.cell.2021.08.024

[Grover et al. 2020], Grover D, Katsuki T, Li J, Dawkins TJ, Greenspan RJ. 2020. Imaging brain activity during complex social behaviors in *Drosophila* with Flyception2. *Nat Commun* 11:623. doi:10.1038/s41467-020-14487-7

In addition, we have expanded a paragraph in the Discussion section to directly address this concern.

“Second, while we have demonstrated the MCAM’s fundamental ability to acquire wide-field fluorescence video of dynamic organism movement (see Supplement), CMOS sensors with increased sensitivity and a higher imaging NA could increase fluorescence signal-to-noise ratio and potentially facilitate fluorescence detect at the cellular scale. Improvements in ratiometric acquisition and the potential addition of optical sectioning could also increase fluorescence signal fidelity, most notably during rapid organism movement. With appropriate effort, the MCAM architecture could soon unlock high-quality measurement of functional fluorescence activity during free organism movement over truly macroscopic areas, thus yielding a new imaging platform for neuroscientific experiment.”

4. Regarding inter-organism interactions, this is overstated and confusing. In tracking microscopes, the field of view restriction is typically only on the microscopy side – the associated behavioral field of view is usually far less restricted. It is not difficult to simultaneously observe the behavior of multiple animals. Numerous existing methods using off-the-shelf parts and open source software already do so (Buchanan et al. 2015 as one of many examples).

We appreciate this helpful comment and apologize for the confusing wording. What we aimed to convey is that, in general, it is helpful to have both a large field-of-view that maintains high resolution when studying freely moving organisms. This is what is unique about the MCAM. While a behavioral arena can be made much larger than a fixed microscope’s field-of-view, a moving organism within the arena will not be seen by a fixed microscope that is “zoomed in” to offer high resolution much of the time. Tracking microscopes are a fantastic solution to this challenge and can follow *one* organism over a large area while maintaining high resolution. But two or more tracking microscopes would be required to track two or more independently moving organisms, which becomes cumbersome, and as far as we are aware has not yet been attempted.

Alternatively, if one “zooms out” to see the entire large arena within the microscope FOV, to track the movement of two or more organisms to observe their interaction will sacrifice image resolution, which limits the ability to capture key morphological details (e.g., eye angle, limb/wing position, etc.). The cited paper [Buchanan et al. 2015] does indeed image multiple organisms, but it does so with this latter strategy, by “zooming out” to view a relatively large area with a single camera at relatively low resolution. Accordingly, it does not attempt to image inter-organism interactions – each organism in the work is in a separate maze and is located within each frame. A large collection of other papers also utilizes “zoomed out” cameras to simultaneously track many zebrafish larvae, *Drosophila*, and a variety of other model organisms within separate wells of a well plate in a parallelized manner – again, not aiming to capture inter-organism interactions. Yet, we agree that methods to switch between “zooming out” and temporally “zooming in” or even reducing resolution to increase acquisition speed have been used for various purposes.

The MCAM addresses the general need to simultaneously resolve small movements or morphological features during such multi-organism interactions (i.e., high FOV and high resolution). To make this goal of the MCAM clearer, we have added the following to the introduction:

“To simultaneously observe many freely moving organisms within a medium petri dish, well plate or an alternative arena that occupies a large FOV, a common strategy is to simply reduce imaging system magnification and resolution (i.e., to “zoom out”) (Buchanan et al., 2015). This is commonly performed in high-content screening experiments in toxicology (Mathias et al., 2012) and pharmacology (Bruni et al., 2014; Rihel et al., 2010), for example, where imaging many organisms is critical to discovery. However, such an approach necessarily must trade off spatial resolution for joint observation and can thus miss certain morphological features and behavioral signatures. “

[Buchanan et al., 2015] S. M. Buchanan et al., “Neuronal control of locomotor handedness in *Drosophila*,” PNAS 112(21), 6700-6705 (2015).

[Mathias et al., 2012] Mathias JR, Saxena MT, Mumm JS. Advances in zebrafish chemical screening technologies. *Future Med Chem*. 2012;4(14):1811-1822.

[Bruni et al., 2014] Bruni G, Lakhani P and Kokel D, “Discovering novel neuroactive drugs through high-throughput behavior-based chemical screening in the zebrafish,” Frontiers in Pharmacology 2014; 5, 153-163.

5. The authors' claim of 147 lp/mm resolution appears to neglect depth of field / defocus issues as animals move axially. According to the lens datasheet, even a 20 lp/mm sample loses substantial contrast as it moves axially over a range of a couple millimeters. This degradation in resolution will be even more severe at 147 lp/mm.

We thank the reviewer for pointing out that we neglected to discuss the system depth-of-field (DOF). In general, there are several ways to define and measure system DOF. To provide a holistic picture of system performance, we have decided to add and describe three unique measures of system DOF to the manuscript.

1) Rayleigh criterion: It is common to present an optical system’s DOF based on the Rayleigh criterion for axial resolution. Under incoherent imaging conditions (applicable to all behavioral and fluorescence imaging experiments reported here), the Rayleigh criterion is defined as λ2 ∗ NA2 , where NA is numerical aperture (see example derivation in Section 3.2 of [Latychevskaia]). For the MCAM, all micro-camera lenses exhibit the same NA for a given focal plane. For the experiments included in this work, this was NA= 0.03, which leads to the approximate “Rayleigh DOF” of λ2 ∗ NA2 = 500 nm2 ∗ 0.032 = 0.28mm for our system. We now include this definition within the Methods section and in Appendix 1.

2) Depth-of-field: We additionally can characterize the DOF with direct experimental measurements, for which there are several options. Following the reviewer’s comment above, we first measure and report the maximum spatial frequency cutoff at a range of depths. To achieve this goal, we acquired experimental images of a planar USAF resolution target at different axial positions by axially translation via a motorized stage in increments of 20 um. Given the system’s full-pitch resolution is 18 µm at the best plane of focus, we define a "useful DOF" as the axial range in which the resolution does not drop by more than a factor of 2 (i.e., remains less than 36 µm = 2 * 18 µm). The following figure demonstrates how the MCAM resolves Group 4 Element 5 (19.68 µm full-pitch line pair spacing) but does not fully resolve Element 6 (17.54 µm full-pitch line pair spacing) on this custom target, leading to our approximation of 18 µm resolution. Then, we found the axial positions where the resulting image resolution drops by a factor of 2, such that it is not possible to resolve Group 3 Element 5 (Appendix 1—figure 2). The total axial range between these two locations leads to a “2X resolution” DOF as 2.54 mm.

3) Contrast based depth of field: A third alternative contrast-based measure of DOF may also be relevant, in which the average image contrast across a range of spatial frequencies is computed and plotted as a function of depth. Specifically, we once again imaged the USAF target, which by definition contains a wide range spatial frequencies with spectral energy centered along the fx and fy axes, at multiple axial locations. For each axial position, we computed a contrast metric based on the mean image gradient magnitude. The full width at half maximum (FWHM) of this global contrast measure averaged over 2 unique camera FOVs (see Appendix 1—figure 2) is 1.95 mm.

[Latychevskaia] T. Latychevskaia, "Lateral and axial resolution criteria in incoherent and coherent optics and holography, near- and far-field regimes," Appl. Opt. 58, 3597-3603 (2019)

Updated text in Appendix 1 Section – MCAM resolution analysis and verification:

“Given the multiple definitions of depth-of-field (DOF) within the imaging community, we here provide 3 unique characterizations of MCAM DOF. First, the Rayleigh criterion for axial resolution under incoherent imaging conditions (applicable to all behavioral and fluorescence imaging experiments reported here) is defined as λ/2NA^2^, where λ and NA is numerical aperture (see example derivation in Section 3.2 of (Latychevskaia, 2019)). Assuming λ=500 nm and NA=0.03 (see above) yields a Rayleigh criterion-defined DOF as D_Ray_ = 0.28 mm. Second, we directly measured the specimen axial range in which MCAM resolution stays within a factor of 2 of its optimal value at the focal plane. Specifically, we determined the axial distance at which the full-pitch resolution deteriorated from 18 µm to 36 µm. This corresponds to a drop in resolution from between USAF target Group 5 Element 5 and 6 to Group 4 Element 5 and 6 in our resolution target. As shown in Appendix 1- figure 2I, resolution decreases to between Group 4 Element 5 and 6 at an axial distance of +/- dz=1.27mm, leading to a 2X resolution drop-defined DOF as D_Res_ = 2.54 mm. Finally, we obtained an image contrast-based measure of DOF, D_Con_, by plotting the normalized image mean image gradient magnitude of USAF target images captured at 20 µm axial increments. The resulting curves in Appendix 1- figure 2I (left) exhibit a FWHM of 1.96 mm and 1.94 mm, respectively, leading us to contrast-based DOF value of D_Con_=1.95 mm.”

Updated text in revised Main text: “System axial resolution defined via the Rayleigh criterion is 0.28 mm, while the axial sample range that supports a 50% reduction in spatial resolution from that measured at the plane of best focus is 2.54 mm (Methods).”

Updated text in revised Methods: “Three quantities are used to characterize MCAM depth-of-field (*D*). First, we use the system’s theoretical Rayleigh criterion for axial resolution under incoherent imaging conditions to estimate *D*_Ray_ = 0.28 mm. Second, we experimentally measured the axial sample range across which lateral resolution remains within a factor of 2 of its maximum value at the plane of best focus as *D*_Res_=2.54 mm, which we found to be a useful measure of practical axial imaging extent. Third, we experimentally measured the full-width half-maximum of image contrast (defined as the average mean image gradient magnitude of a resolution target) as a function of axial distance from the plane of best focus as *D*_Con_=1.95 mm. Data and experimental details regarding these measurements are in Appendix 1: MCAM resolution analysis and verification and Appendix 1- figure 2.”

6. "We performed standard flat-field correction to compensate for vignetting and other brightness variations within and between cameras (Methods)". This needs more detail. What are "other brightness variations"? Give numbers for the amount of brightness variation within and between cameras. This is especially important because inexpensive smartphone sensors assume a tilted chief ray angle (CRA) as a function of image height, which the authors do not discuss. How mismatched is the CRA with respect to the sensor's design assumption, and how much does this impact vignetting?

We thank the reviewer for pointing out that more details were needed to describe our flat-field correction procedure. Below is a response to each of the included questions and requests:

a) By “other brightness variations”, apart from image vignetting effects, we primarily meant the impact of slight non-uniformities in our wide-field incoherent illumination. We have added this note to the text:

“We performed standard flat-field correction to compensate for lens vignetting and illumination brightness variations within and between cameras…”

b)To provide numbers for the amount of brightness variation within and between cameras, we performed an experiment in which we imaged a uniform diffuser illuminated in a trans-configuration with our standard wide-area illumination unit (used for most of the included behavioral imaging experiments). The mean intensity for each camera, as well as the standard deviation of pixel values within each camera, are displayed below. For inter-camera variation, we now report in the manuscript the mean pixel value for all cameras as (120) and standard deviation of the means across all cameras as (10). For intra-camera variation, we now report the average of the standard deviation of the raw pixel values of each sensor (6).

Appendix1—figure 2F. Plots of inter- and intra-camera brightness variation for a uniformly diffuse specimen. (left) Mean of raw pixel values for each camera within the 8x12 array, and standard deviation across all per-sensor raw pixel values for each sensor within the 8x12 array.

In addition, for 4 randomly selected cameras, we plot the normalized mean value of each column. Slight vignetting effects are clear (note y-axis for mean intensity drops to 94%). Moreover, the vignetting profile for camera slightly shifts, suggesting its primary cause is our wide-field illumination, which is not fully diffuse at the sample plane.

Appendix1—figure 2G. Plot of average normalized pixel value across the vertical camera dimension for 4 randomly selected micro-cameras when imaging a fully diffuse target. Relative brightness varies <6% across the array. Primary contributions arise from illumination, lens, and sensor-induced vignetting effects. As described in Appendix 1– Image stitching below, digital flat-field correction is used for correction.

c)In terms of CRA, we are not aware of any physical mechanisms within the architectural design of the bare CMOS sensors employed by the MCAM that attempt to account for a deviation of CRA. Instead, we would expect such CRA effects to be accounted for within software post-processing of the smartphone imagery. As we have access to the raw pixel data from the employed CMOS sensors, we do not expect any such effects to impact our measurements. In either case, the trace plots above provide some experimental insight into their impact. If there were an impact on a mismatch between assumed sensor CRA and our measurements, we would expect it to be symmetric with respect to the optical axis (i.e., the center of the sensor). The asymmetry of the plots above thus suggest its impact is minimal, and instead other causes of vignetting are more important.

We now include the above data in Appendix 1 Section 2, with an updated Appendix 1- figure 2 and associated caption, as well as a brief description that summarizes our experimental findings:

“Finally, as noted in the main text, we employed a standard flat-field correction procedure to account for brightness variations both within and across the employed CMOS sensors before proceeding to image stitching. We hypothesize that brightness variations were primarily caused by our employed large-area bright-field illumination source, which was slightly non-uniform and not fully diffuse at the specimen plane, as well as lens vignetting and possible sensor vignetting effects. To explore the properties of these brightness variations in more detail, we experimentally imaged a large diffuser with the MCAM, under illumination from our standard trans-illumination source. We captured and averaged a series of 3 images while slightly moving the diffuser to unique locations to produce an average “white” image. The resulting mean of the raw pixel values (0-255) of this white image are plotted in Appendix 1- figure 2F (left). The inter-sensor standard deviation of the raw pixel values is similarly reported in Appendix 1- figure 2F (right). The spatial profile of per-sensor brightness variations is assessed in Appendix 1- figure 2G. From this data, we measured the standard deviation of the mean pixel value across all 96 image sensors to be 10, where the mean of all pixel values was 120. Similarly, via our measurements of intra-camera brightness variation, we found the average of the standard deviation of the raw pixel values of each sensor to be 6.”

7. "Using this stereoscopic depth measurement approach yields approximately 100 μm accuracy in depth localization over a >5 mm depth range". How was this 100 μm accuracy number determined?

We thank the reviewer for raising this point. Details of our experimental assessment of depth accuracy are provided in Appendix 1 – Depth tracking as well as Appendix 1- figure 7. We now note the location of these details in the main text with this addition:

“…(see Appendix 1- figure 7 and associated Appendix 1 for experimental support).”

Here is a short summary. Once we established our depth detection pipeline, we constructed an experiment to validate the MCAM's depth detection accuracy and range: As reported in l in Appendix 1 – Depth tracking, we placed a standard 1951 USAF resolution test chart on a motorized stage (accuracy of 2 um) within the imaging area of the microscope. For validation, the stage was moved between 50 positions, each 100 μm apart, across a total depth range of 5 mm. At each position, a set of images was captured (one from each of the two utilized micro-cameras). Three bars within the resolution target were used for centroid detection. The relative position of each centroid is compared across adjacent cameras to estimate depth. Please see Author response image 1, as well as its residual. The average normalized mean-squared error between measured and expected depth was found to be approximately 25 µm. Due to the use of large axial step sizes (100 µm) within this experiment, we decided to conservatively report the accuracy of our depth localization as approximately 100 µm.

**Author response image 1. sa2fig1:** 

8. There is a troubling disregard to axial extent. In one instance, there is a reference to 5 mm height. In the case of zebrafish fluorescence imaging, the height is stated as 1-2 mm. Axial defocus should be characterized, and some statement of the practical usable axial imaging extent should be provided.

We thank the reviewer for pointing out this issue. Please see additional details regarding this question in our response to Question 5 above. In summary, we now provide three different measures of the impact of axial defocus on our system:

1) The theoretical Rayleigh axial resolution of the MCAM system, which is often used as a measure of sensitivity to axial defocus, as *D_Ray_* = 0.28 mm

2) We measured the axial range that supports a 50% reduction in spatial resolution from that measured at the plane of best focus as *D_Res_* = 2.54 mm

3) We measured the FWHM of the curve defining image contrast as a function of axial location near the in-focus sample plane as *D_Res_* = 2.54 mm. Contrast here is defined as the mean image gradient magnitude of the image of a resolution target.

We have added the following text to the Methods section to note the practical usable axial imaging extent: “…we experimentally measured the axial sample range across which lateral resolution remains within a factor of 2 of its maximum value at the plane of best focus as D_Res_=2.54 mm, which we found to be a useful measure of practical axial imaging extent.”

9. With regard to the siamese network / melanophore patterns, are the embeddings stable over recording time, animal pose, etc? There is insufficient detail to evaluate this. With regard to "250 augmented images per fish", how many real images of the animal were used, before augmentation? Is Figure 2c based on training or test? The number of real images used for training and test appears much too low to address questions of stability over time and animal pose, for example. Why is there no quantitative comparison of performance of training vs test?

We thank the reviewer for requesting the above clarifications. First, our Siamese network demonstration was intended to show that accurate embedding of MCAM data is possible and is relatively direct to achieve with existing post-processing tools and a limited amount of data annotation. While we did not perform experiments to assess stability over recording time and animal pose, the train and test data was randomly selected from frames acquired during several minutes (i.e., several hundred unique video frames) with completely unconstrained zebrafish larvae who were free to change pose, and did all change pose at least slightly, during that time period.

Second, we derived our augmented images from 50 original MCAM recording frames that were cropped around the coordinates of nine fish, meaning 50 real cropped images per animal were used before augmentation.

Third, Figure 2c is based on test data, as we believe is the more relevant and standard performance demonstration for supervised learning networks.

Fourth, we agree with the reviewer that our demonstration does not address stability issues across long periods of time and pose. Zebrafish larvae are capably of a wide variety of poses, so a significant amount of experimental effort is required to demonstrate true robustness across animal and pose and long periods of time, via capturing and annotating a substantial amount of data. We are currently endeavoring to carefully demonstrate accurate long-term organism tracking across a variety of conditions in ongoing experimental work. We have also made all of our captured data from this work open-source, to encourage others to explore this interesting research area. By demonstrating the Siamese network with just 50 images per organism, our goal was actually quite the opposite of addressing stability issues, and was instead included to show that unique organism identification is at all possible with such a small number of images. We hypothesize that this unique capability is facilitated by the MCAM’s significantly higher resolution as compared to alternative imaging systems used for organism tracking (i.e., the MCAM is capable of resolving individual melanophores, while standard cameras over a similar FOV are not). We are currently working to carefully test this hypothesis.

Last, we include a quantitative measure of performance for training (validation) in Supplemental Figure 6c, which can be compared to the test error of effectively 0 for our limited dataset size that is included in Figure 2c.

Fluorescence imagingThere are some deeply concerning issues with the design that are not discussed in the manuscript. The manuscript seems to neglect the fact that existing microscopes have axial resolution, in addition to lateral resolution. Setting aside the necessity for more accurate characterization, there are some fundamental issues that are inherent to the imaging approach.

First, we thank the reviewer for their time and provision of this detailed list of comments and suggestions regarding the MCAM’s fluorescence video data, despite the many limitations. Following the recommendations of the editor and the steps laid out in the *Revision Action Plan* at the beginning of this document, we have agreed to follow the reviewer’s and editor’s suggestions to remove the fluorescence imaging data from the main text, and instead primarily focus the manuscript on the former bright-field imaging aspect of the MCAM imaging system. We have shifted the highly modified discussion of fluorescence to Appendix 1.

The reviewer also raises a very helpful point here regarding lateral and axial resolution. The MCAM design only allows acquisition of *wide-field* fluorescence image data. As a wide-field fluorescence microscope, the system’s lateral and axial optical resolution are tied directly to the system numerical aperture. As noted in response to Reviewer 1, Question 5 and Reviewer 1, Question 8 above, we now provide 3 measures of axial resolution within the manuscript, as well as a direct measure via an acquired 3D point-spread function (see response to Reviewer 1, Question 18).

Fundamental problems with the design:1. Without any axial sectioning capability, the fluorescent signal can be corrupted by the position and orientation of the animal. As the animal moves across the field of view of each microcamera, the detection PSF (point spread function) tilts in a position-dependent way, such that a different set of signals across the axial extent of the thick tissue are projected onto the same sensor pixel. Similarly, each time the pose of the animal changes, each pixel now is the summation of signals from a fundamentally different set of cells and even regions across the brain.

We thank the reviewer for raising this concern. First addressing the concern about animal position, as the system uses well-corrected micro-vision camera lenses with a somewhat large focal length (25 mm) that are designed to form a relatively shift-invariant point-spread function across the image plane. Accordingly, we do not expect the lateral position shift of an object of interest to significantly impact the MCAM’s acquired optical signal. We verified this shift-invariance with, for example, the resolution target characterization in Appendix 1- figure 2D-E, where we characterize the resolution target at both the center and edge of a single micro-camera FOV to show effectively no drop in associated image resolution (i.e., minimal introduction of aberrations). As the resolution target in this example was illuminated with fully incoherent light, we do not expect this optical characterization of shift invariance to significantly deviate from the case of imaging a fluorescence emitter with a narrowband emission filter. This analysis does not of course consider the shape of the PSF at planes away from the object plane-of-best-focus, where it could certainly deviate from the center to the edge of one micro-camera’s field-of-view.

Second, concerning organism orientation, the reviewer here raises an excellent point. Due to an elongated PSF, the effects of absorption, as well as scattering, the acquired per-pixel signal will vary as a function of organism rotation. In our fluorescence measurements, we provided an average measure of regional fluorescence fluctuation across the entire zebrafish larvae brain (approximately several hundred pixels). By averaging acquired fluorescent signals across many pixels, we expect such corruptive effects to diminish. However, we did not perform a careful assessment of this assumption. Accordingly, we have removed all fluorescent imaging data from the manuscript. To remove any possible connections between the MCAM’s wide-field fluorescence imaging and the acquisition of fluorescent “signals”, we have removed all post-processing analysis of acquired fluorescent image data from both the manuscript and Appendix 1.

2. Ratiometric imaging cannot be done when the green and red fluorescence comes from different cells! This combined with point #1 above, means that every time the animal changes its pose, a given red fluorescent pixel is potentially summing photons from a different set of cells and brain regions than the "matching" green fluorescent pixel. This is fundamentally not sound. (The dual-color images in Figure 3h show obvious labelling differences, as expected from the use of different promoters and cell types in the two color channels.)

We thank the reviewer for raising this astute observation. Indeed, the green and red fluorescence can come from different cells within our recordings, which can certainly impact the resulting ratiometric analysis. In short, our dual-color imaging approach was not designed to provide two unique spectral measurements from exactly the same scene point (i.e., pixel). In our presentation of pan-neuronal signal measurement, we integrated all acquired ratiometric values across the entire larval brain (approximately 200 pixels, each a projection of many fluorescence emitters along the axial dimension). Accordingly, our final signal was a spatial integration of green fluorescence signal and red fluorescence signal from many individual cells (potentially thousands). In other words, the red and green signal measured from every image is approximately a summation of the signal over the same large number of cells. While we expect this average ratiometric signal to remain relatively robust to organism movement, we have decided to remove our ratiometric imaging demonstration from the manuscript. We simply show the ability to acquire (but not directly compare) dual-channel fluorescence measurements in Appendix 1. We will attempt to provide a more careful experimental presentation in future work.

3. Related to #2, the red and green fluorescent light is collected from different cameras. This means that due to point #1, even if the same promoter was used for the green and red channels (as it should), each sensor pixel of the green image is sampling a different projection through the thick tissue compared to the corresponding sensor pixel of the red image. Applying cross-channel normalization of thick fluorescent tissue using this imaging geometry is fundamentally unsound.

We thank the reviewer for raising this astute observation. Indeed, the depth and angle of the animal, as well as absorption and scattering from locations above the emitters, can impact the MCAM’s acquisition of an accurate localized fluorescence activity signal. As noted above, in our presentation of pan-neuronal signal measurement, we integrated all acquired ratiometric values across the entire larval brain (many dozens of pixels, each a projection of many fluorescence emitters along the axial dimension), such that the red and green signal used for ratiometric analysis from every image is approximately a summation of signal over the same large number of cells. While we expect this average ratiometric signal to remain relatively robust to organism movement, we have removed our ratiometric imaging demonstration from the manuscript.

4. Both lateral and axial resolution are depth dependent. Without axial sectioning capability, as the animals moves up and down in z, each point in the sample (e.g. a cell) contributes a different angular distributions of photons to a given pixel on the camera. To put this simply, a pixel on the camera is sampling a different distribution of signals depending on the axial position of the animal. This cannot be solved by ratiometric imaging. The information contained within each pixel is fundamentally changing as a function of depth. There is a reason that standard microscopy techniques go to great length to adjust focus as a function of depth.

We agree and thank the reviewer for pointing out that system resolution is a function of depth. In our fluorescence imaging experiments with freely moving zebrafish larvae, we indirectly constrained axial movement by using a shallow dish of water – approximately 1-2 mm, which is slightly larger than the measured imaging depth-of-field. Depth-dependent variations of organisms within images thus primarily manifested as lateral disparity shifts, which were accounted for by aligning ratiometric image pairs via the software steps outlined in Appendix 1 – Depth tracking section. That said, to remove any confusion within the manuscript, we have removed our ratiometric imaging demonstration from the manuscript.

We now explicitly mention the loss of resolution and the challenges with accurate fluorescence signal acquisition without axial sectioning in the associated Appendix 1 section (Fluorescence imaging with the MCAM):

“The current MCAM setup measures wide-field fluorescence, and thus integrates fluorescent signal from across its full depth-of-field. While this capability can potentially be useful for non-localized (i.e., low-resolution) fluorescence monitoring across many freely swimming larvae, effects caused by organism axial movement and tilt must be accounted for during such an analysis.”

5. The authors do not provide standard lateral and axial point spread functions (PSF) (i.e. using a submicron fluorescent bead), or demonstrate how both lateral and axial PSF change across x, y, and z. This is a standard optical characterization for a fluorescent imaging system.

We agree that a PSF measurement is a useful way to characterize optical performance. We have captured fluorescent PSFs with the MCAM both at the center and edge of a single micro-camera FOV (see plots below). Specifically, we imaged 6 µm microspheres placed on a microscope slide that was positioned atop a mechanical z-stage, which allowed us to rapidly capture images at different axial locations. We moved the stage in 100 µm increments across a total travel range of 7.7 mm. Cross sections of the raw pixel measurements of the PSFs are shown in Appendix 1- figure 2 with an associated description.

Appendix 1 Section 2: “To provide additional data regarding lateral and axial resolution, we measured the 3D fluorescent point-spread function (PSF) response of a randomly selected micro-camera. Specifically, we outfitted the MCAM for wide-field fluorescence (see Appendix 1 – Fluorescence imaging with the MCAM) and acquired images of a monolayer of 6 µm fluorescent microspheres (Fluoresbrite YG Microspheres CATALOG NUMBER: 17156-2, Emission max. = 486 nm) axially translated by 100 µm across 7.7 mm. The similarity of PSFs at the center and edge of the FOV show a minimal impact of lens aberrations.”

6. Illumination brightness is not expected to be homogenous across x y and z. There is no discussion of brightness correction for either excitation or collection light across 3 dimensions. For characterization, the authors would need to show the results of a thin (submicron) fluorescent sample across the entire field of view, and across z.

We apologize for the insufficient detail regarding the brightness and homogeneity of the employed excitation illumination. Our widefield fluorescence setup has uniform intensity along the axial direction, as we employed large, relatively incoherent LEDs to illuminate the entire sample across a wide area from above (see Appendix 1- figure 8A for geometry). While the suggestion of an experiment with a thin (submicron) fluorescent sample across the entire FOV is an excellent one, we instead opted for a direct measure of excitation illumination brightness. To assess two-dimensional lateral excitation brightness uniformity, we measured the brightness profile of the excitation illumination by placing a uniform diffuser at the image plane and removed the emission filters from all micro-cameras. Intra-camera brightness variations, primarily due to vignetting, followed nearly the exact same behavior as shown in response to Reviewer 1, Question 6 above (see updated Appendix 1- figure 2 G). Cameras exhibited <6% variation across the FOV. A summary of inter-camera brightness variation is shown in the plots below. The mean across all 24 cameras in the MCAM array (6x4) varied <5% (see normalized standard deviations in Author response image 2).

We now report this excitation uniformity quantification with the following text in Appendix 1 – Fluorescence imaging with the MCAM section:“Lateral excitation illumination uniformity was experimentally assessed by measuring the intensity response of the MCAM to a uniform diffuser at the specimen plane, which yielded <6% intra-camera intensity variation (primarily caused by vignetting) and <5% inter-camera mean intensity variation.”

7. Need a detailed analysis of signals from the red fluorescent protein as control. In the ideal case, the measurement from the red channel should of course be flat, but reporting the reality is important to establish the effective noise floor of the technique. The most problematic is systematic noise – intensity changes related to movement, lateral or axial position, animal pose or brain angle, all of which can lead to spurious correlations that complicate the use and interpretation of the fluorescence data in quantitative studies of animal behavior. Claiming detection of movement-related GCaMP activity is not that meaningful, as numerous studies have recently demonstrated that movement causes widespread activation throughout the brain.

We agree with the reviewer regarding the various sources of noise that can impact any imaging and specifically wide-field fluorescence measurements. Following the revision action plan, we have removed these results from the main text, as carefully and adequately addressing these concerns would go beyond the scope of the revised manuscript. In addition, we have removed the claim that signal from red fluorescent proteins (RFP) can be used as a precise control measurement for ratiometric analysis. Instead, we simply report that it is possible to acquire such a signal. We additionally note the various sources of possible noise that can impact wide-field fluorescence image capture, as suggested by the reviewer, in Appendix 1 (see updated Appendix 1 Section – Fluorescence imaging with the MCAM and updated Appendix 1- figure 8).

“The current MCAM setup measures wide-field fluorescence, and thus integrates fluorescent signal from across its full depth-of-field. While this capability can potentially be useful for non-localized (i.e., low-resolution) fluorescence monitoring across many freely swimming larvae, effects caused by organism axial movement and tilt must be accounted for during such an analysis. Other sources of noise include slightly non-uniform excitation illumination, lens vignetting, animal pose and brain angle, and a low imaging numerical aperture. Additionally, the use of Bayer-patterned image sensors further limited the fluorescence sensitivity of these initial experimental tests. In the reported experiments, we employed constant wide-field excitation throughout the video recording, which can potentially lead to photobleaching effects. Our longer video acquisitions (e.g., of the freely moving zebrafish) lasted 250 seconds, within which we did not observe significant photobleaching effects. A strobed excitation source could help to alleviate photobleaching issues. We further note that the MCAM presented here collects 8-bit image data. Due to inhomogeneities in sample fluorescence intensity and illumination, 8-bit image data can lead to severe dynamic range and quantization noise effects, as compared to the use of 12-bit or 16-bit image data acquisition. We aim to address many of these shortcomings with updated hardware and software for MCAM fluorescence imaging in future research efforts.”

8. The system is designed for visualizing inter-animal interactions, but there is no fluorescent data obtained during inter-animal interactions. When animals come into contact with one another, or overlap on z, a given pixel on the camera may become the summation of photons from both animals, further corrupting the neural signal.

We have removed the notion of examining inter-animal interactions with fluorescence from within this work. However, we generally agree that this would be an ambitious research goal. As this will go beyond the scope of this revised manuscript, we hope to address this in future work.

9. If the system is to be evaluated as a microscope, the authors should provide a bleaching curve. Is the blue light on all the time, or is it strobed? How long can a user realistically image with this system?

We apologize to the reviewer for this lack of information in the original manuscript. Over the short imaging periods, we did not detect major bleaching over the five minutes of our recording. It is reasonable to assume that recordings will be similarly stable as other wide field recordings in freely swimming zebrafish GCAMP imaging experiments (Muto et al. 2013). Therefore, we can expect to be realistically image fluorescence from a few minutes up to an hour without major bleaching.

Yet, as noted above, we have removed all fluorescence imaging data from the main text, and instead include a much more limited set of demonstrations within the Appendix 1. There, we have added some additional information regarding the use of our excitation source and we make clear how long our imaging experiments lasted with the following new text in Appendix 1 – Fluorescence imaging with the MCAM:

“In the reported experiments, we employed constant wide-field excitation throughout the video recording, which can potentially lead to photobleaching effects. Our longer video acquisitions (e.g., of the freely moving zebrafish) lasted 250 seconds, within which we did not observe significant photobleaching effects. A strobed excitation source could help to alleviate photobleaching issues.”

Muto, Akira, Masamichi Ohkura, Gembu Abe, Junichi Nakai, and Koichi Kawakami. 2013. “Real-Time Visualization of Neuronal Activity during Perception.” Current Biology: CB 23 (4): 307–11. https://doi.org/10.1016/j.cub.2012.12.040.

10. There is no discussion on how a user is supposed to calibrate this system in 3D for fluorescent imaging. Though it's unclear to what extent calibration can alleviate the core conceptual issues embedded in the design.

We thank the reviewer for pointing out this lack of detail. We did not mean to imply that 3D fluorescence imaging was possible within this manuscript. 3D and fluorescence acquisition are effectively two separate imaging modalities. To avoid any confusion, we have removed the fluorescence imaging results from the main text of the manuscript, and now include a limited subset of results within the Appendix 1. In addition, we now include the following text to clearly state the limitations of wide-field fluorescence imaging of moving organisms:

“Similar to the case of bright-field imaging, fluorescence acquisition of signal from moving organisms also suffers from axial blurring. Effective results were collected when organism axial movement was constrained by using a relatively shallow water depth (typically several mm).”

11. The 2x2 Bayer filter is problematic, particularly for the fluorescence imaging case, where the effective number of imaging pixels isn't a gigapixel, or even a quarter of that, but in fact an 1/8th (green) or a 1/16th (red) of that. In the red channel, the effective pixel pitch appears to be around 18 um, which means the claimed 18 μm two-point separation is demonstrably false. For example, consider two (red fluorescent) point sources located at the midpoints of the 18 μm effective pixel pitch, spaced apart by 18 um. These will be detected as one bright red pixel flanked by two half-intensity red pixels, with no separation whatsoever. In the green channel, the effective pitch on one diagonal axis is twice the effective pitch of the other diagonal axis, and the same resolution argument holds.

We thank the reviewer for pointing out that the Bayer filters on our first MCAM’s sensor do indeed impact the resolution of the resulting fluorescence image. We have added to our original explanation on the negative impact of the Bayer filters to detection resolution and sensitivity, to explicitly state their impact to fluorescence imaging resolution, in Appendix 1 – MCAM resolution analysis and verification (new text in red):

“The inclusion of a Bayer filter over each CMOS pixel array further limits detector resolution when utilized to produce snapshot color images. Furthermore, the Bayer filters also reduce sensor sensitivity in general. Specifically, when used for fluorescence imaging, the Bayer filter pattern leads to two problematic effects. First, it leads to reduced resolution per fluorescence channel image (below the 18 µm pixel resolution for white light). Second, it leads to reduced sensitivity. For applications that could benefit from higher spatial resolution and higher fluorescence sensitivity, future designs will use unfiltered monochrome pixels”

And in Appendix 1 – Fluorescence imaging with the MCAM: “Additionally, the use of Bayer-patterned image sensors further limited the fluorescence sensitivity of these initial experimental tests.”

12. The system that has been presented collects only 8 bit image data. Typical bit depths in microscopy systems range from 12 bit to 16 bit. The restriction to 8 bit is particularly problematic because due to inhomogeneities in fluorescence intensity of the sample and intensity of the illumination, combined with the desire to avoid saturated pixels which prevent extraction of quantitative signals, with 8 bit data one ends up with a severely limited dynamic range and severe quantization noise. The authors make no mention of the unconventional use of 8 bit data for fluorescence imaging, nor the associated caveats.

We thank the reviewer for pointing out this concern regarding 8-bit data acquisition for fluorescence imaging. We now explicitly state the limitations encountered when using 8-bit image sensor for fluorescence imaging in the associated section in Appendix 1 – Fluorescence imaging with the MCAM:

“We further note that the MCAM presented here collects 8-bit image data. Due to inhomogeneities in sample fluorescence intensity and illumination, 8-bit image data can lead to severe dynamic range and quantization noise effects, as compared to the use of 12-bit or 16-bit image data acquisition.”

Claims that are not sufficiently substantiated or slightly misleading:"To reduce motion artifacts, we started by embedding zebrafish in low melting point agarose and recorded spontaneous neural activity for 5 min". This should be removed from the paper, as it avoids all of the problems introduced by the MCAM imaging approach, most importantly brightness and focus changes over the FOV of each microcamera and over the 1-2 mm bath depth, as well as animal position-dependent and pose-dependent axial projection of thick brain tissue onto the 2D image sensor.

We thank the reviewer for this helpful suggestion. We have removed the embedded zebrafish data from the fluorescence imaging section within the Appendix 1.

The legend text greatly overstates: "Average ∆F/F traces of these ROIs show that differences in neural activity are easily resolved by the MCAM." To even partially address the many challenges posed by a freely behaving MCAM calcium imaging experiment, the authors should suspend the embedded animal from a motorized stage (e.g. a glass slide overhanging a typical lab XYZ translation stage) and then capture images of the brain over the ~ 38 mm x 19 mm field of view (including the corners of the rectangular field of view, which represent the most aberrated condition of the imaging lens), as well as the 1-2 mm depth axis that freely behaving animals move in. Note, I estimated the field of view of each microcamera to be 38 mm x 19 mm based on the sensor pitch of 19 mm and the 2:1 overlap strategy, but the authors should clearly state the actual numbers in the methods.

Thank you for pointing the various issues surrounding our calcium activity measurements. Following the reviewer and editor guidance and the Revision Action Plan, we have removed this portion of the legend and all associated results. We point the reviewer to our PSF measurements (see R1Q18) for a measure of the impact of aberrations on acquired fluorescence. We also now provide the exact FOV of the micro-cameras within Appendix 1 Section 2: “The approximate FOV of a single micro-camera within the array is 39 mm x 19.5 mm”.

" While the MCAM in the current configuration cannot detect single-neuron fluorescence activity in the zebrafish brain, the data do show that our multi-sensor architecture can measure regionally localized functional brain activity data in freely moving fish". The data do not demonstrate sound optical principles and analysis. It's not just a matter of sensor resolution – the complete lack of axial sectioning/resolution is a fundamental problem. The wording appears to imply that MCAM eventually could, but this is impossible to evaluate.

Thank you for pointing out the issues surrounding our fluorescence image capture. As noted above, we have removed the fluorescence imaging results from the main text. We have also removed our processing of neural activity data from the main text and Appendix 1 and have removed the sentence noted in this comment.

Calibrated videos of freely swimming transgenic zebrafish with pan-neuronal GCaMP6s expression verify that our system can non-invasively measure neural activity in >10 organisms simultaneously during natural interactions. The word "calibrated" is unclear. What precisely was done? If the authors simply mean "flat-field corrected", then say that, or else add a description of this "calibration" in the methods and refer to it here.

Thank you for pointing out this unclear language. We have removed the term “calibrated” from the associated text within the Appendix 1, as no additional calibration was performed for the cited experiments.

"…the latter can increase both the information content and accuracy of quantitative fluorescence measurement in moving specimens". Due to the differences in view angle dependent axial projection of thick tissue onto a 2D sensor, angle-dependent sample occlusion (e.g. overlying pigment or blood cells), and other above-mentioned uncorrected differences in imaging performance across a large 3D imaging area, the current approach is fundamentally unsound. Claiming that it suitable for accurate quantitation is indefensible.

Thank you for pointing out the issues surrounding this wording. To avoid any confusion, we have changed the associated text to the following:

“We have also demonstrated two unique capabilities of the MCAM’s multi-lens architecture – 3D tracking and simultaneous acquisition of fluorescence images from two unique spectral bands (see example in Appendix 1- figure 8). The former functionality opens up a new dimension to behavioral analysis, while the latter could lead to novel fluorescence imaging experiments with additional development and refinement.”

"…there are likely a wide variety of alternative exciting functionalities that MCAM overlapped sampling can facilitate". As an existence proof, please give one or more examples.

Thank you for raising this clarifier – the associated sentence now includes an example a new reference and reads as follows:

"There are likely a wide variety of alternative exciting functionalities that MCAM overlapped sampling can facilitate, such as dense height map estimation at high resolution over large areas via photogrammetric methods (Zhou et al., 2021)."

[Zhou et al., 2021] K. C. Zhou et al., “Mesoscopic Photogrammetry With an Unstabilized Phone Camera,” Proceedings of the IEEE/CVF Conference on Computer Vision and Pattern Recognition (CVPR), 7535-7545 (2021).

"i.e., the ability to resolve two 9 μm bars spaced 9 μm apart": This is misleading as stated. This should be described as "a center to center spacing of 18 um".

Thank you very much for the suggested clarification, we now state:

"…a center to center spacing of 18 μm".

"…we used custom Python code to produce gigapixel videos from the image data streamed directly to four separate hard drives". A solid state drive is not a "hard drive". Just say "solid state drives" or "rapid storage drives".

Thank you for the suggestion, we replaced ‘hard drives’ with “solid state drives” in the revised manuscript.

Reviewer #2 (Recommendations for the authors):This manuscript provides a useful tool for mesoscale imaging in complex biological systems. Simultaneous imaging at a large field-of-view and high resolution has been a challenge for biological studies, resulting from the optical aberration and data throughput of sensors. In this work, Thomson et al. proposed a flexible framework that enables multi-channel parallel imaging of 96 cameras from a field of view up to 20 cm*20cm and at a resolution of 18 μm. They have demonstrated various interesting experiments, which only be achieved with these mesoscale imaging and may arouse great interest.Strengths:This paper remains a great engineering effort. In order to achieve a large field view and high resolution, the authors build the 0.96 gigapixel camera array with 8*12 camera sensors, with 16 FPGA routing the image data to the computer. Two different imaging modes are developed: a full-frame mode with 96 cameras at 1 HZ and a single sensor mode with 4 cameras at 12 Hz, corresponding to fast and slow applications.Also, I like the idea of stereo-imaging. Spatial overlapping is inevitable in image stitching, and most previous works tried to reduce the overlapping area to increase the data throughput and to reduce the inhomogeneous image. In this manuscript, all the imaging area is imaged simultaneously by at least two cameras, so the whole image is homogenous and the extra multi-view information is utilized for axial localization.The authors have verified their system parameters in a series of biological experiments, such as zebrafish, *C. elegans*, and Black carpenter ants, which indicates its wide scope of applications. I believe all these experiments are quite interesting to a broad audience, e.g. social behavior for sociologists. Overall, I think the MCAM configuration is a promising initial scaffold for parallel image acquisition.

We thank the reviewer for their positive feedback and are happy to hear that the reviewer appreciates the great engineering effort involved in this work, likes our idea of stereo-imaging, and thinks the MCAM is a promising approach.

Weaknesses:1. Although the paper does have strengths in principle, the weakness of the paper is that the strength of this system is not well demonstrated. The fundamental limitation of this system is the imaging rate is far too slow for the dynamic recording of various demonstrated applications, such as zebrafish's collective behaviors. The authors have realized this drawback and discussed it in the manuscript, but they do not give a practical estimation of how fast it could be, with up-to-date technologies. Is it possible to improve the speed by using different computers simultaneously for distributed data acquisition, as in Fan et al. Nature Photonics 13:809-816, 2019.?

We very much appreciate this insightful comment. We agree that the system’s low frame rate limitation is not a fundamental one but is instead a practical limit that was encountered during the technical development of our prototype. The MCAM’s 1 Hz imaging rate corresponds to a total data rate of 960 MB/second. We encountered this 960 MB/sec limit at the step of USB data transfer – the image sensors can capture at significantly higher total data rates, and the electronics from the sensors to the data transfer line can operate at much faster data rates.

Since the development of our first prototype, USB data transfer rates have significantly improved: revisions of the USB protocol now allow up to 2500 MB/sec (2.5 GB/sec) transfer on a single link [1]. With our prototype’s 4X USB link approach, this would allow us to increase the MCAM’s total data rate to 10,000 MB/sec (10 GB/sec, saving approximately 10.4X more image data per sec). Accordingly, using an updated USB protocol, we would be able to capture at approximately 10.4 frames/sec on a single desktop computer without significantly increasing the complexity or cost of the system. This, of course, would necessitate a more serious architectural effort. As the reviewer suggests, additional speed-up could also be achieved by saving data to multiple desktop computers in parallel, with a slight increase in system cost and complexity. In fact, we have been working to develop an updated MCAM prototype that approaches these data rates. Finally, if one wishes to free themselves from the limitations of USB, modern GPUs and networking equipment can communicate with a processor at data transfer rates of nearly 32GB/sec. We have added the following sentences to the Discussion to elaborate on this point:

“…we note here that a number of strategies are available to increase sampling rates. The most straightforward paths to directly increasing the MCAM frame rate are (1) utilizing an updated version of USB that now allows approximately 10X faster data transfer, (2) distributing data transmission across multiple desktop computers, (3) adopting an alternative FPGA data transfer scheme (e.g., replacing USB data transmission with a faster PCIe protocol), and (4) executing on-chip pixel binning, image cropping, and/or lossless compression to increase temporal sampling rates at the expense of spatial sampling. Various combinations of the above strategies are currently being explored to facilitate true video-rate (24-30 Hz) MCAM imaging and to achieve significantly higher frame rates and improve cross-camera snapshot synchronization.”

[1] SuperSpeed USB, USB 3.2 Specification. Available from: https://www.usb.org/superspeed-usb (Accessed July 21, 2022)

2. Another concern is the asynchronized trigger mode for different cameras. When the exposure time of each image is short to avoid image blur, it is likely that all images are taken at different time points. Is it possible to use an external trigger to synchronize all cameras, or is it possible to output timestamp for each image, so that the error of time difference could be estimated?

We thank the reviewer for raising this excellent point. In our first prototype, triggering was not possible due to the architecture of the utilized FPGAs. However, it is possible to output a timestamp for each image, and that is a very helpful suggestion. Since writing this first manuscript, we have updated our MCAM prototype to utilize a different FPGA design. In this updated MCAM prototype, all sensors now capture images at effectively the same time (up to a 6 µs timing offset). This partially alleviates the need for a timestamp. The update prototype’s FPGA’s are then tasked with transmitting the rapidly captured image data to the computers at the data rate that the transfer link can sustain. We have added a sentence discussing this concern and our work towards addressing it:

Discussion:

“Various combinations of the above strategies are currently being explored to facilitate true video-rate (24-30 Hz) MCAM imaging and to achieve significantly higher frame rates and improve cross-camera snapshot synchronization.”

Appendix 1 Section – MCAM Data Management:

“Such asynchronous image capture is a limitation of the electronics layout of our first MCAM design. It can be addressed by several enhancements, including the potential addition of an external trigger, the output a precise timestamp for each frame, and/or a shift to an updated FPGA architecture that enables synchronous frame capture and collection.”

3. In abstract and supplementary figure 2, the authors claim their system has 5 μm sensitivity, but this is confusing. The fact that this system can capture light from 5 μm beads, but not from 3 μm beads, is that the signal from 3 μm beads is not strong enough to distinguish itself from background noise. For example, if the illumination is strong enough, it is possible for microscopic systems to image small nanoparticles, but this does not indicate those systems have nanometer sensitivity. I understand that light collection efficiency is a serious concern for such low NA microscopic systems, but they should compare the light efficiency in a more persuasive way.

We thank the reviewer for raising this concern about our claim of sensitivity, and we have removed it from Appendix 1- figure 2 and the main text, to avoid any confusion. Instead, we directly report the two-point lateral resolution, as well as axial resolution when it is needed, which are more standard and typically preferred resolution metrics.

4. Another concern is the dual-color imaging. The fact that this system enables stereo-imaging seems to contradict dual-color imaging for 3d objects. Imagine there are two light sources at different depths, they could overlap on one camera but separate on another. I think more proper demonstrations are 2D samples, such as a disk of dual-color cells or worms.

This is another excellent point that we neglected to directly address in the manuscript draft, as also raised by Reviewer 1. In short, our dual-color imaging approach was not designed to provide two unique spectral measurements from exactly the same scene point (i.e., pixel). If the specimens of interest are 2D, then this could be possible via careful registration of acquired green and red image data. However, as the reviewer correctly asserts, such registration is very challenging for 3D objects (i.e., the two points would need to be appropriately registered in the two cameras, yet their depth is unknown). We did not attempt this in our demonstrations, and instead integrated the signal for each color channel across many (200+) pixels that contained the larval zebrafish brain before using a ratiometric analysis. To avoid any confusion, we have removed the dual-color imaging description and results from the manuscript. We will attempt to provide a more careful experimental presentation in future work.

Reviewer #3 (Recommendations for the authors):The authors aimed at a low-cost technical solution for large-array gigapixel live imaging for biological applications. The endeavor is highly appreciated by the community as existing devices that are capable of such high volume of biological imaging are unaffordable or inaccessible by ordinary biological research labs. However, their current manuscript presented with design and engineering caveats that causally resulted in poor biological signal acquisition performance, as follows:

We would like to first thank the reviewer for their kind comments regarding our work. Indeed, a main motivation of our efforts are to make high volume (i.e., large area, high resolution) biological imaging more accessible and affordable.

1) The low numerical aperture (NA, = 0.03) is inacceptable for imaging applications that require cellular resolution. The statement of the authors that their device can acquire cellular-level neuronal activities is unsupported by both optical principle and their demonstrated imaging data (Figure 3). This is due to the authors' optical design using simple lens per each sensor chip. Correct design strategy should be combinatorial, i.e., one set of lens per a subarray of sensor chips, that the optical NA can be sufficiently high to satisfy realistic cellular resolution in living organisms.

We thank the reviewer for this point, and we certainly acknowledge the limited resolution of our first prototype. When imaging with one lens per image sensor, an appropriate working distance must be selected to ensure that all the fields-of-view (FOV) of the cameras overlap. This in turn limits the per-lens numerical aperture (NA) We further reduced the NA to demonstrate stereo-based detection. As discussed, and sketched in Appendix 1 Section 2, a tighter inter-camera spacing leads to the ability to increase the NA while still maintaining continuous FOV coverage, as does giving up on stereo-based detection. And of course, one may design a microscope array to not achieve continuous FOV coverage, in which case the per-camera lens NA can be made arbitrarily large. In an updated prototype design, we have reduced the inter-camera spacing from 18 mm to 9 mm, which among other changes has allowed us to increase the numerical aperture by approximately a factor of 2 while still maintaining continuous FOV coverage. If the design used multiple sensors behind a single lens, one would observe gaps in the FOV, as it is not possible to have the separate sensor chips exactly line up against one another (i.e., there are inevitable gaps). We have added the following text to the Discussion to make the limited resolution of our approach clearer to the reader:

“Finally, while the demonstrated lateral resolution in this work (18 µm full-pitch) is typically not sufficient to resolve individual cells unless they are sparsely labeled, tighter camera packing and updated imaging optics may unlock the ability to observe cellular-scale detail in future designs.”

2) The image data streaming rate as 1Hz for full-frame Gigapixel image is too low for most biological applications that require temporal resolution to study behavior or cellular dynamics. However, this is an engineering bottleneck, not methodological. As a single sensor chip used in this study, the OMNIVISION's OV10823, does indeed support full-frame streaming at 30 Hz – which is nowadays commonly accepted in biological imaging community as video-rate imaging. The bottleneck is caused by the choice of low-cost USB3.0 transfer protocol and a single PC workstation. Solution to this issue is simple: parallelize the data transfer by using multiple PCs with either timestamp markers (e.g., synchronized indicator LEDs through the optical system per each sensor) or synchronized triggered acquisition. There is of course, a new concern to balance the cost/performance ratio.

Yes, the reviewer is correct that the demonstrated 1Hz frame rate of our prototype MCAM is relatively slow for many applications of interest, but it is not a fundamental limitation. As we responded to Reviewer 2’s first comment, USB data transfer rates have significantly improved since the planning and construction of the demonstrated MCAM prototype. Recent revisions of the USB protocol now allow up to 2500 MB/sec (2.5 GB/sec) transfer on a single link (See Ref. [1] in response to R2 Q1). With our prototype’s 4X USB link approach, this would allow us to increase the MCAM’s total data rate by a factor of 10.4X. This would in turn enable 10.4 Hz video capture from the same MCAM sensor architecture on a single desktop computer, thus without significantly increasing the complexity or cost of the system. To reach 30 Hz capture, as the reviewer suggests, we could further distribute data transmission across multiple PC workstations (potentially 3 workstations) with a small increase to system cost/complexity. To address these points, we have added the following text to the Discussion section:

“…we note here that a number of strategies are available to increase sampling rates. The most straightforward paths to directly increasing the MCAM frame rate are (1) utilizing an updated version of USB that now allows approximately 10X faster data transfer, (2) distributing data transmission across multiple desktop computers, (3) adopting an alternative FPGA data transfer scheme (e.g., replacing USB data transmission with a faster PCIe protocol), and (4) executing on-chip pixel binning, image cropping, and/or lossless compression to increase temporal sampling rates at the expense of spatial sampling. Various combinations of the above strategies are currently being explored to facilitate true video-rate (24-30 Hz) MCAM imaging and to achieve significantly higher frame rates and improve cross-camera snapshot synchronization.”

3) the claim of 'cellular-level' signal or resolution has to be removed throughout the manuscript as the authors did not provide any evidence that what they claim as cellular objects or signals are indeed from individual cells. Proper wording for this type of low-resolution, low-sensitivity signal can be: cell population activity.

We appreciate this concern and have removed claims of 'cellular-level' signal or resolution from throughout the manuscript. Where appropriate, we now use the terms “cell population activity” and “structures that approach the cellular-scale”.

4) The authors made major efforts in applying machine learning algorithms for identifying and tracking objects in the stitched Gigapixel image, however, the authors did not consider the cost for implementing such working pipelines to handle such high volume of raw imaging data for obtaining biologically meaningful signals and analysis results. In particular, for tracking just a few animals as the authors presented in data, how much gain in performance is enabled by their device as compared to existing well-established methods using a single or few imaging sensors?These concerns together suggest that the authors' design and engineering did not enable their claim of expected performance. Low-cost device is good for advancing science, however, only in condition when scientific criteria can be met.

We thank the reviewer for this excellent point and should use it to clarify our initial strategy and future directions. Our initial strategy in the current paper was not to adapt algorithms such as faster-RCNN to work as quickly on our large image frames as it works on standard small images -- typically such algorithms are applied to images smaller than one megapixel (1000 x 1000 pixels). Rather, our goal in this work was instead to adapt such off-the-shelf deep learning algorithms to work offline on images that are *many* orders of magnitude larger than one megapixel. This adaption process was non-trivial and required several novel insights that are outlined in “Supplemental Section: Large-scale object detection pipeline”. We also created an open-source code repository for others to use to apply their frozen models to extremely large images. Last, and most importantly, the object tracking algorithm that we developed is directly parallelizable: it is applied across different patches of the image independently that are later combined using a very fast (intersection over union) operation. Accordingly, while we did not demonstrate it in this work, our object detection process could be run across multiple GPUs in parallel for a significant speed increase, which would be an interesting future direction. We did clearly state this in the original manuscript, so have added this possibility in the discussion of the network in Appendix 1, and thank the reviewer for pushing us toward this realization:

“Note that within this modified faster-RCNN architecture, each sub-image window is processed independently, so this algorithm could easily be parallelized across multiple GPUs for significant speed gains.”